# Preference Optimization on Pareto Sets:
# On a Theory of Multi-Objective Optimization

**Abhishek Roy**[*]
Texas A&M University
abhishekroy@tamu.edu

**Geelon So**[*]
UC San Diego
geelon@ucsd.edu

**Yi-An Ma**
UC San Diego
yianma@ucsd.edu

## Abstract

In multi-objective optimization, a single decision vector must balance the trade-offs across many objectives. Pareto-optimal solutions are those achieving optimal trade-offs, where improving any objective comes at a cost to another. As many different decisions can be Pareto optimal, this raises the question of which solution to pick and how. We formulate this problem as one of optimizing a preference function over the set of Pareto-optimal solutions, or *Pareto-constrained optimization* for short. It poses significant challenges: not only is the constraint set defined implicitly, but it is also generally non-convex and non-smooth, even when the objectives are strongly convex. We propose an equivalent formulation of the problem where the constraint set is the simplex, leading to clearer notions of optimality and stationarity that improve upon existing definitions in literature. We give an algorithm with a last-iterate convergence rate of $O(K^{-1/2})$ to stationarity when the preference function is Lipschitz smooth and when the objective functions are strongly convex and Lipschitz smooth. Motivated by applications like Reinforcement Learning with Human Feedback (RLHF), we also extend this algorithm to the case where access to the preference function is only available through dueling feedback.

## 1 Introduction

Modern, large-scale machine learning often draws data from diverse sources, are simultaneously deployed across many settings to perform many tasks, or need to perform well across many different metrics. These learning settings can naturally be formulated as *multi-objective optimization* (MOO) problems, including multi-task, multi-distribution, meta-learning; multi-calibration, multi-group learning; learning from crowds, or from heterogeneous/multi-fidelity sources; personalization, preference learning; hyperparameter optimization, model fine-tuning and model fusion (Jin, 2007; Sener and Koltun, 2018; Huang et al., 2015; Haghtalab et al., 2022; Ye et al., 2021; Reed et al., 2022; Martinez et al., 2020; La Cava, 2023; Kamani et al., 2021; Globus-Harris et al., 2022; Lee et al., 2022; Tosh and Hsu, 2022; Raykar et al., 2010; Chen et al., 2025a). As a result, MOO has increasingly attracted the interest of the learning community. But, in contrast to the single-objective case, far less has been established theoretically and algorithmically for the multi-objective setting—even very basic questions may not have rigorous definitions or solutions. We formalize and tackle such a problem.

To give an intuitive problem description, consider the concrete example of large language model (LLM) alignment (Houlsby et al., 2019; Liu et al., 2022; Ding et al., 2023). In this problem, we aim to finetune LLM outputs to achieve a number of desiderata: positivity of the tone, succinctness of the answer, consistency with lexical conventions, and so on. While no model is generally optimal under all criteria, it is *Pareto optimal* if it makes an optimal trade-off between the objectives. We would like our decision to at least be Pareto optimal, as this means no other decision was strictly better by all metrics. Many different trade-offs can usually be made; we call the set of all such decisions the

---

[*]Equal contribution.

39th Conference on Neural Information Processing Systems (NeurIPS 2025).

*Pareto set*. Eventually, as we must single out one from this set to deploy, we ask how to select the most preferred Pareto-optimal solution. That is, out of the many different possible trade-offs that can be made, which one do we choose and how do we discover it?

There are two main frameworks to making this selection (Hwang and Masud, 2012). The first is through *scalarization*, where multiple objectives are aggregated into one. Thus, it reduces the problem back to the familiar single-objective setting (Mahapatra and Rajan, 2020; Lin et al., 2024). The second, indirect approach is to find a representative *subsample* of the Pareto set; this helps a decision maker by paring down the number of solutions that need to be inspected (Lin et al., 2019; Liu et al., 2021; Kobayashi et al., 2019; Guerreiro et al., 2021).

Neither approach quite fully answers to how to select the final trade-off. It might not be clear how to meaningfully scalarize a multi-objective problem, especially when the objectives are incomparable. And while subsampling can reduce the complexity of the problem, the actual decision is left open. For classical settings, these gaps may not be crucial. But modern, large-scale decision-making settings can be high-dimensional, incorporate many objectives, or have high-throughput and need to be automatic. It may be impossible to scalarize the problem by hand, and even a representative subsample of the Pareto set can become untenably large, scaling exponentially with the number of objectives (Papadimitriou and Yannakakis, 2000). There is a need for a more principled selection, which motivates the question we ask:

> *Given a set of objectives $(f_1, \ldots, f_n)$ and a preference $f_0$, how do we define a suitable notion of the most preferred Pareto solution, and how do we efficiently approximate it?*

The **preference-based formulation** of multi-objective optimization, with the inclusion of a preference function $f_0$, refines more common versions of MOO given in standard references (e.g., Ehrgott (2005)). While the objectives $f_1, \ldots, f_n$ and the preference $f_0$ are mathematically the same type of objects, the latter can conceptually represent the more ineffable desiderata that human decision-makers or users may have, as in the earlier example about LLM alignment. For this reason, we also consider learning from *dueling preference feedback* in this paper. In that setting, instead of direct access to $f_0$, we can only ask users to *rank* different options $x$ according to their preferences $f_0(x)$.

Besides LLM finetuning, several standard problems in machine learning also fit this preference-based framework of MOO. In fairness-aware learning, the objectives can represent subgroup utilities, while the preference $f_0$ is a social welfare function. In neural architecture search, objectives can be accuracy, latency, and energy consumption while $f_0$ can be user-specific preferences reflecting their priorities. In portfolio optimization, objectives could be return, risk, and liquidity, while $f_0$ might capture the risk aversion tendency of an investor. We now formalize the problem mathematically.

## 1.1 Pareto-Constrained Optimization

Let $F \equiv (f_1, \ldots, f_n) : \mathbb{R}^d \to \mathbb{R}^n$ be a set of $n$ objectives that are jointly minimized over a shared decision space $\mathbb{R}^d$ and let $\mathrm{Pareto}(F)$ be the set Pareto-optimal solutions, consisting of decision vectors $x \in \mathbb{R}^d$ that make an optimal trade-off between objectives. As it is often not clear *a priori* which trade-off to make, we consider the *Pareto-constrained optimization problem*, where the aim is to optimize a preference function $f_0 : \mathbb{R}^d \to \mathbb{R}$ constrained to the Pareto set of $F$:

$$\min_{x \in \mathrm{Pareto}(F)} f_0(x). \tag{1}$$

and we call a solution $x$ of this optimization problem *preference optimal*. This problem is also called *semivectorial bilevel optimization* or *optimization on efficient sets*, and it can be considered as an instantiation of bi-level optimization (Bolintinéanu, 1993b; Yamamoto, 2002; Bonnel and Morgan, 2006; Dempe, 2018; Ye and Liu, 2022). It has applications for economics, portfolio management, manufacturing planning, and machine learning (Thach et al., 1996; Yamamoto, 2002; Ye and Liu, 2022). While heuristics have been proposed, little more is known. In fact, even notions such as stationarity have not been formalized, which is generally needed to study convergence.

**Main Challenges** There are major challenges to this problem.

1. *Theoretical Challenges.* The Pareto set is generally non-smooth and non-convex. It can have "needle-like extensions" and "knees" (Kulkarni et al., 2022), or "singularities" (Sheftel et al., 2013). These are possible even when the objectives are quadratics (see Figure 1). Moreover,

in the setting where both objectives and preferences are linear, the problem is known to be NP-hard (Fülöp, 1993). Thus, there is a need for appropriate relaxations of the problem that are algorithmically attainable. But even defining a reasonable notion of stationarity is not straightforward, given the complicated nature of the constraint set (Zhang et al., 2020; Kornowski and Shamir, 2021; Li et al., 2020; Jordan et al., 2023; Kornowski et al., 2024).

2. *Algorithmic Challenges.* The Pareto set is non-smooth, non-convex, and an implicitly defined object. It is not clear how to analytically parameterize $\mathrm{Pareto}(F)$ or to specify $\mathrm{Pareto}(F)$ as a feasible set of a system of inequalities. Pareto-constrained optimization (1) becomes even more challenging when we do not have access to the preference function $f_0$, but only preference comparisons between two decision vectors, as is the case of RLHF for LLM alignment.

## 1.2 Main Results

We consider Pareto-constrained optimization with strongly convex and smooth objectives $f_1, \ldots, f_n$, and smooth, but potentially nonconvex preference function $f_0$. We list our main contributions below.

1. We introduce the *Pareto manifold* of $F$ (Definition 3), a '*lifting*' of the Pareto set, that recovers the Pareto set when projected down to $\mathbb{R}^d$. We show that the Pareto manifold is a *smooth* manifold diffeomorphic to the $(n-1)$-simplex. This leads to a clear notion of stationarity.

2. We use the connection with the simplex to introduce an (approximate) stationarity condition (Definitions 3 and 5) for the Pareto-constrained optimization. We show that any non-trivial, local stationarity condition requires more than local first-order information about $F$.

3. We propose the Pareto Majorization-Minimization algorithm (Algorithm 1), which converges to an $(\varepsilon_0, \varepsilon)$-approximate preference stationary point of Pareto-constrained optimization with *iteration complexity* $\tilde{O}(\varepsilon_0^{-2})$, ignoring logarithmic factors, under *first-order feedback* $\nabla f_0$ (Theorem 10), and noisy *dueling preference feedback* (Theorem 11).

## 1.3 Related Work

Selecting the most-preferred decision vector out of the Pareto set is a classical problem in MOO (see Bolintinéanu (1993b) and related works therein for classical motivation), and it has also gained renewed interest from the machine learning community. However, it is well-established to be challenging (e.g. Fülöp (1993)), and most prior work studying the Pareto-constrained optimization problem have largely focused on (i) linear preferences (Philip, 1972; Benson, 1984; Liu and Ehrgott, 2018), (ii) linear objectives (Dauer, 1991; Bolintinéanu, 1993a; Tao et al., 1996; Yamamoto, 2002), or (iii) specific choices of preference functions (Steuer, 1989; Mahapatra and Rajan, 2020).

To our knowledge, there are only two prior works that have studied the problem more generally in the nonlinear setting. The first work is Bolintinéanu (1993b), which considers a regularized version of the problem: they balance the preference $f_0$ with a penalty term capturing the Pareto set of $F$. While regularized solutions are suboptimal for the original problem, they show that these solutions asymptotically become optimal as the weight on the penalty term goes to infinity. They also describe a necessary condition for the regularized solution, which can be approximated via nonlinear programming. The second work is Ye and Liu (2022), which provides a heuristic for the same problem based on a similar but distinct penalty function. They also propose a stationary condition—stationarity with respect to their optimization dynamics. But, this turns out to not be a necessary condition, meaning that their dynamics can actively avoid optimal points (see Appendix D).

These two conditions in both papers entangle independent aspects of an ideal solution: (a) being close to preference optimal and (b) being close to the Pareto set. As a result, these conditions can seem somewhat opaque. In this work, we first clarify the manifold structure of the Pareto set under strong convexity of the objectives. This enables us to keep these two aspects disentangled, and to derive standard, necessary relaxations of preference optimality. While existing work have studied the smoothness structure of the Pareto set (Hillermeier, 2001a,b; Hamada et al., 2020), the prior focus has been on extrinsic smoothness within the decision space. We instead leverage the intrinsic geometry of the Pareto manifold, which is diffeomorphic to the simplex, to simplify optimization.

The algorithm we propose along with its analysis draw on majorization-minimization and trust-region ideas to handle the implicit nature of the problem (Lange et al., 2000; Marumo et al., 2023). This algorithm can make use of both first-order preference information or dueling feedback. For dueling

feedback, we work under a standard preference learning model (see Section 6.2) from psychology, statistics, and also more recent learning literature (Bradley and Terry, 1952; Agresti, 2012; Wang et al., 2023). For this setting, we also make use of ideas from derivative-free or zeroth-order optimization (Jamieson et al., 2012; Saha et al., 2025; Cai et al., 2022). In terms of guarantees, we provide the first finite-time convergence result to approximate preference stationarity. Additionally, we provide non-asymptotic guarantees on the suboptimality of approximately preference-optimal solutions, which parallel the asymptotic guarantee of Bolintinéanu (1993b).

Following an earlier version of this article shared on arXiv,[2] Chen et al. (2025b) developed results for the Pareto-constrained optimization problem that goes beyond strictly-convex objectives.

**Organization** In Section 2 we discuss some preliminaries on Pareto set. In Section 3, we introduce the Pareto manifold. In Section 4, we define preference stationarity. In Section 5, define approximate preference stationarity and formalize our assumptions. In Section 6, we present Algorithm 1, which solves Pareto-constrained optimization under two feedback models: (a) access to a first-order feedback, and (b) access to preference comparisons. In Section 7, we provide rates of convergence. For a detailed glossary, see Appendix A. Full proofs are provided in the appendix.

## 2 The Pareto Set

From multi-objective optimization, recall that a decision $x$ is Pareto optimal if there is no way to improve any one $f_i$ without also worsening some other $f_j$. Formally:

**Definition 1** (Pareto optimality). *Given objectives $f_1, \ldots, f_n$, we say that a decision vector $x \in \mathbb{R}^d$ is* Pareto optimal *if for all $x' \in \mathbb{R}^d$:*

$$f_i(x') < f_i(x) \qquad \Longrightarrow \qquad \exists j \quad \text{s.t.} \quad f_j(x') > f_j(x).$$

*The set of Pareto optimal decision vectors of $f_1, \ldots, f_n$ forms the* Pareto set*, denoted $\mathrm{Pareto}(F)$.*

For smooth objectives, a related, first-order condition called *Pareto stationarity* is necessary for Pareto optimality Maruşciac (1982). In the following, let $\Delta^{n-1}$ denote the $(n-1)$-simplex, and for all $\beta \in \Delta^{n-1}$, let $f_\beta$ denote the *linear scalarization*:

$$f_\beta(x) := \beta_1 f_1(x) + \cdots + \beta_n f_n(x). \tag{2}$$

**Definition 2** (Pareto stationarity). *Given objectives $f_1, \ldots, f_n$, we say that a decision $x \in \mathbb{R}^d$ is* Pareto stationary *if $\nabla f_\beta(x) = 0$ for some $\beta \in \Delta^{n-1}$.*

When the objectives $f_1, \ldots, f_n$ are twice-differentiable and strictly convex, Pareto stationarity is also necessary for Pareto optimality Fliege et al. (2009). We shall assume throughout that the objectives are smooth and strongly convex (Assumption A), so these two notions coincide. Given these definitions, optimization constrained to the Pareto set as defined in (1) is highly non-trivial: the Pareto set remains implicit, and it is generally non-smooth and non-convex. The following gives the example in Figure 1.

**Example 1** (A singular Pareto set arising from strongly convex objectives). *Define three positive-definite quadratic objectives $f_i : \mathbb{R}^2 \to \mathbb{R}$ of the form $f_i(x) = \frac{1}{2}(x - x_i)^\top A_i(x - x_i)$,*

$$A_1 = \begin{bmatrix} 1 & 0 \\ 0 & 1 \end{bmatrix} x_1 = \begin{bmatrix} 0 \\ 0 \end{bmatrix}, \quad A_2 = \begin{bmatrix} 0.25 & 0 \\ 0 & 1 \end{bmatrix} x_2 = \begin{bmatrix} 1 \\ 0.5 \end{bmatrix}, \quad A_3 = \begin{bmatrix} 1 & 0 \\ 0 & 0.25 \end{bmatrix} x_3 = \begin{bmatrix} 0.5 \\ 1 \end{bmatrix}.$$

*Even in this highly-structured setting where all objectives are strongly convex quadratics, the Pareto set is non-convex and has a singularity. This example is visualized in Figure 1.*

## 3 Lifting to the Pareto Manifold

We can overcome the issue of non-smoothness of the Pareto set by lifting the problem into a higher-dimensional space $\mathbb{R}^d \times \Delta^{n-1}$, which contains what we call the *Pareto manifold*. This is the set of all tuples $(x, \beta)$ such that $x$ is a Pareto stationary point of the linearly scalarized objective $f_\beta$.

**Definition 3** (Pareto manifold). *The Pareto manifold $\mathcal{P}(F) \subset \mathbb{R}^d \times \Delta^{n-1}$ is the zero set:*

$$\mathcal{P}(F) = \big\{ (x, \beta) : \nabla f_\beta(x) = 0 \big\}.$$

---

[2]Earlier title: *Optimization on Pareto Sets: On a Theory of Multi-Objective Optimization* (Roy et al., 2023).

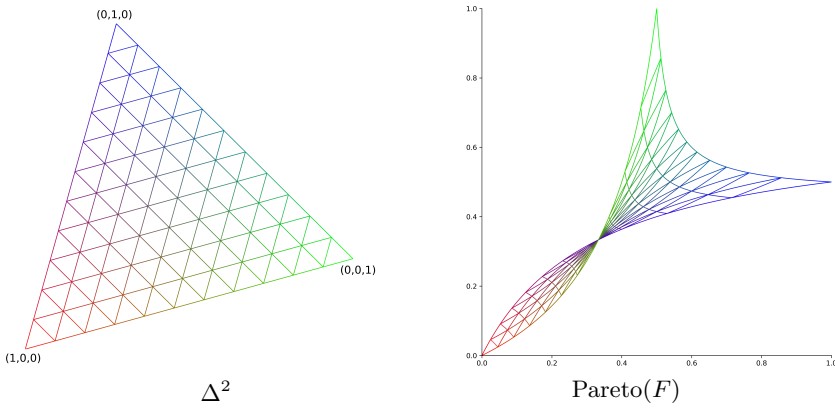

Figure 1: The Pareto set of the three positive-definite quadratic objectives given by Example 1. (Left) The 2-simplex parametrizes all scalarizations $f_\beta$. (Right) The Pareto set in $\mathbb{R}^2$. For each $\beta \in \Delta^2$, the minimizer $x^*(\beta)$ of $f_\beta$ is Pareto optimal; the colors preserve this correspondence.

In the case of strictly convex objectives, the projection of the Pareto manifold $\mathcal{P}(F)$ to its first component in the decision space $\mathbb{R}^d$ precisely yields the Pareto set, as shown by Lemma 1. As this projection can also collapse any smoothness structure that $\mathcal{P}(F)$ has, it becomes clear how non-smoothness arises on the Pareto set. On the other hand, Proposition 2 shows that it is a smooth submanifold of $\mathbb{R}^d \times \Delta^{n-1}$ diffeomorphic to the $(n-1)$-simplex.

**Lemma 1.** *Let $F \equiv (f_1, \ldots, f_n)$ be a collection of smooth and strictly convex objectives. Then:*

$$x \in \text{Pareto}(F) \quad \Longleftrightarrow \quad (x, \beta) \in \mathcal{P}(F) \quad \text{for some } \beta \in \Delta^{n-1}.$$

**Proposition 2** (Characterization of the Pareto manifold)**.** *Let $F \equiv (f_1, \ldots, f_n)$ be a collection of smooth and strictly convex objectives. Define $x^* : \Delta^{n-1} \to \text{Pareto}(F)$ by:*

$$x^*(\beta) \equiv x_\beta := \arg\min_{x \in \mathbb{R}^d} f_\beta(x). \tag{3}$$

*Let $\nabla F(x) \in \mathbb{R}^{n \times d}$ be the Jacobian. Then, the map $x^*$ has derivative:*

$$\nabla x^*(\beta) = -\nabla^2 f_\beta(x_\beta)^{-1} \nabla F(x_\beta)^\top, \tag{4}$$

*so that the map $\beta \mapsto (x_\beta, \beta)$ is a diffeomorphism of $\Delta^{n-1}$ with the Pareto manifold $\mathcal{P}(F)$.*

The main tool used is the implicit function theorem; we defer the full proofs to Appendix B.

## 4 Solution Concept: Preference Stationarity

The tight connection between the Pareto set and the Pareto manifold given by Lemma 1 allows us to smoothly lift the preference optimization problem (1) onto the Pareto manifold, as follows:

$$\min_{(x, \beta) \in \mathcal{P}(F)} f_0(x). \tag{5}$$

And as Proposition 2 shows, the Pareto manifold is diffeomorphic to the simplex, so (5) further reduces to the following smooth optimization over the convex set $\Delta^{n-1}$.

$$\min_{\beta \in \Delta^{n-1}} (f_0 \circ x^*)(\beta). \tag{6}$$

This is an equivalent formulation: if $\beta$ solves (6), then $x^*(\beta)$ solves (1) and (5). Furthermore, as the constraint set is the simplex, we can appeal to convex optimization for the following notion of *preference stationarity*, which is necessary for preference optimality (Nesterov, 2003).

**Definition 3** (Preference stationarity)**.** *We say that a point $x \in \text{Pareto}(F)$ is* weakly preference stationary *if there exists some $\beta \in \Delta^{n-1}$ where $(x, \beta) \in \mathcal{P}(F)$ such that:*

$$-\nabla(f_0 \circ x^*)(\beta)^\top (\beta' - \beta) \leq 0, \qquad \forall \beta' \in \Delta^{n-1}, \tag{7}$$

*with $\nabla x^*$ as in (4).[3] If (7) holds for all $\beta$ where $(x, \beta) \in \mathcal{P}(F)$, then $x$ is* preference stationary.

---

[3]As $\nabla F(x_\beta)^\top \beta = \nabla f_\beta(x_\beta) = 0$, (7) reduces to $-\nabla(f_0 \circ x^*)(\beta)^\top \beta' \leq 0$, for all $\beta' \in \Delta^{n-1}$.

**Proposition 4** (Necessary condition (Lemma 3.1.19 of Nesterov (2003))). *Preference optimality implies (weak) preference stationarity.*

It turns out that developing reasonable relaxations of preference optimality without lifting the problem to Pareto manifold is surprisingly subtle. For example, a prior notion of stationarity (e.g., Ye and Liu (2022)) is not necessary for preference optimality (Appendix D.1). Any optimization dynamics that achieve stationarity conditions that are not necessary can actively avoid optimal decisions. We show that any local first-order condition suffers from this limitation (Appendix D).

## 5 Algorithmic Solution Concept: Approximate Preference Stationarity

In first-order optimization, notions of *approximate* stationarity allow us to provide meaningful, finite-time guarantees about the solutions that are reached by optimization algorithms in practice; likewise, we introduce the following $\varepsilon$-approximate version of preference stationarity. Again, we appeal to a standard notion of approximate solutions Nesterov (2013); Marumo et al. (2023).

**Definition 5** (Approximate preference stationarity). *Let $\varepsilon_0, \varepsilon \geq 0$. A point $(x, \beta) \in \mathbb{R}^d \times \Delta^{n-1}$ is $(\varepsilon_0, \varepsilon)$-preference stationary if:*

$$-\nabla f_0(x_\beta)^\top \nabla x^*(\beta)(\beta' - \beta) \leq \varepsilon_0 \|\beta' - \beta\|_1, \qquad \forall \beta' \in \Delta^{n-1}. \tag{8a}$$

$$\|\nabla f_\beta(x)\|_2 \leq \varepsilon \tag{8b}$$

Intuitively, if $(\hat{x}, \hat{\beta})$ is approximately preference stationary, then (a) there is a ball around $\hat{\beta}$ within which $f_0 \circ x^*$ decreases at most at an $O(\varepsilon_0)$-rate when moving away from $\hat{\beta}$, and (b) the point $\hat{x}$ is $O(\varepsilon)$-close to $x_{\hat{\beta}}$. We will formalize this geometric interpretation in Proposition 6. To do so, we need to formalize the assumptions, which are standard and used in the rest of the paper.

**Assumption A.** *Let the objectives $f_1, \ldots, f_n : \mathbb{R}^d \to \mathbb{R}$ be twice differentiable, $\mu$-strongly convex, and have $L$-Lipschitz continuous gradient. That is, $\mu \mathbf{I} \preceq \nabla^2 f_i(x) \preceq L\mathbf{I}$ for all $i = 1, \ldots, n$. We also define $\kappa := L/\mu$ and $r := \max_{i,j \in [n]} \|\arg\min f_i(x) - \arg\min f_j(x)\|_2$.*

**Assumption B.** *Let the objectives $f_1, \ldots, f_n : \mathbb{R}^d \to \mathbb{R}$ have $L_H$-Lipschitz continuous Hessian. That is, for all $x, y \in \mathbb{R}^d$ and $i = 1, \ldots, n$, we have $\|\nabla^2 f_i(x) - \nabla^2 f_i(y)\|_2 \leq L_H \|x - y\|_2$.*

**Assumption C.** *Let the preference function $f_0 : \mathbb{R}^d \to \mathbb{R}$ have $L_0$-Lipschitz continuous gradient. That is, for all $x, y \in \mathbb{R}^d$, we have $\|\nabla f_0(x) - \nabla f_0(y)\|_2 \leq L_0 \|x - y\|_2$.*

Assumption A allows us to bound the diameter of the Pareto set (Lemma E.1); Assumption B bounds the curvature of the Pareto manifold (Lemma E.2); and all three lead to bounds on approximation errors (Lemma E.5 and Lemma E.6) needed as the constraint set is implicit. They also lead to the following geometric intuition for approximate preference stationarity (see Appendix E for proof).

**Proposition 6** (Geometric meaning of approximate stationarity). *Let $(\hat{x}, \hat{\beta})$ be $(\varepsilon_0, \varepsilon)$-preference stationary. With Assumptions A,B,C, and $R := \mathrm{diam}(\mathrm{Pareto}(F))$ and $s := 2\mu^2\varepsilon_0/(L_0 L^2 R^2)$, then:*

*(a) if $\|\beta - \hat{\beta}\|_1 \leq s$, then $f_0(x_\beta) - f_0(x_{\hat{\beta}}) \geq -2\varepsilon_0\|\beta - \hat{\beta}\|_1$, and*

*(b) $\|\hat{x} - x_{\hat{\beta}}\|_2 \leq \varepsilon/\mu$.*

### 5.1 Overcoming Implicitness: Approximating Local Information

The final issue is that these notions of (approximate) preference stationarity are *implicit*. They depend on $\nabla x^*(\beta)$ and $x^*$, which come from solving the optimization problem:

$$x^*(\beta) \equiv x_\beta = \arg\min_{x \in \mathbb{R}^d} f_\beta(x), \tag{3}$$

$$\nabla x^*(\beta) = -\nabla^2 f_\beta(x_\beta)^{-1} \nabla F(x_\beta)^\top. \tag{4}$$

Since $x_\beta$ does not generally have a closed form, we cannot exactly compute $\nabla x^*(\beta)$, which is needed to directly check whether the pair $(\hat{x}, \hat{\beta})$ is (approximately) preference stationary. Since we cannot compute $\nabla x^*(\beta)$ exactly, an obvious estimator is the following, which uses local information $\nabla^2 f_\beta(x)$ and $\nabla F(x)$ at $x$ as a proxy for the corresponding local information at $x_\beta$:

$$\widehat{\nabla} x^*(x, \beta) := -\nabla^2 f_\beta(x)^{-1} \nabla F(x)^\top. \tag{10}$$

---

**Algorithm 1** Pareto majorization-minimization (PMM)

---

**Input:** objectives $F \equiv (f_1, \ldots, f_n)$; preference function $f_0$; black-box optimizer $\widehat{\arg\min}$; a family of majorizing surrogates $\{g(\cdot\,; x, \beta) : (x, \beta) \in \mathbb{R}^d \times \Delta^{n-1}\}$, exact (13) or approximate (16)
**Initialize:** $(x_0, \beta_0) \in \mathbb{R}^d \times \Delta^{n-1}$

1: **for** $k = 1, \ldots, K$ **do**
2:      Select the (approximate) majorizing surrogate $g_k(\cdot) \leftarrow g(\cdot\,; x_k, \beta_k)$
3:      Compute approximate minimizers

$$\beta_{k+1} \leftarrow \widehat{\arg\min_{\beta \in \Delta^{n-1}}} g_k(\beta) \quad \text{and} \quad x_{k+1} \leftarrow \widehat{\arg\min_{x \in \mathbb{R}^d}} f_{\beta_{k+1}}(x). \tag{9}$$

4: **end for**
5: **return** $(x_{K+1}, \beta_{K+1})$

---

With the above assumptions, approximate information is enough to verify approximate stationarity. Again, we defer proofs to Appendix E.

**Lemma 7** (Verifiability of approximate stationarity). *Under Assumptions A,B,C, the point $(\hat{x}, \hat{\beta})$ is $(\varepsilon_0, \varepsilon)$-preference stationary if $\|\nabla f_{\hat{\beta}}(\hat{x})\|_2 \leq \varepsilon$, and for some $x \in \mathbb{R}^d$ and $\alpha \in (0, 1)$,*

*(a) an $\alpha \cdot \varepsilon_0$-approximate stationary condition holds for all $\beta' \in \Delta^{n-1}$:*

$$-\nabla f_0(x)^\top \widehat{\nabla} x^*(x, \hat{\beta})(\beta' - \hat{\beta}) \leq \alpha \cdot \varepsilon_0 \|\beta' - \hat{\beta}\|_1, \tag{11}$$

*(b) an error bound holds:* $\mathrm{err}_{\nabla f_0}(\hat{\beta}, x) \leq (1 - \alpha) \cdot \varepsilon_0$, *where:*

$$\mathrm{err}_{\nabla f_0}(x, \beta) := \frac{1}{\mu}\left(\frac{M_1}{2M_0}\|\nabla f_0(x)\|_2 + L_0 M_0\right)\|\nabla f_\beta(x)\|_2,$$

*and $M_0 = \kappa R$, $M_1 = 2\kappa^2 R(1 + L_H R/\mu)$.*

## 6 Algorithm Design: Pareto Majorization-Minimization

In this section, we present our algorithm to solve (6). While this is conceptually the optimization of a smooth function $f \circ x^*$ over the convex set $\Delta^{n-1}$, the main challenge is that the objective is implicit. Methods like gradient descent require both $x^*$ and $\nabla x^*$, to compute the gradient via chain rule:

$$\nabla(f_0 \circ x^*)(\beta) = \nabla f_0\big(x^*(\beta)\big)^\top \nabla x^*(\beta).$$

But, if we can estimate $x^*(\beta)$ to arbitrary precision by solving the optimization in (3), then $\nabla x^*(\beta)$ can also be approximated arbitrarily well: both of its terms, $\nabla F(x)$ and $\nabla^2 f_\beta(x)^{-1}$, are continuous in $x$ and $\beta$ by assumption. Thus, the estimator $\widehat{\nabla} x^*(x, \beta)$ defined in (10) approaches $\nabla x^*(\beta)$ as $x$ approaches $x_\beta$. We will quantify the validity of the estimator by using it to construct a majorizing surrogate function, a function that upper bounds $f_0 \circ x^*$, where better estimators yield tighter bounds.

**Definition 4** (Majorizing surrogate). *A function $g : \Delta^{n-1} \to \mathbb{R}$ majorizes $f_0 \circ x^*$ if:*

$$f_0(x_{\beta'}) \leq g(\beta'), \tag{12}$$

*for all $\beta' \in \Delta^{n-1}$. We say that $g$ is a (majorizing) surrogate of $f$.*

The following uses the estimator $\widehat{\nabla} x^*(x, \beta)$ to construct a family of quadratic surrogates for $f_0 \circ x^*$. An error term $\mathrm{err}_{\nabla f_0}(x, \beta)$ appears as a constant in (13), which Lemma E.6 shows goes to zero as $(x, \beta)$ goes to $(x_\beta, \beta)$. Thus, the estimator becomes more 'valid' with better estimates of $x^*(\beta)$.

**Proposition 8** (A family of majorizing surrogates). *Let $F$ and $f_0$ satisfy Assumptions A,B,C. Let $\mathrm{err}_{\nabla f_0}(x, \beta)$ be as defined in Lemma E.6. Define the family indexed by $(x, \beta) \in \mathbb{R}^d \times \Delta^{n-1}$:*

$$g(\beta'; x, \beta) := f_0(x_\beta) + \nabla f_0(x)^\top \widehat{\nabla} x^*(x, \beta)(\beta' - \beta) + \tfrac{1}{2}\mu_g \|\beta' - \beta\|_2^2 + \mathrm{err}_{\nabla f_0}(x, \beta), \tag{13}$$

*where $\mu_g := n L_0 M_1$. Then $g(\beta'; x, \beta)$ majorizes $f_0 \circ x^*$, satisfying (12).*

This result is proved in Appendix F. Technically, we still cannot explicitly compute $g(\beta'; x, \beta)$ because it contains the term $f_0(x_\beta)$. However, to minimize $g$ using any gradient-based method, we only need $\nabla g(\beta'; x, \beta)$, which does not require the knowledge of $f_0(x_\beta)$ nor $\mathrm{err}_{\nabla f_0}(x, \beta)$.

## 6.1 Minimizing the Majorizing Surrogate with First-Order Preference Information

The first algorithm we describe assumes access to a first-order feedback to $f_0$, which allows us to compute the majorizing surrogate $g(\beta'; x, \beta)$ from Proposition 8. We then directly minimize the surrogate. The idealized *Pareto majorization-minimization* (PMM) algorithm proceeds in rounds:

1. majorization: query $\widehat{\nabla} x^*(x_k, \beta_k)$ to construct majorizing surrogate $g_k(\beta) \equiv g(\beta; x_k, \beta_k)$,
2. minimization: make updates $\beta_{k+1} \leftarrow \underset{\Delta^{n-1}}{\arg\min} \, g_k(\beta)$ and $x_{k+1} \leftarrow \underset{x \in \mathbb{R}^d}{\arg\min} \, f_{\beta_{k+1}}(x)$.

The majorizing property of the surrogate ensures that $f_0(x_{\beta_k})$ improves every iteration. In fact, there is no need to fully optimize $g_k$ and $f_{\beta_{k+1}}(x)$ in the second step. By relaxing this step, we obtain Algorithm 1 when it uses the exact surrogates (13) and any black-box optimizer. In Section 7, we provide a last-iterate convergence rate for Algorithm 1 to an $(\varepsilon_0, \varepsilon)$-preference stationary point.

## 6.2 A Model of Dueling Feedback

In the next section, we will extend Algorithm 1 to the *dueling feedback* setting. Especially in cases of human preferences, exact forms of $f_0$ or $\nabla f_0$ are unknown, but we can ask users to rank different options by preference. Such a setting arises, for example, in supervised fine-tuning or RLHF (e.g. Zhu et al. (2023); Ziegler et al. (2019)). We assume a comparison oracle that provides noisy, binary responses to the queries of the form: *do you prefer $x_1$ over $x_2$?*

**Definition 5** (Preference feedback model). *Given two options $x_1$ and $x_2$, the comparison oracle returns a binary random variable $Y(x_1, x_2) \in \{0, 1\}$ where:*

$$\mathbb{E}\big[Y(x_1, x_2)\big] = \sigma\big(f_0(x_2) - f_0(x_1)\big), \tag{15}$$

*and $\sigma(\cdot)$ is a link function. A response $Y(x_1, x_2) = 1$ indicates that users prefer $x_1$ over $x_2$.*

This is a standard model from preference learning, psychology, and RLHF (e.g. Bradley and Terry (1952); Wang et al. (2023); Chen et al. (2025a)), and it aims to capture the phenomenon where preference judgments tend to be noisier when it is difficult to discriminate a clear winner (Nunnally and Bernstein, 1994). The *link function* $\sigma$ formalizes the relationship between the 'random component' in the observed responses to the 'systematic component' of the underlying preference values $f_0(x)$ (Nelder and Wedderburn, 1972; Agresti, 2012). For analysis, we make standard assumptions, which are satisfied by all commonly-used link functions including the logistic, softplus, and tanh functions.

**Assumption D** (Link function). *Let $B$ satisfy $B > \sup |f_0 \circ x^*|$. Let $\sigma(\cdot) : [-B, B] \to (0, 1)$ be a known link function that is a smooth, $L_1$-Lipschitz, and monotonically increasing function. We assume that $\sigma(0) = \frac{1}{2}$, and the inverse $\sigma^{-1}(p)$ is locally $L_\sigma(1 + (p(1-p))^{-1})$-Lipschitz continuous.*

## 6.3 Optimizing Preferences with Dueling Feedback

In Section 6.1, we described a version of PMM using first-order preference information, where the gradient $\nabla f_0(x)$ gives rise to the majorizing surrogate function $g(\beta; x_k, \beta_k)$ from Proposition 8. Under dueling feedback, the gradient is not directly available, but we construct an estimator $\widehat{\nabla} f_0(x)$, as described in Algorithm 2. The basic idea is to estimate $\nabla f_0(x)$ by querying preferences between the pair $x_1 = x + \gamma U$ and $x_2 = x - \gamma U$, where $U$ is a uniform-at-random unit vector and $\gamma$ is a precision parameter. A single comparison can be seen as an estimate of $p_U = \sigma(2\gamma \nabla f_0(x)^\top U)$. Since we assume knowledge of $\sigma$, we can invert this to measure $\nabla f_0(x)$. This leads to the following family of approximate majorizing surrogate, indexed by $(x, \beta) \in \mathbb{R}^d \times \Delta^{n-1}$:

$$\hat{g}(\beta'; x, \beta) := f_0(x_\beta) + \widehat{\nabla} f_0(x)^\top \widehat{\nabla} x^*(x, \beta)(\beta' - \beta) + \tfrac{1}{2}\mu_g \|\beta' - \beta\|_2^2 + \mathrm{err}_{\nabla f_0}(x, \beta). \tag{16}$$

This family of approximate surrogates can also be used in Algorithm 1. Of course, it performance depends on the quality of the estimator $\widehat{\nabla} f_0(x)$. It also requires a separate analysis, which will use the following bounds on the bias and the variance of $\widehat{\nabla} f_0(x)$, proved in Appendix H.1.

**Lemma 9** (Bias and variance of dueling gradient estimator). *Under Assumptions A to D, let $\widehat{\nabla} f_0(x)$ be defined as in (14) with $\alpha = 1/8$, $m \asymp d^4 \log(d/\varepsilon_0)/\varepsilon_0^4$, $b \asymp d/\varepsilon_0^2$, and $\gamma \asymp \varepsilon_0/d$. Then:*

$$\|\mathbb{E}[\widehat{\nabla} f_0(x)] - \nabla f_0(x)\|_2 \lesssim \varepsilon_0, \qquad \text{and} \qquad \mathbb{E}[\|\widehat{\nabla} f_0(x) - \nabla f_0(x)\|_2^2] \lesssim \varepsilon_0^2,$$

In the next section, we provide the results on the iteration complexity of Algorithm 1 with first-order feedback (Theorem 10) and dueling feedback (Theorem 11).

**Algorithm 2** Approximation of preference gradient with dueling feedback

---

**Input:** decision vector $x$, weights $\beta$, clipping threshold $\alpha$, precision $\gamma$, number of batches $b$, batch size $m$, and a comparison oracle $Y(x_1, x_2)$ satisfying (15)

1: Sample a unit vector in $\mathbb{R}^d$ uniformly at random, $U_i \sim \mathrm{Unif}(S^{d-1})$ for each $i \in [b]$
2: Query comparison oracle for $Y_{ij} := Y\big(x + \gamma U_i, x - \gamma U_i\big)$ for each $(i, j) \in [b] \times [m]$
3: Compute clipped Bernoulli parameter estimators for each $i \in [b]$ and the gradient estimator:

$$\hat{p}_i \leftarrow \mathrm{CLIP}_{[\alpha, 1-\alpha]}\left(\frac{1}{m} \sum_{j \in [m]} Y_{ij}\right) \qquad \text{and} \qquad \hat{\nabla} f_0(x) \leftarrow \frac{1}{b} \sum_{i \in [b]} \frac{d}{2\gamma} \sigma^{-1}(\hat{p}_i) \cdot U_i \qquad (14)$$

where $\mathrm{CLIP}_{[\alpha, 1-\alpha]}(p)$ is the projection of $p$ into the interval $[\alpha, 1 - \alpha]$.
4: **return** $\hat{\nabla} f_0(x)$

---

## 7 Convergence Analysis

We establish the convergence rate of Algorithm 1 in terms of the convergence guarantees of its two black-box optimizers: one for the surrogate $g(\,\cdot\,; x, \beta)$ and another for the scalarized objective $f_\beta(\cdot)$. For our convergence results, we assume that the optimizers achieve the guarantees:

**Assumption E** (Black-box optimizers). *Let $\hat{\beta}$ and $\hat{x}_{\hat{\beta}}$ be the approximate solutions that are returned by the black-box optimizers for $g(\,\cdot\,; x, \beta)$ and $f_{\hat{\beta}}(\cdot)$ in (9) of Algorithm 1:*

$$\hat{\beta} \leftarrow \widehat{\arg\min_{\beta' \in \Delta^{n-1}}} g(\beta'; x, \beta) \qquad \text{and} \qquad \hat{x}_{\hat{\beta}} \leftarrow \widehat{\arg\min_{x \in \mathbb{R}^d}} f_{\hat{\beta}}(x).$$

*We assume that there are constants $c_1, c_2 > 0$ such that:*

1. *the approximate minimizer $\hat{\beta}$ is $O(\varepsilon_0)$-approximately stationary:*

$$\forall \beta' \in \Delta^{n-1}, \qquad \nabla g(\hat{\beta}; x, \beta)(\beta' - \hat{\beta}) \leq c_1 \cdot \varepsilon_0 \|\beta' - \hat{\beta}\|_2,$$

2. *the approximate minimizer $\hat{x}_{\hat{\beta}}$ is an $O(\varepsilon_0^2)$-approximate solution: $\|\nabla f_{\hat{\beta}}(\hat{x}_{\hat{\beta}})\| \leq c_2 \cdot \varepsilon$.*

**Theorem 10** (Convergence of PMM with first-order feedback). *Suppose that $F$ and $f_0$ satisfy Assumptions A to C, and that the black-box optimizers satisfy Assumption E. Fix $0 < \varepsilon^{1/2} \leq \varepsilon_0 \leq 1$. Let $(x_k, \beta_k)_k$ be the iterates of Algorithm 1 using the family of surrogates (13). There exist $c_1(f_0, F)$ and $c_2(f_0, F)$ bounded away from zero and a stopping time $K$ such that $(f_0 \circ x^*)(\beta_k)$ is monotonically decreasing for $k \in [K]$ and $(x_K, \beta_K)$ is an $(\varepsilon_0, \varepsilon)$-preference stationary point. Furthermore:*

$$K \leq \frac{2\mu_g \cdot \big(f^* - f_*\big)}{c_1^2 \cdot \varepsilon_0^2},$$

*where $f^* := \max f_0(x)$ and $f_* = \min f_0(x)$ are optimized over the compact set $\mathrm{Pareto}(F)$.*

*Proof sketch.* We use a standard approach and show that every iteration, either we have converged to an $\varepsilon_0$-preference stationary point and can stop, or we can find a way to improve the preference by $\Omega(\varepsilon_0^2)$. To do so, we require the optimizer for the surrogate $g$ to also achieve $O(\varepsilon_0)$-approximate stationarity, which is enough to achieve an $O(\varepsilon_0^2)$-approximately optimal point (Lemmas G.4 and G.5):

$$(\hat{\beta}; x, \beta) < g(\beta^*; x, \beta) + O(\varepsilon_0^2),$$

where $\beta^*$ minimizes the surrogate. The surrogate contains an approximation error $\mathrm{err}_{\nabla f_0}(\beta, x)$. If this error term is $\Omega(\varepsilon_0^2)$, then it is possible for the surrogate to fail to either (i) decide that the current iterate $\beta$ is $\varepsilon_0$-preference stationary or (ii) make progress by finding some $\hat{\beta}$ that certifiably improves on $f_0$. We preclude this by requiring the optimizer for $f_\beta$ to achieve $O(\varepsilon_0^2)$-optimality. $\square$

The full proof of Theorem 10 is in Appendix G.

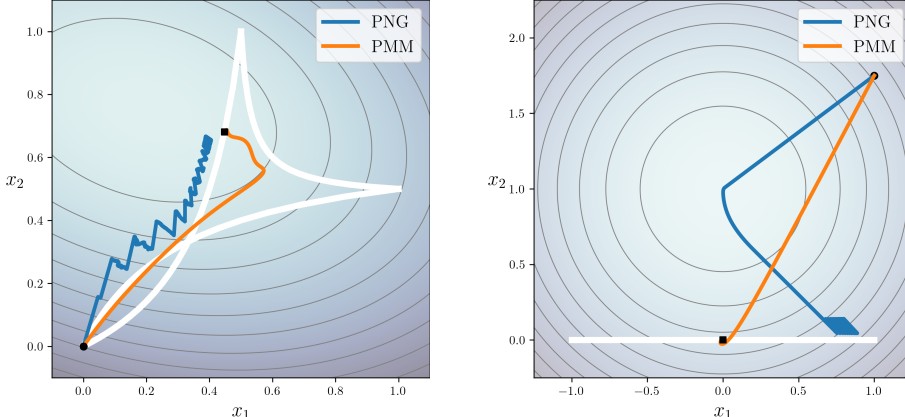

Figure 2: Visualization of learning dynamics for two Pareto-constrained optimization problems; ours Algorithm 1 (PMM) is in orange, and the existing method (PNG) introduced by Ye and Liu (2022) is in blue. In both cases, the dynamics begin at the black dot. The ground truth preference-optimal solution, found by exhaustive search, is marked by a black square. The boundary of the Pareto set of $F$ is colored white. The preference function $f_0$ is visualized by the heatmap and contour lines. All objectives and preferences are strongly-convex quadratics. See Appendix I for details.

**Remark 1.** *Algorithm 1 makes calls to sub-routines at each iteration to solve two sub-problems. As the problems are strongly-convex and Lipschitz-smooth, they can be solved using (projected) gradient descent with iteration complexity $O(\log(1/\varepsilon_0))$. And so, taking the computational cost of the sub-problems into account only increases the rate obtained in Theorem 10 by logarithmic factors.*

**Theorem 11** (Convergence of PMM with dueling feedback). *In addition to the assumptions of Theorem 10, suppose that the link function $\sigma$ satisfies Assumption D. Let the family of approximate surrogates (16) be constructed by Algorithm 2 with input parameters $(\alpha, \gamma, b, m)$ that are specified by Lemma 9. Let $(x_k, \beta_k)_k$ be the iterates of Algorithm 1 using this family of surrogates. There is a stopping time $K$ such that $\mathbb{E}[(f_0 \circ x^*)(\beta_k)]$ is monotonically decreasing for $k \in [K]$ and $(x_K, \beta_K)$ is an $(\varepsilon_0, \varepsilon)$-preference stationary point in expectation, i.e., $\|\nabla f_{\beta_K}(x_K)\|_2 \leq \varepsilon$, and conditions (a) and (b) of Lemma 7 hold in expectation for $(x_K, \beta_K)$. Moreover, $K = O\left(\varepsilon_0^{-2}\right)$.*

The complete theorem statement and its proof are given in Appendix H.2.

**Remark 2.** *The rate obtained in Theorem 10 is dimension-independent ensuring its applicability to large-scale problems. However, under dueling feedback, to achieve the rate in Theorem 11, $m$ and $b$ needs to be dimension-dependent, which is unavoidable (Wang et al., 2023; Chen et al., 2025a; Saha et al., 2021). One might expect a rate dependent on $n \ll d$ under more informative dueling feedback, e.g., preferences between $\beta$ vectors rather than $x$. Exploring such strategies is left for future work.*

## 8  Conclusion

In this work, we provide a principled and efficient way to select a decision vector from the Pareto set of a set of objectives $f_1, \ldots, f_n$ given an additional preference function $f_0$. The primary motivation is to seek the most preferred solution from a large model like LLM that is pretrained to satisfy a number of desiderata. A main contribution of this work is to provide a geometrically-meaningful notion of (approximate) preference stationarity. This is non-trivial due to the non-smoothness and non-convexity of the Pareto set. We achieve this by reformulating the constraint set as the Pareto manifold instead of the Pareto set. We also provide a simple algorithm that achieves $\varepsilon_0$-approximate stationarity with iteration complexity of $O(\varepsilon_0^{-2})$, under both first-order and dueling feedback.

There are several promising directions for future research. For example, extending this work to nonconvex $F$ is significantly more challenging. Another impactful direction is incorporating deterministic dueling feedback to enhance practical applicability. A high-dimensional setup, where not only the decision vector is high-dimensional, but the number of objectives $f_1, f_2, \cdots, f_n$ are allowed to increase with the problem scale, will also be quite interesting to explore.

## Acknowledgments and Disclosure of Funding

The research is partially supported by the NSF award CCF-2112665 (TILOS). It is also supported in part by the U.S. Department of Energy, the Office of Science, DARPA AIE FoundSci, as well as CDC-RFA-FT-23-0069 from the CDC's Center for Forecasting and Outbreak Analytics.

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

# A  Notation

| Symbol | Usage |
|---|---|
| $\Delta^{n-1}$ | the $(n-1)$-simplex equipped with the $\ell_1$-metric
$\Delta^{n-1} = \left\{ \beta \in \mathbb{R}^n : \sum_{i \in [n]} \beta_i = 1 \text{ and } \forall i \in [n], \ \beta_i \geq 0 \right\}$ |
| $\Delta^{n-1}(x)$ | the set of $\beta$ satisfying $\nabla f_\beta(x) = 0$, (17) |
| $\nabla x^*, \widehat{\nabla} x^*$ | derivative of the map $x^*$ and its approximation, (4), (10) |
| $\mathrm{err}_{\nabla f_0}(x, \beta)$ | bound on the approximation error of $\nabla(f_0 \circ x^*)$, Lemma E.6 |
| $F, (f_1, \dots, f_n)$ | the set of objective functions |
| $f_0$ | the preference function |
| $f_\beta(x)$ | the scalarized objective $\sum_i \beta_i f_i(x)$, (2) |
| $g(\beta'; x, \beta)$ | majorizing surrogate for $f(x_{\beta'})$, (12) |
| $\kappa$ | condition number $\kappa := L/\mu$ for $\nabla^2 f_i$, Assumption A |
| $L, L_H, L_0$ | Lipschitz parameters for $\nabla f_i$, $\nabla^2 f_i$, and $\nabla f_0$, Assumptions A, B, C |
| $M_0, M_1$ | Lipschitz parameters for $x^*$ and $\nabla x^*$, Lemma E.2 |
| $\mu$ | strong convexity parameter for $f_i$, see Assumption A |
| $\mu_g$ | strong convexity parameter $nL_0 M_1$ for the surrogate $g$, (12) |
| $\mathrm{Pareto}(F)$ | the set of Pareto optimal solutions of $F$, Definition 1 |
| $r$ | distance between the minimizers of $f_1, \dots, f_n$, Assumption A |
| $\mathcal{P}(F)$ | the Pareto manifold, Definition 3 |
| $R$ | $\mathrm{diam}\big(\mathrm{Pareto}(F)\big) := \sup \left\{ \|x - x'\|_2 : x, x' \in \mathrm{Pareto}(F) \right\}$, Lemma E.1 |
| $x^*(\beta), x_\beta$ | stationary point for $f_\beta$, (3) |
| $\sigma$ | link function for comparison oracle, (15) |

# B  Proofs for Section 3

**Lemma 1.** *Let $F \equiv (f_1, \ldots, f_n)$ be a collection of smooth and strictly convex objectives. Then:*

$$x \in \mathrm{Pareto}(F) \quad \Longleftrightarrow \quad (x, \beta) \in \mathcal{P}(F) \quad \text{for some } \beta \in \Delta^{n-1}.$$

*Proof.* The condition on the left states that $x$ is Pareto optimal. The one on the right states that $x$ is Pareto stationary. When the objectives are smooth and strongly convex, Pareto optimality and Pareto stationarity are equivalent (Theorem 3.1, Fliege et al. (2009)). $\square$

**Proposition 2** (Characterization of the Pareto manifold). *Let $F \equiv (f_1, \ldots, f_n)$ be a collection of smooth and strictly convex objectives. Define $x^* : \Delta^{n-1} \to \mathrm{Pareto}(F)$ by:*

$$x^*(\beta) \equiv x_\beta := \arg\min_{x \in \mathbb{R}^d} f_\beta(x). \tag{3}$$

*Let $\nabla F(x) \in \mathbb{R}^{n \times d}$ be the Jacobian. Then, the map $x^*$ has derivative:*

$$\nabla x^*(\beta) = -\nabla^2 f_\beta(x_\beta)^{-1} \nabla F(x_\beta)^\top, \tag{4}$$

*so that the map $\beta \mapsto (x_\beta, \beta)$ is a diffeomorphism of $\Delta^{n-1}$ with the Pareto manifold $\mathcal{P}(F)$.*

*Proof.* The map $x^*$ is well-defined because $f_\beta$ is strongly convex—it is the convex combination of strongly convex objectives, so it has a unique minimizer. And as the objectives are smooth, the stationarity condition $\nabla f_\beta(x) = 0$ uniquely holds at $x^*(\beta)$:

$$\nabla f_\beta(x_\beta) = 0.$$

Define the map $\zeta(x, \beta) = \nabla f_\beta(x)$. Then, the Pareto manifold is precisely the zero set $\mathcal{P}(F) = \zeta^{-1}(0)$, and which can be parametrized by simplex $\Delta^{n-1}$ via the map $\beta \mapsto (x_\beta, \beta)$.

In fact, it is a smooth parametrization. To see this, we apply the implicit function theorem (Theorem B.1), which states that the map $x^*$ is smooth at $\beta$ when $\nabla_x \zeta(x_\beta, \beta)$ is invertible. Indeed, we have that $\zeta$ is continuously differentiable, with:

$$\nabla_x \zeta(x, \beta) = \sum_{i \in [n]} \beta_i \nabla^2 f_i(x) = \nabla^2 f_\beta(x),$$

$$\nabla_\beta \zeta(x, \beta) = \nabla_\beta \left( \sum_{i \in [n]} \beta_i \nabla f_i(x) \right) = \nabla F(x)^\top.$$

As $f_\beta$ is strictly convex, it has positive definite Hessian, so that $\det \nabla_x \zeta(x_\beta, \beta) \neq 0$. Furthermore, Theorem B.1 also implies that the derivative of $\nabla x^*$ is given by Equation (4). It follows that the map $\beta \mapsto (x_\beta, \beta)$ is smooth. It also has a smooth inverse. Namely, the projection onto the second component $(x_\beta, \beta) \mapsto \beta$. Thus, $\mathcal{P}(F)$ is diffeomorphic with $\Delta^{n-1}$. $\square$

**Theorem B.1** (Implicit function theorem, Spivak (2018)). *Let $f : \mathbb{R}^d \times \mathbb{R}^n \to \mathbb{R}^d$ be continuously differentiable on an open set containing $(a, b)$ and let $f(a, b) = 0$. Let $\nabla_u f(u, v)$ be the $d \times d$ matrix:*

$$\left[ \nabla_u f(u, v) \right]_{ij} = \nabla_{u_j} f_i(u, v).$$

*If $\det \nabla_u f(a, b) \neq 0$, there are open sets $U \subset \mathbb{R}^d$ and $V \subset \mathbb{R}^n$ containing $a$ and $b$ respectively with the following property: for each $v \in V$ there is a unique $g(v) \in U$ such that $f(g(v), v) = 0$. Furthermore, the map $g$ is differentiable with derivative given by:*

$$\nabla g(v) = -\left[ \nabla_u f(g(v), v) \right]^{-1} \nabla_v f(g(v), v).$$

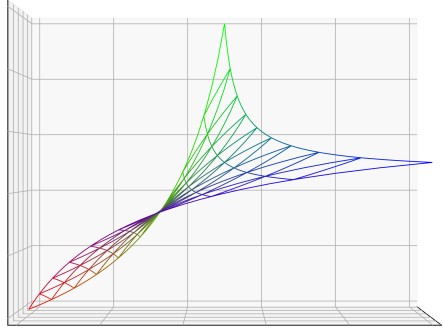

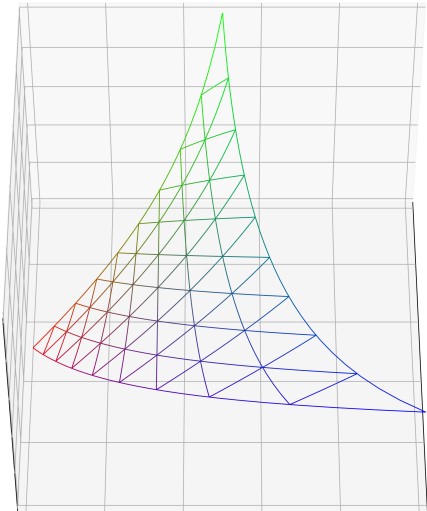

Figure 3: An embedding of the Pareto manifold from (1) into three dimensions from two camera angles. The singularity seen in (1) is an artifact of the projection of the manifold into the decision space, rather than an intrinsic irregularity in the manifold.

# C   A comparison of solution concepts

In this section, we elaborate on the relationship between different solution concepts (optimality, stationarity, approximate stationarity) for the Pareto-constrained optimization problem:

$$\underset{\beta \in \Delta^{n-1}}{\text{minimize }} (f_0 \circ x^*)(\beta). \tag{6}$$

The solution concepts are related by:

$$\begin{array}{ccccccc} \text{preference} & & \text{preference} & & \text{weak preference} & & \text{approximate preference} \\ \text{optimality} & \subset & \text{stationarity} & \subset & \text{stationarity} & \subset & \text{stationarity} \end{array}$$

It is fairly clear that the first and last inequalities are strict. We discuss the inner inequality.

It turns out that a point $x$ can be weakly preference stationary without being preference stationary. However, this can only happen if $x$ is also a point of singularity in $\text{Pareto}(F)$. Geometrically, if we consider $\text{Pareto}(F)$ as the projection of $\mathcal{P}(F)$ onto its first component in $\mathbb{R}^d$, the this means that multiple points are collapsed onto $x$. Algebraically, this means that the set of gradients $\nabla f_1(x), \ldots, \nabla f_n(x)$ fails to have full (Pareto) rank (Smale, 1973; Hamada et al., 2020).

To elaborate, define the set:

$$\Delta^{n-1}(x) := \left\{ \beta \in \Delta^{n-1} : \nabla f_\beta(x) = 0 \right\}. \tag{17}$$

Then, $x$ is Pareto stationary if there is some $\beta$ in $\Delta^{n-1}(x)$, so that:

$$\sum_{i \in [n]} \beta_i \nabla f_i(x) = 0,$$

and the rank of this set of gradients is at most $n-1$. Since $\Delta^{n-1}$ does not contain any collinear vectors, if $\Delta^{n-1}(x)$ contains more than a single point, then the rank of the set of gradients must be strictly less than $n-1$. This leads us to the definition:

**Definition C.1** (Pareto genericity). *Let* $\{v_1, \ldots, v_n\} \subset \mathbb{R}^d$. *This set is* Pareto generic *if:*

$$\beta_1 v_1 + \cdots + \beta_n v_n = 0, \qquad \text{for some } \beta \in \Delta^{n-1},$$

*and the non-degeneracy condition holds:* $\text{rank}(v_1, \ldots, v_n) = n - 1$.

If $\nabla F(x)$ is Pareto generic, then $\Delta^{n-1}(x)$ contains a unique $\beta$, so we immediately have:

**Proposition C.1** (Weak and strong preference stationarity). *When the set of gradients $\nabla F(x)$ is Pareto generic and $x$ is weakly preference stationary, then $x$ is preference stationary.*

However, when the gradients $\nabla F(x)$ are not Pareto generic, then weak preference stationarity can be strictly weaker. Let $(x, \beta)$ where $\beta \in \Delta^{n-1}(x)$ be weakly preference stationary, so that:

$$-\nabla f_0(x)^\top \underbrace{\left( -\nabla^2 f_\beta(x)^{-1} \nabla F(x)^\top (\beta' - \beta) \right)}_{\nabla x^*(\beta)(\beta' - \beta)} \le 0, \qquad \forall \beta' \in \Delta^{n-1}.$$

We can simplify this by using the fact that $\nabla f_\beta(x) = \nabla F(x)^\top \beta = 0$. Then, one way for the stationary condition to be fulfilled is for the underlined term to be normal to $\nabla f_0(x)$:

$$-\nabla^2 f_\beta(x)^{-1} \nabla F(x)^\top \beta' \in \text{span}\big(\nabla f_0(x)\big)^\perp, \qquad \forall \beta' \in \Delta^{n-1}.$$

This statement has the following geometric interpretation. These vectors are contained in the Clarke tangent cone of $\text{Pareto}(F)$ at $x$. If these are the only vectors in the tangent cone, then this condition states that $-\nabla f_0(x)$ is contained in the normal cone of $\text{Pareto}(F)$ at $x$.

But, in general, the tangent cone contains the union of subspaces:

$$\bigcup_{\beta \in \Delta^{n-1}(x)} \left\{ -\nabla^2 f_\beta(x)^{-1} \nabla F(x)^\top \beta' : \beta' \in \Delta^{n-1} \right\}.$$

And so, when $\Delta^{n-1}(x)$ does not contain a unique vector, the tangent cone can contain more vectors. By selecting different $\beta$'s, we recover different slices of the tangent cone. This also means that even if the above normality condition holds for one $\beta$, it may fail to hold for another $\tilde{\beta} \in \Delta^{n-1}(x)$. Then, $(x, \beta)$ is weakly preference stationary while $(x, \tilde{\beta})$ may not be.

# D   Insufficiency of first-order information

The notion of preference stationarity uses second-order information in $F$, for the term $\nabla^2 f_\beta(x)^{-1}$. A natural question is whether there are reasonable notions of stationarity that only use first-order information. It turns out there are none, if we require the stationarity condition to be (i) non-trivial, (ii) necessary for preference optimality and (iii) decidable from local information at a single point $x$.

The reason is that from $\nabla F(x)$ alone, we can only determine whether the point $x$ is contained in the Pareto manifold. It is not enough to determine locally how the manifold curves. For example, (4) shows two different Pareto sets that share the same gradients at a point $x$. The local optimality of $x$ with respect to $f_0$ depends on its neighboring Pareto points. It turns out that to attain a non-trivial and necessary condition, we would either need to look at higher-order information, or first-order information at more than a one point.

To formalize this, we define stationarity conditions as *decision functions*. These are functions that return Boolean outcomes `true` or `false` (we interpret these functions as ways of classifying points as stationary or not). In particular, we consider first-order stationary conditions, which makes decisions given only the gradients $\nabla f_0$ and $\nabla f_1, \ldots, \nabla f_n$ at a single point:

**Definition D.1** (Stationarity function). *A first-order stationary condition is a decision function whose input is a tuple of $n + 1$ vectors in $\mathbb{R}^d$:*

$$\text{Stationary} : \mathbb{R}^d \times \overbrace{\cdots}^{n+1 \text{ times}} \times \mathbb{R}^d \to \{\text{true}, \text{false}\}.$$

*Let $f_0$ be a smooth preference function and $f_1, \ldots, f_n$ be smooth, strongly convex objectives. We say that a first-order condition is* necessary *if the following holds:*

$$x \text{ is preference optimal} \quad \implies \quad \text{Stationary}\big(\nabla f_0(x), \ldots, \nabla f_n(x)\big) = \text{true}.$$

There are two specific cases in which it is possible to determine that a decision $x$ is *not* preference optimal from first-order information. First, if $x$ is not even Pareto optimal, then it cannot be preference optimal. This occurs whenever $\nabla f_\beta(x) \neq 0$ for all $\beta \in \Delta^{n-1}$. The second case occurs if $x$ is Pareto optimal, but $\nabla f_0(x)$ is in the convex cone spanned by the columns of $\nabla F(x)$,

$$\nabla f_0(x) = \sum_{i \in [n]} \lambda_i \nabla f_i(x), \qquad \text{where } \lambda_i \geq 0.$$

In particular, let $x = x_\beta$. Then, let $\gamma : [0,1] \to \Delta^{n-1}$ be the curve parametrized by $\gamma(t) = t\lambda + (1-t)\beta$. Then whenever $\nabla f_0(x_\beta) \neq 0$:

$$
\begin{aligned}
\frac{d}{dt} f_0\big(x^*(\gamma(t))\big)\Big|_{t=0} &= -\nabla f_0(x_\beta) \nabla^2 f_\beta(x_\beta)^{-1} \nabla F(x_\beta)^\top (\lambda - \beta) \\
&= -\nabla f_0(x_\beta) \nabla^2 f_\beta(x_\beta)^{-1} \nabla f_0(x_\beta) < 0.
\end{aligned}
$$

Intuitively, these solutions are not preference optimal because we can find more preferable Pareto solutions by weighting those objectives that align with $f_0$ more. To exclude these specific cases, we introduce the notion of *preference genericity*. Then, Proposition D.1 shows that any necessary, first-order stationary condition must accept all preference generic sets.

**Definition D.2** (Preference genericity). *Let $\{v_0, v_1, \ldots, v_n\} \subset \mathbb{R}^d$ where $1 < n \leq d$. This set is* preference generic *if there exists some $\beta \in \Delta^{n-1}$ such that $\beta_1 v_1 + \cdots \beta_n v_n = 0$, and:*

$$v_0 \notin \text{span}(v_1, \ldots, v_n).$$

**Proposition D.1** (Necessary first-order conditions are trivial). *If* Stationary *is a necessary, first-order stationary condition, then it is trivial in the following sense:*

$$\text{Stationary}(v_0, \ldots, v_n) = \text{true},$$

*for any preference generic set of $v_0, \ldots, v_n \in \mathbb{R}^d$.*

*Proof.* It suffices to show that there exist $f_0$, $F$, and $x^\star$ such that $x^\star$ is preference optimal and for $i = 0, \ldots, n$:

$$v_i = \nabla f_i(x^\star). \tag{18}$$

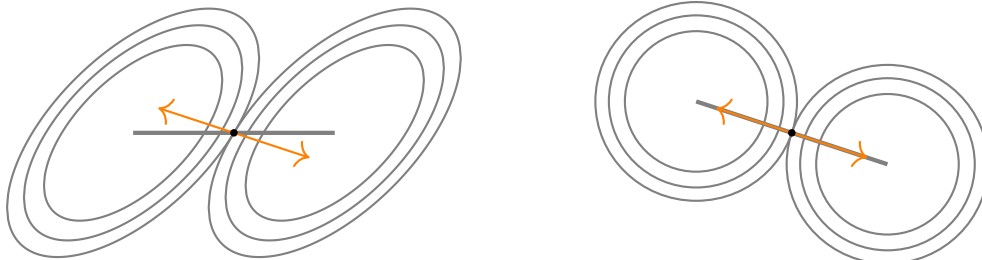

Figure 4: Two instances of $\mathrm{Pareto}(f_1, f_2)$ are shown (thick gray lines), where $f_1$ and $f_2$ are positive-definite quadratic objectives in $\mathbb{R}^2$ (visualized by contour lines). At $x$ (the black dot), the two instances share the same local information $-\nabla f_1(x)$ and $-\nabla f_2(x)$ (orange arrows); they cross the contour lines at right angles. When $n = 2$, the Pareto set contains all $z$ such that $\nabla f_1(z) = -\lambda \nabla f_2(z)$ for $\lambda \geq 0$. Notice that if $f_0$ is strictly convex and $x$ does not minimize $f_0$ over $\mathbb{R}^2$, then $x$ cannot be stationary for both instances.

And since $x^\star$ is preference optimal, any necessary stationary condition must accept:

$$\mathrm{Stationary}(v_0, \ldots, v_n) = \mathsf{true}.$$

Without loss of generality, let $x^\star = 0$ by an affine transformation. To construct $f_0$ and $F$, we can simply consider a family of positive-definite quadratics:

- Let the preference function $f_0$ be:

$$f_0(x) = \frac{1}{2}\|x + v_0\|^2.$$

  Notice that $\nabla f_0(x^*) = v_0$.

- Let the objectives $f_1, \ldots, f_n$ share the same Hessian:

$$f_i(x) = \frac{1}{2}\|A(x - z_i)\|^2,$$

  where $A \in \mathbb{R}^{d \times d}$ is full rank and $z_i \in \mathbb{R}^d$. Let $H = A^\top A$ for short.

We show that we can set $A$ and the $z_i$'s so that $x^\star$ is preference optimal while (18) holds.

By Lemma D.1, the Pareto set is the convex hull $\mathcal{C} := \mathrm{conv}(z_1, \ldots, z_n)$. Notice that the choice of $H$ and $v_i$'s determines the $z_i$'s, since we require $\nabla f_i(x^*) = v_i$, which expands to:

$$z_i = -H^{-1}v_i, \qquad \forall i \in [n].$$

From convex optimization, $x^\star = 0$ is preference optimal if (i) $x^\star \in \mathcal{C}$ and (ii) $\mathcal{C}$ is normal to $\nabla f_0$. Indeed, these two conditions can be fulfilled:

(i) Because $v_1, \ldots, v_n$ is assumed to be Pareto generic, zero is a convex combination of the $v_i$'s. As the $z_i$'s are related to the $v_i$'s by a linear transformation, this also implies that zero is a convex combination of the $z_i$'s (with the same set of convex weights).

(ii) We need to show that the subspace $\mathrm{span}(v_1, \ldots, v_n)$ can be mapped into $\mathrm{span}(v_0)^\perp$ by the map $v \mapsto -H^{-1}v$ where $H$ is positive definite. Lemma D.2 shows that such a map $H$ exists as long as $v_0 \notin \mathrm{span}(v_1, \ldots, v_n)$, which is assumed from preference genericity.

Thus, there exists $f_0$ and $F$ that is preference optimal at $x^\star$ with matching first-order information. A necessary stationary condition must therefore be accepted. $\qquad\square$

**Remark D.1.** *Suppose that* Stationary *is not necessary, but that we can design some optimization method that provably converges to a stationary point in* $\{x : \mathrm{Stationary}(x) = \mathsf{true}\}$. *Then, there are settings in which the method provably avoids preference optimal points.*

In the remainder of this section, we prove Lemma D.1 and Lemma D.2 used above.

**Lemma D.1.** *Let $f_1, \ldots, f_n : \mathbb{R}^d \to \mathbb{R}$ be positive-definite quadratics with a shared Hessian:*

$$f_i(x) = \frac{1}{2} \|A(x - z_i)\|^2,$$

*where $A \in \mathbb{R}^{d \times d}$ is full rank and $z_i \in \mathbb{R}^d$. Then, the Pareto set is the convex hull:*

$$\mathrm{Pareto}(f_1, \ldots, f_n) = \mathrm{conv}(z_1, \ldots, z_n).$$

*Proof.* As the objectives $f_1, \ldots, f_n$ are strongly convex, optimality is equivalent to stationarity. Thus, $x \in \mathrm{Pareto}(f_1, \ldots, f_n)$ if and only if there exists some $\beta \in \Delta^{n-1}$ such that:

$$0 = \sum_{i \in [n]} \beta_i \nabla f_i(x),$$

which, when expanded, states that:

$$(A^\top A) x = (A^\top A) \sum_{i \in [n]} \beta_i z_i.$$

But as $A$ is invertible, this is equivalent to:

$$x = \sum_{i \in [n]} \beta_i z_i,$$

which is to say that $x \in \mathrm{conv}(z_1, \ldots, z_n)$. $\qquad \square$

**Lemma D.2.** *Let $U$ and $V$ be linear subspaces of $\mathbb{R}^d$ such that $U \cap V^\perp = \{0\}$. Then, there exists some positive definite map $H : \mathbb{R}^d \to \mathbb{R}^d$ such that $H(U) \subset V$.*

*Proof.* If $S \subset \mathbb{R}^d$ is a subspace, let $\Pi_S : \mathbb{R}^d \to \mathbb{R}^d$ be the projection onto $S$. Define the map:

$$H := \Pi_V + \Pi_{V^\perp} \Pi_{U^\perp}.$$

Then $H$ satisfies the following:

- $H$ is positive definite. To see this, let $0 \neq x \in \mathbb{R}^d$ have decomposition $x = x_1 + x_2$, where $x_1 \in U$ and $x_2 \in U^\perp$. Then:

$$x^\top H x = \underbrace{x_1^\top \Pi_V x_1 + 2x_1 \Pi_V x_2 + x_2^\top \Pi_V x_2}_{x^\top \Pi_V x} + \underbrace{x_1^\top \Pi_{V^\perp} x_2 + x_2^\top \Pi_{V^\perp} x_2}_{x^\top \Pi_{V^\perp} \Pi_{U^\perp} x}$$

$$= \|\Pi_V x_1\|^2 + \underbrace{x_1^\top x_2}_{0} + x_1 \Pi_V x_2 + \|x_2\|^2 \geq \frac{1}{2} \|\Pi_V x_1 + x_2\|^2 > 0,$$

  where the last inequality is strict because $x \neq 0$ and $U \cap V^\perp = \{0\}$.

- $H(U) \subset V$. If $x \in U$, then by definition $\Pi_{U^\perp} x = 0$ so that $Hx = \Pi_V x \in V$.

$\qquad \square$

## D.1  An example of a first-order stationarity condition avoiding optimality

In this section, we discuss the first-order stationarity condition of Ye and Liu (2022), defined to as stationarity with respect to their optimization dynamics, *Pareto navigating gradient descent* (PNG). We show that it fails to be a necessary condition for preference optimality.

Despite that, their condition and dynamics have appealing properties since (i) they do not require second-order information, which is computationally more expensive, and (ii) their dynamics largely satisfies what they call the *Pareto improvement property*, which ensures that each objective enjoys monotonic improvement during optimization:

$$\frac{d}{dt} f_i(x_t) \leq 0, \qquad \text{for all } i \in [n].$$

As Pareto improvement can be at odds with preference optimality, this leads to an open question: when and how should we balance Pareto improvement with preference optimality?

**Definition D.3** (PNG stationarity, Ye and Liu (2022))**.** *Let $c > 0$. The* PNG *vector $v_c(x)$ is defined:*

$$v_c(x) := \arg\min_{v \in \mathbb{R}^d} \frac{1}{2}\|\nabla f_0(x) - v\|^2$$

$$\text{s.t. } \nabla f_i(x)^\top v \geq c, \qquad \text{for all } i \in [n].$$

*Let $\varepsilon > 0$. A vector $x \in \mathbb{R}^d$ is $(c, \varepsilon)$-PNG stationary if $v_c(x) = \lambda \nabla f_0(x)$ for some $\lambda \leq 0$ and:*

$$\min_{\beta \in \Delta^{n-1}} \|\nabla f_\beta(x)\| = \varepsilon.$$

In the following, we consider an example in $\mathbb{R}^2$ with $n = 2$ objectives. Let the standard basis be denoted $\mathbf{e}_1, \mathbf{e}_2 \in \mathbb{R}^2$, and let the objective functions $f_1, f_2 : \mathbb{R}^2 \to \mathbb{R}$ be defined:

$$f_1(x) = \frac{1}{2}\|A(x + \mathbf{e}_1)\|^2 \qquad \text{and} \qquad f_2(x) = \frac{1}{2}\|A(x - \mathbf{e}_1)\|^2, \tag{19}$$

where $A \in \mathbb{R}^{2\times 2}$ is full-rank. Lemma D.1 shows that the Pareto set of the objectives $\text{Pareto}(f_1, f_2)$ is the line segment from $-\mathbf{e}_1$ to $\mathbf{e}_2$. Thus, the Pareto set is invariant under change in $A$, while PNG stationarity is not, as the constraint set changes with $A$:

$$\{v : (x + \mathbf{e}_1)^\top H v \geq c\} \cap \{v : (x - \mathbf{e}_1)^\top H v \geq c\},$$

and $H = A^\top A$. Due to this discrepancy, PNG stationary points can be preference suboptimal.

**Example D.1.** *Let the preference function be: $f_0(x) = \frac{1}{2}\|x - \mathbf{e}_2\|^2$, and let the objectives $f_1, f_2$ be defined as in the above (19) with:*

$$H = A^\top A = \begin{bmatrix} 1 & 1 \\ 1 & 2 \end{bmatrix}. \tag{20}$$

*Then, the unique preference optimal point is the origin 0. However, the $(c, \varepsilon)$-PNG stationary point is bounded away from 0. It even converges to $\mathbf{e}_1$ as the error tolerance $\varepsilon$ goes to zero.*

*Proof.* Consider the PNG vector $v_c(x)$ when $x$ is in the region:

$$\mathcal{C} = \{x \in \mathbb{R}^2 : \nabla f_0(x)^\top \nabla f_i(x) < 0, \text{ for } i = 1, 2\} \cap \{\mathbf{e}_2^\top x > 0\}.$$

Here, both constraints $\nabla f_i(x)^\top v \geq c$ are active in the constrained optimization problem that defines the PNG vector; and so, $v_c(x)$ is the vertex point of the constraint set, satisfying:

$$\nabla f_1(x)^\top v_c(x) = \nabla f_2(x)^\top v_c(x) = c.$$

Expanding out the gradients, we obtain:

$$(x + \mathbf{e}_1)^\top H v_c(x) = c \qquad \text{and} \qquad (x - \mathbf{e}_1)^\top H v_c(x) = c.$$

This implies that $\mathbf{e}_1^\top H v_c(x) = 0$. Now suppose that $x_{\text{PNG}} \in \mathcal{C}$ is PNG stationary. Then, by definition, it must satisfy $\nabla f_0(x_{\text{PNG}}) \in \text{span}(v_c(x_{\text{PNG}}))$, so it has the form:

$$x_{\text{PNG}} = \mathbf{e}_2 + \lambda u, \qquad \text{where } \mathbf{e}_1^\top H u = 0.$$

Whenever the standard basis vectors are not eigenvectors of $H$, the line $\mathbf{e}_2 + \lambda u$ intersects $\text{Pareto}(f_1, f_2)$ away from 0. For example, let $A$ satisfy (20).

Then, the line $\mathbf{e}_2 + \lambda u$ runs through $\mathbf{e}_1$ and $\mathbf{e}_2$. We can verify that $\mathcal{C}$ contains all points on this line between its two endpoints. When $x = \mathbf{e}_2 + \lambda(\mathbf{e}_1 - \mathbf{e}_2)$ and $\lambda \in (0, 1)$, we have:

$$\nabla f_0(x)^\top \nabla f_1(x) = (x - \mathbf{e}_2)^\top H(x + \mathbf{e}_1)$$
$$= \lambda(\mathbf{e}_1 - \mathbf{e}_2)^\top H((1 + \lambda)\mathbf{e}_1 + (1 - \lambda)\mathbf{e}_2) = -\lambda(1 - \lambda),$$

and similarly, we have:

$$\nabla f_0(x)^\top \nabla f_2(x) = (x - \mathbf{e}_2)^\top H(x - \mathbf{e}_1)$$
$$= \lambda(\mathbf{e}_1 - \mathbf{e}_2)^\top H((\lambda - 1)(\mathbf{e}_1 - \mathbf{e}_2)) = -\lambda(1 - \lambda).$$

This implies that for all $c > 0$ and $\varepsilon > 0$, the $(c, \varepsilon)$-PNG stationary point is bounded away from 0, converging to $\mathbf{e}_1$ as $\varepsilon$ goes to zero. $\qquad\square$

This example is visualized in Figure 2.

# E   Proofs for Section 5

In this section, we derive the following implications from the assumptions from Section 5:

- Assumption A allows us to bound the size of the Pareto set (Lemma E.1).
- Assumption B additionally bounds the curvature of the Pareto manifold: we show that $\nabla x^*$ is well-behaved (Lemma E.2) and is well-approximated by $\widehat{\nabla} x^*$ (Lemma E.5).
- Assumption C further leads to error bounds when approximating gradient of $f_0 \circ x^*$ (Lemma E.6). It also allows us to define a notion of approximate preference stationarity that is geometrically meaningful (Proposition 6) and can be verified using approximate information (Lemma 7).

Let's recall from Assumption A that the condition number of the objectives $\nabla^2 f_i$ for $i \in [n]$ is upper bounded by $\kappa := L/\mu$. We also defined the scale parameter $r$ as the maximum distance between any of the minimizers of the objectives:

$$r := \max_{i,j \in [n]} \left\| \arg\min f_i(x) - \arg\min f_j(x) \right\|_2.$$

**Lemma E.1** (Size of Pareto set). *Let $F$ satisfy Assumption A. Then $R \leq \sqrt{\kappa} r$, where:*

$$R := \operatorname{diam}\big(\operatorname{Pareto}(F)\big) \equiv \sup \big\{ \|x - x'\|_2 : x, x' \in \operatorname{Pareto}(F) \big\}.$$

*Proof.* Because each $f_i$ is $\mu$-strongly convex and $L$-Lipschitz smooth, so is the convex combination $f_\beta$. This implies the upper and lower bounds:

$$\frac{1}{2}\mu \sum_{i \in [n]} \beta_i \|x - x_i\|_2^2 \leq f_\beta(x) \leq \frac{1}{2}L \sum_{i \in [n]} \beta_i \|x - x_i\|_2^2.$$

It follows that the minimizer of $f_\beta$ is bounded:

$$f_\beta(x_\beta) \leq \frac{1}{2}Lr^2.$$

On the other hand, if a point $\|x - x_i\| > 2s$ for some $i \in [n]$, then by reverse triangle inequality, $\|x - x_j\| > s$ for all $j \in [n]$. This implies that:

$$\|x - x_i\| > 2s \qquad \Longrightarrow \qquad f_\beta(x) > \frac{1}{2}\mu s^2.$$

It follows that if $\|x - x_i\| > 2\sqrt{L/\mu}$ for some $i$, then $x$ is not a Pareto optimal point. $\qquad\square$

**Lemma E.2** (Smoothness of $x^*$). *Let $F$ satisfy Assumptions A,B. Then, $x^* : \Delta^{n-1} \to \mathbb{R}^d$ is $M_0$-Lipschitz continuous and has $M_1$-Lipschitz continuous gradients, where:*

$$M_0 := \kappa R \qquad and \qquad M_1 := 2\kappa^2 R \left(1 + \frac{L_H R}{\mu}\right).$$

**Proof of Lemma E.2**   That $x^*$ is Lipschitz continuous with Lipschitz continuous gradients follows from the following two lemmas:

**Lemma E.3.** *Let $F \equiv (f_1, \ldots, f_n)$ be a set of twice-differentiable objectives and let $f_0$ be a smooth preference. Suppose the objectives are $L$-Lipschitz smooth and $\mu$-strongly convex:*

$$\mu \mathbf{I} \preceq \nabla^2 f_i \preceq L \mathbf{I}.$$

*Let $R := \operatorname{diam}\big(\operatorname{Pareto}(F)\big)$. Then, the map $x^* : (\Delta^{n-1}, \ell_1) \to (\mathbb{R}^d, \ell_2)$ is $LR/\mu$-Lipschitz.*

*Proof.* Recall from (4) that $\nabla x^*(\beta) = -\nabla^2 f_\beta(x_\beta)^{-1} \nabla F(x_\beta)^\top$. Then:

$$\|\nabla x^*(\beta)\|_{1,2} \overset{(i)}{\leq} \left\|\nabla^2 f_\beta(x_\beta)^{-1}\right\|_2 \cdot \left\|\nabla F(x_\beta)^\top\right\|_{1,2}$$

$$\overset{(ii)}{\leq} \frac{1}{\mu} \cdot LR,$$

where (i) is a property of the $\|\cdot\|_{1,2}$-norm, (ii) uses $\mu \mathbf{I} \preceq \nabla^2 f_\beta(x_\beta)$ and Lemma G.1. $\qquad\square$

**Lemma E.4.** *Let $\beta, \beta' \in \Delta^{n-1}$. Then,*

$$\left\|\nabla x^*(\beta) - \nabla x^*(\beta')\right\|_{1,2} \leq \frac{2L^2 R}{\mu^2}\left(1 + \frac{L_H R}{\mu}\right) \cdot \|\beta - \beta'\|_1.$$

*Proof.* By definition, we have:

$$\left\|\nabla x^*(\beta) - \nabla x^*(\beta')\right\|_{1,2} = \left\| - \nabla^2 f_\beta(x_\beta)^{-1} \nabla F(x_\beta)^\top + \nabla^2 f_{\beta'}(x_{\beta'})^{-1} \nabla F(x_{\beta'})^\top \right\|_{1,2}.$$

We can add and subtract $\nabla^2 f_\beta(x_\beta)^{-1} \nabla F(x_{\beta'})^\top$ inside the right-hand side (RHS):

$$\underbrace{\left\| - \nabla^2 f_\beta(x_\beta)^{-1} \cdot \left[\nabla F(x_\beta) - \nabla F(x_{\beta'})\right]^\top + \left[\nabla^2 f_\beta(x_\beta)^{-1} - \nabla^2 f_{\beta'}(x_{\beta'})^{-1}\right] \cdot \nabla F(x_{\beta'})^\top \right\|_{1,2}}_{\text{(RHS)}}.$$

We can bound the two terms in the norm separately. For the first:

$$\left\| - \nabla^2 f_\beta(x_\beta)^{-1} \cdot \left[\nabla F(x_\beta) - \nabla F(x_{\beta'})\right]^\top \right\|_{1,2} \overset{(i)}{\leq} \frac{L}{\mu} \cdot \|x_\beta - x_{\beta'}\| \overset{(ii)}{\leq} \frac{L^2 R}{\mu^2}\|\beta - \beta'\|_1,$$

where (i) follows the same argument as Lemma E.5, and (ii) applies Lemma E.3. For the second term, we can add and subtract $\nabla^2 f_{\beta'}(x_\beta)^{-1}\nabla F(x_{\beta'})^\top$ to obtain:

$$\left\|\left[\nabla^2 f_\beta(x_\beta)^{-1} - \nabla^2 f_{\beta'}(x_{\beta'})^{-1}\right] \cdot \nabla F(x_{\beta'})^\top\right\|_{1,2}$$
$$= \left\|\left[\nabla^2 f_\beta(x_\beta)^{-1} - \nabla^2 f_{\beta'}(x_\beta)^{-1} + \nabla^2 f_{\beta'}(x_\beta)^{-1} - \nabla^2 f_{\beta'}(x_{\beta'})^{-1}\right] \cdot \nabla F(x_{\beta'})^\top\right\|_{1,2}$$
$$\leq \left(\frac{L}{\mu^2}\|\beta - \beta'\|_1 + \frac{L_H}{\mu^2}\frac{LR}{\mu}\|\beta - \beta'\|_1\right) \cdot LR.$$

where $\nabla^2 f_\beta(x)^{-1} - \nabla^2 f_{\beta'}(x)^{-1}$ is bounded by Lemma G.3; $\nabla^2 f_\beta(x)^{-1} - \nabla^2 f_\beta(x')^{-1}$ is bounded by Lemma G.2 and Lemma E.3; $\|\nabla F(x_{\beta'})^\top\|_{1,2}$ is bounded by Lemma G.1. $\square$

The result follows by substituting in the definitions of $M_0$ and $M_1$. $\blacksquare$

**Lemma E.5** (Approximability of $\nabla x^*$). *If $F$ satisfies Assumptions A,B. Then:*

$$\|\nabla x^*(\beta) - \widehat{\nabla} x^*(x, \beta)\|_{1,2} \leq \frac{1}{\mu}\frac{M_1}{2M_0}\|\nabla f_\beta(x)\|_2.$$

*Proof.* Recall that $x_\beta := x^*(\beta)$. Then, by definition, we have:

$$\left\|\widehat{\nabla} x^*(x, \beta) - \nabla x^*(\beta)\right\|_{1,2} = \left\| - \nabla^2 f_\beta(x)^{-1}\nabla F(x)^\top + \nabla^2 f_\beta(x_\beta)^{-1}\nabla F(x_\beta)^\top \right\|_{1,2}.$$

We can add and subtract $\nabla^2 f_\beta(x)^{-1}\nabla F(x_\beta)^\top$ inside the right-hand side (RHS) to get:

$$(\text{RHS}) = \left\| - \nabla^2 f_\beta(x)^{-1} \cdot \left[\nabla F(x) - \nabla F(x_\beta)\right]^\top + \left[\nabla^2 f_\beta(x)^{-1} - \nabla^2 f_\beta(x_\beta)^{-1}\right] \cdot \nabla F(x_\beta)^\top \right\|_{1,2}$$

$$\overset{(i)}{\leq} \frac{L}{\mu} \cdot \|x - x_\beta\| + \frac{L_H}{\mu^2}\|x - x_\beta\| \cdot LR$$

$$\overset{(ii)}{\leq} \frac{L}{\mu^2}\left(1 + \frac{L_H R}{\mu}\right) \cdot \|\nabla f_\beta(x)\|,$$

where (i) the first blue term uses $\mu\mathbf{I} \preceq \nabla^2 f_\beta$ and the $L$-Lipschitz smoothness of the objectives, while the bracket orange term follows from Lemma G.2 and the final purple term follows from Lemma G.1, and (ii) uses the $\mu$-strong convexity of $f_\beta$. $\square$

**Lemma E.6** (Approximability of $\nabla(f_0 \circ x^*)$). *If $F, f_0$ satisfy Assumptions A,B,C. Then:*

$$\left\|\nabla(f_0 \circ x^*)(\beta)^\top - \nabla f_0(x)^\top \widehat{\nabla} x^*(x, \beta)\right\|_{1,2} \leq \text{err}_{\nabla f_0}(x, \beta),$$

*where* $\text{err}_{\nabla f_0}(x, \beta) := \frac{1}{\mu}\left(\frac{M_1}{2M_0}\|\nabla f_0(x)\|_2 + L_0 M_0\right)\|\nabla f_\beta(x)\|_2$.

*Proof.* Add and subtract $\nabla f_0(x)^\top \nabla x^*(\beta)$ within the norm on the right-hand side:

$$(\text{RHS}) = \left\| \left(\nabla f_0(x_\beta)^\top - \nabla f_0(x)\right)^\top \nabla x^*(\beta) + \nabla f_0(x)^\top \left(\nabla x^*(\beta) - \widehat{\nabla} x^*(x, \beta)\right) \right\|_{1,2}$$

$$\leq L_0 M_0 \|x_\beta - x\| + \|\nabla f_0(x)\| \cdot \frac{1}{\mu} \frac{M_1}{2M_0} \|\nabla f_\beta(x)\|_2,$$

where we use the fact that $f_0$ is $L_0$-Lipschitz smooth by Assumption C, that $x^*$ is $M_0$-Lipschitz continuous by Lemma E.2, and that $\|\nabla x^*(\beta) - \widehat{\nabla} x^*(\beta)\|_{1,2}$ is bounded by Lemma E.5. The result follows from upper bounding $\|x_\beta - x\|$ by $\mu$-strong convexity of $f_\beta$:

$$\|x_\beta - x\| \leq \frac{1}{\mu} \|\nabla f_\beta(x)\|.$$

$\square$

**Proposition 6** (Geometric meaning of approximate stationarity). *Let $(\hat{x}, \hat{\beta})$ be $(\varepsilon_0, \varepsilon)$-preference stationary. With Assumptions A,B,C, and $R := \mathrm{diam}\big(\mathrm{Pareto}(F)\big)$ and $s := 2\mu^2\varepsilon_0/(L_0 L^2 R^2)$, then:*

(a) *if $\|\beta - \hat{\beta}\|_1 \leq s$, then $f_0(x_\beta) - f_0(x_{\hat{\beta}}) \geq -2\varepsilon_0\|\beta - \hat{\beta}\|_1$, and*

(b) *$\|\hat{x} - x_{\hat{\beta}}\|_2 \leq \varepsilon/\mu$.*

*Proof.* (a) Recall that $x_\beta$ is the minimizer of $f_\beta$, by definition. Because $f_\beta$ is $\mu$-strongly convex, we can bound the distance between $x$ and $x_\beta$ by:

$$\|x - x_\beta\| \leq \frac{1}{\mu}\|\nabla f_\beta(x)\| \leq \frac{\varepsilon}{\mu},$$

where the second inequality follows from condition (8a).

(b) Let $\beta_s := (1-s)\beta + s\beta'$ parametrize the line connecting $\beta$ and $\beta'$. Let $\gamma : [0,1] \to \mathrm{Pareto}(F)$ be the path $\gamma(s) := x^*(\beta_s)$, so that:

$$d\gamma(s) = \nabla x^*(\beta_s)(\beta' - \beta)\,ds.$$

We can now upper bound the difference:

$$f_0(x_\beta) - f_0(x_{\beta'}) = -\int_\gamma \nabla f_0(x_{\beta_s})^\top d\gamma(s)$$

$$= -\int_\gamma \left[\nabla f_0(x_{\beta_s}) - \nabla f_0(x_\beta) + \nabla f_0(x_\beta)\right]^\top d\gamma(s)$$

$$\leq \int_\gamma L_0 \|x_{\beta_s} - x_\beta\|\,|d\gamma(s)| + \int_\gamma \left(-\nabla f_0(x_\beta)^\top d\gamma(s)\right).$$

Let's bound the integrals separately. Since $x_{\beta_s} = \int_0^s d\gamma(s)(\beta' - \beta)$, we have by Lemma E.3:

$$\|x_{\beta_s} - x_\beta\| \leq \frac{LR}{\mu}\|\beta - \beta'\|_1 \cdot s.$$

We also have $|d\gamma(s)| \leq \mu^{-1} LR\|\beta - \beta'\|_1$, by Lemma E.3. The first integral is bounded by:

$$\int_\gamma L_0\|x_{\beta_s} - x_\beta\|\,|d\gamma(s)| \leq \int_0^1 \frac{L_0 L^2 R^2}{\mu^2}\|\beta - \beta'\|_1^2 \cdot s\,ds = \frac{1}{2}\frac{L_0 L^2 R^2}{\mu^2}\|\beta - \beta'\|_1^2.$$

For the second integral, first note that condition (8b) implies:

$$-\nabla f_0(x_\beta)^\top d\gamma(s) = -\nabla f_0(x_\beta)^\top \nabla x^*(x_\beta)(\beta' - \beta) \leq \varepsilon_0\|\beta - \beta'\|_1,$$

yielding the other bound:

$$\int_\gamma \left(-\nabla f_0(x_\beta)^\top d\gamma(s)\right) \leq \int_0^1 \varepsilon_0\|\beta - \beta'\|_1\,ds = \varepsilon_0\|\beta - \beta'\|_1.$$

Putting these two together, we obtain:

$$f_0(x_\beta) - f_0(x_{\beta'}) \leq \frac{1}{2} \frac{L_0 L^2 R^2}{\mu^2} \|\beta - \beta'\|_1^2 + \varepsilon_0 \|\beta - \beta'\|_1.$$

It follows that if we restrict $\|\beta - \beta'\|_1 \leq \frac{2\mu^2 \varepsilon_0}{L_0 L^2 R^2}$, one of the factors of $\|\beta - \beta'\|_1$ in the first term can be absorbed into the constant, proving the result:

$$f_0(x_\beta) \leq f_0(x_{\beta'}) + 2\varepsilon_0 \|\beta - \beta'\|_1.$$

$\square$

**Lemma 7** (Verifiability of approximate stationarity). *Under Assumptions A,B,C, the point $(\hat{x}, \hat{\beta})$ is $(\varepsilon_0, \varepsilon)$-preference stationary if $\|\nabla f_{\hat{\beta}}(\hat{x})\|_2 \leq \varepsilon$, and for some $x \in \mathbb{R}^d$ and $\alpha \in (0,1)$,*

*(a) an $\alpha \cdot \varepsilon_0$-approximate stationary condition holds for all $\beta' \in \Delta^{n-1}$:*

$$-\nabla f_0(x)^\top \widehat{\nabla} x^*(x, \hat{\beta})(\beta' - \hat{\beta}) \leq \alpha \cdot \varepsilon_0 \|\beta' - \hat{\beta}\|_1, \tag{11}$$

*(b) an error bound holds: $\mathrm{err}_{\nabla f_0}(\hat{\beta}, x) \leq (1 - \alpha) \cdot \varepsilon_0$, where:*

$$\mathrm{err}_{\nabla f_0}(x, \beta) := \frac{1}{\mu} \left( \frac{M_1}{2M_0} \|\nabla f_0(x)\|_2 + L_0 M_0 \right) \|\nabla f_\beta(x)\|_2,$$

*and $M_0 = \kappa R$, $M_1 = 2\kappa^2 R(1 + L_H R/\mu)$.*

*Proof.* For $(\varepsilon, \varepsilon_0)$-preference stationarity, we require $\|\nabla f_{\hat{\beta}}(\hat{x})\|_2 \leq \varepsilon$ and:

$$\nabla f_0(x_{\hat{\beta}})^\top \nabla x^*(\hat{\beta})(\beta' - \hat{\beta}) + \varepsilon_0 \cdot \|\beta' - \hat{\beta}\|_1 \geq 0.$$

Then by Lemma E.6, the left-hand side is lower bounded:

$$\nabla f_0(x)^\top \widehat{\nabla} x^*(x, \hat{\beta})(\beta' - \hat{\beta}) - \mathrm{err}_{\nabla f_0}(\hat{\beta}, x) \cdot \|\beta' - \hat{\beta}\|_1 + \varepsilon_0 \|\beta' - \hat{\beta}\|_1,$$

$$= \underbrace{\nabla f_0(x)^\top \widehat{\nabla} x^*(x, \hat{\beta})(\beta' - \hat{\beta}) + \alpha \cdot \varepsilon_0 \|\beta' - \hat{\beta}\|_1}_{\geq 0}$$

$$+ \underbrace{(1 - \alpha) \cdot \varepsilon_0 \|\beta' - \hat{\beta}\|_1 - \mathrm{err}_{\nabla f_0}(\hat{\beta}, x) \cdot \|\beta' - \hat{\beta}\|_1}_{\geq 0},$$

for $\alpha \in (0, 1)$. The two terms are lower bounded by zero by conditions (1) and (2). $\square$

# F  Proofs for Section 6

**Proposition 8** (A family of majorizing surrogates). *Let $F$ and $f_0$ satisfy Assumptions A,B,C. Let* $\mathrm{err}_{\nabla f_0}(x, \beta)$ *be as defined in Lemma E.6. Define the family indexed by* $(x, \beta) \in \mathbb{R}^d \times \Delta^{n-1}$:

$$g(\beta'; x, \beta) := f_0(x_\beta) + \nabla f_0(x)^\top \widehat{\nabla} x^*(x, \beta)(\beta' - \beta) + \tfrac{1}{2}\mu_g\|\beta' - \beta\|_2^2 + \mathrm{err}_{\nabla f_0}(x, \beta), \quad (13)$$

*where $\mu_g := nL_0M_1$. Then $g(\beta'; x, \beta)$ majorizes $f_0 \circ x^*$, satisfying* (12).

*Proof.* As $f_0$ and $x^*$ are respectively $L_0$- and $M_1$-Lipschitz smooth (Assumption C and Lemma E.5), their composition is $L_0M_1$-Lipschitz smooth, and so it admits the standard quadratic upper bound (e.g., Lemma 1.2.3, Nesterov (2003)):

$$f_0(x_{\beta'}) \le f_0(x_\beta) + \nabla(f_0 \circ x^*)(\beta)^\top(\beta' - \beta) + \frac{1}{2}nL_0M_1\|\beta' - \beta\|_2^2,$$

where we have used $\|\beta' - \beta\|_1^2 \le n\|\beta' - \beta\|_2^2$. This could be an easy choice for our majorizing surrogate $g$ but the gradient $\nabla(f_0 \circ x^*)(\beta)$ is implicit. Substituting in the approximation $\nabla f_0(x)^\top \widehat{\nabla} x^*(x, \beta)$ yields the result, where the error term comes from Lemma E.6. $\square$

## G  Convergence for Pareto majorization-minimization

**Theorem 10** (Convergence of PMM with first-order feedback). *Suppose that $F$ and $f_0$ satisfy Assumptions A to C, and that the black-box optimizers satisfy Assumption E. Fix $0 < \varepsilon^{1/2} \leq \varepsilon_0 \leq 1$. Let $(x_k, \beta_k)_k$ be the iterates of Algorithm 1 using the family of surrogates (13). There exist $c_1(f_0, F)$ and $c_2(f_0, F)$ bounded away from zero and a stopping time $K$ such that $(f_0 \circ x^*)(\beta_k)$ is monotonically decreasing for $k \in [K]$ and $(x_K, \beta_K)$ is an $(\varepsilon_0, \varepsilon)$-preference stationary point. Furthermore:*

$$K \leq \frac{2\mu_g \cdot (f^* - f_*)}{c_1^2 \cdot \varepsilon_0^2},$$

*where $f^* := \max f_0(x)$ and $f_* = \min f_0(x)$ are optimized over the compact set $\mathrm{Pareto}(F)$.*

*Proof.* Fix $k > 1$. For short, we let:

$$(x, \beta) \equiv (x_{k-1}, \beta_{k-1}) \qquad \text{and} \qquad (\hat{x}, \hat{\beta}) \equiv (x_k, \beta_k).$$

*Claim.* At each iteration, either (i) the preference improves by at least a constant:

$$f_0(x_{\hat{\beta}}) - f_0(x_{\beta}) \leq -\frac{1}{2}\frac{c_1}{\mu_g} \cdot \varepsilon_0^2,$$

or (ii) the point $(\hat{x}, \hat{\beta})$ is $(\varepsilon_0, \varepsilon)$-preference stationary.

Assuming the claim holds, the theorem immediately follows: if the algorithm in $K$ steps has not found an $(\varepsilon_0, \varepsilon)$-preference stationary point, then the value $f_0(x_{\beta_k})$ must decrease every iteration by a constant. But because $f_0 \circ x^*$ is lower bounded over $\Delta^{n-1}$ by $f^*$, this can happen at most:

$$\frac{2\mu_g \cdot (f^* - f_*)}{c_1^2 \cdot \varepsilon_0^2} \quad \text{times.}$$

*Proof of the claim.* Let $\beta^* := \arg\min_{\beta' \in \Delta^{n-1}} g(\beta'; x, \beta)$. Lemma G.4 shows that an approximate stationary point $\hat{\beta}$ of a strongly convex function is close to the exact point $\beta^*$:

$$\|\hat{\beta} - \beta^*\|_2 \leq \frac{c_1 \varepsilon_0}{\mu_g} =: \delta, \tag{21}$$

where we let $\delta$ denote this constant for short.

We can analyze $\hat{\beta}$ through $\beta^*$. There are two cases, leading to either (1) $O(\varepsilon_0)$-preference stationarity or (2) $O(\varepsilon_0^2)$-constant descent. The two cases depend on the suboptimality of $\beta$.

Case 1: $\|\beta^* - \beta\|_2 < 2\delta$. Here, $\beta$ is fairly close to the optimum $\beta^*$ of the surrogate. We show that the approximate stationarity of $\hat{\beta}$ with respect to the surrogate implies approximate preference stationarity. We do so via Lemma 7, which states that $(\hat{x}, \hat{\beta})$ is $(\varepsilon_0, \varepsilon)$-preference stationary provided:

$$\|\nabla f_{\hat{\beta}}(\hat{x})\|_2 \leq \varepsilon \tag{22}$$

$$-\nabla f_0(x)^\top \widehat{\nabla} x^*(x, \hat{\beta})(\beta' - \hat{\beta}) \leq \frac{1}{2}\varepsilon_0 \|\beta' - \hat{\beta}\|_1, \qquad \forall \beta' \in \Delta^{n-1} \tag{23}$$

$$\mathrm{err}_{\nabla f_0}(x, \hat{\beta}) \leq \frac{1}{2}\varepsilon_0 \tag{24}$$

While (22) is immediate from our choice of $c_2$, defined in the last section of the proof, the others do not follow automatically from approximate stationarity with respect to the surrogate: the surrogate is derived from local information at $(x, \beta)$, while we would like guarantees at $(x, \hat{\beta})$. But because $\beta^*$ is close to both $\beta$ and $\hat{\beta}$, we can control all of these. By triangle inequality:

$$\|\beta - \hat{\beta}\|_2 \leq \|\beta - \beta^*\|_2 + \|\beta^* - \hat{\beta}\|_2 < 3\delta, \tag{25}$$

combining (21) and the assumption that $\|\beta^* - \beta\|_2 < 2\delta$.

We now show (23). We have for all $\beta' \in \Delta^{n-1}$,

$$-\nabla f_0(x)^\top \widehat{\nabla} x^*(x, \hat{\beta})(\beta' - \hat{\beta})$$

$$\overset{(i)}{\leq} -\nabla f_0(x)^\top \widehat{\nabla} x^*(x, \beta)(\beta' - \hat{\beta}) + \|\nabla f_0(x)^\top (\widehat{\nabla} x^*(x, \hat{\beta}) - \widehat{\nabla} x^*(x, \beta))\|_\infty \cdot \|\beta' - \hat{\beta}\|_1$$

$$\overset{(ii)}{\leq} c_1 \varepsilon_0 \cdot \|\beta' - \hat{\beta}\|_1 + \|\nabla f_0(x)\|_2 \cdot \frac{L}{\mu^2} \|\beta - \hat{\beta}\|_2 \cdot \|\beta' - \hat{\beta}\|_1$$

$$\overset{(iii)}{\leq} \frac{1}{2} \cdot 2 \left( c_1 \varepsilon_0 + \frac{L\|\nabla f_0(x)\|_2}{\mu^2} \cdot 3\delta \right) \cdot \|\beta' - \hat{\beta}\|_1 \tag{26}$$

$$\overset{(iv)}{\leq} \frac{1}{2} \varepsilon_0 \cdot \|\beta' - \hat{\beta}\|_1,$$

where (i) adds and subtracts $\nabla f_0(x)^\top \widehat{\nabla} x^*(x, \beta)(\beta' - \hat{\beta})$ and applies Hölder's inequality, (ii) substitutes in Condition 1 for the first term and bounds the second via Lemma G.3, and (iii) bounds $\|\beta - \hat{\beta}\|_2$ using (25), and (iv) applies the definition of $c_1$, set in the last section of the proof.

To show (24), we have:

$$\mathrm{err}_{\nabla f_0}(x, \hat{\beta}) \overset{(i)}{=} \mathrm{err}_{\nabla f_0}(x, \beta) + \frac{1}{\mu} \left( \frac{M_1}{2M_0} \|\nabla f_0(x)\|_2 + L_0 M_0 \right) \left( \|\nabla f_{\hat{\beta}}(x)\|_2 - \|\nabla f_\beta(x)\|_2 \right)$$

$$\overset{(ii)}{\leq} \mathrm{err}_{\nabla f_0}(x, \beta) + \frac{1}{\mu} \left( \frac{M_1}{2M_0} \|\nabla f_0(x)\|_2 + L_0 M_0 \right) \|\nabla F(x)^\top\|_2 \cdot \|\hat{\beta} - \beta\|_2$$

$$\overset{(iii)}{\leq} \frac{1}{2} \cdot \frac{2}{\mu} \left( \frac{M_1}{2M_0} \|\nabla f_0(x)\|_2 + L_0 M_0 \right) \left\{ c_2 \varepsilon + \|\nabla F(x)^\top\|_2 \cdot 3\delta \right\} \tag{27}$$

$$\overset{(iv)}{\leq} \frac{1}{2} \varepsilon_0.$$

where (i) expands out $\mathrm{err}_{\nabla f_0}$, (ii) uses the fact that $\beta \mapsto \|\nabla F(x)^\top \beta\|_2$ is $\|\nabla F(x)^\top\|_2$-Lipschitz in $\beta$ with respect to the $\ell_2$-norm, (iii) applies the definition of $\mathrm{err}_{\nabla f_0}$ and the inequality (25), and (iv) follows by definition of $c_1$ and $c_2$, set in the last section of the proof.

As (22, 23, 24) hold, Lemma 7 shows that $(\hat{x}, \hat{\beta})$ is $(\varepsilon_0, \varepsilon)$-preference stationary.

Case 2: $\|\beta^* - \beta\|_2 \geq 2\delta$. Here $\beta$ is suboptimal and $\beta^*$ achieves a large descent:

$$f_0(x_{\beta^*}) - f_0(x_\beta) \overset{(i)}{\leq} g(\beta^*; x, \beta) - f_0(x_\beta)$$

$$\overset{(ii)}{\leq} \mathrm{err}_{\nabla f_0}(x, \beta) - \frac{1}{2} \mu_g \|\beta^* - \beta\|_2^2$$

$$\overset{(iii)}{\leq} \frac{1}{\mu} \left( \frac{M_1}{2M_0} \|\nabla f_0(x)\|_2 + L_0 M_0 \right) c_2 \varepsilon_0^2 - 2\mu_g \delta^2 \tag{28}$$

$$\overset{(iv)}{\leq} -\frac{3}{2} \mu_g \delta^2, \tag{29}$$

where (i) uses the majorizing property of $g$, (ii) follows from Lemma G.5, (iii) applies the definition of $\mathrm{err}_{\nabla f_0}(x, \beta)$ and the assumption that $\varepsilon \leq \varepsilon_0^2$, and (iv) uses the definition of $c_2$.

The large descent also carries over to $\hat{\beta}$ because it is approximately stationary:

$$f_0(x_{\hat{\beta}}) - f_0(x_\beta) \overset{(i)}{\leq} g(\hat{\beta}; x, \beta) - f_0(x_\beta)$$

$$\overset{(ii)}{=} g(\beta^*; x, \beta) - f_0(x_\beta) + \left( g(\hat{\beta}; x, \beta) - g(\beta^*; x, \beta) \right)$$

$$\overset{(iii)}{\leq} -\frac{3}{2} \mu_g \delta^2 + c_1 \varepsilon_0 \cdot \delta = -\frac{1}{2} \frac{c_1}{\mu_g} \cdot \varepsilon_0^2,$$

where (i) uses the majorizing property of $g$, (ii) adds and subtracts $g(\beta^*; x, \beta)$ and (iii) applies (29) and Lemma G.4.

Thus, the preference improves by at least a constant. To finish proving the claim, we need to verify that it is indeed possible to set $c_1$ and $c_2$ appropriately.

Setting $c_1$ and $c_2$: we tabled a few inequalities above. Recall:

For (22), we need:

$$c_2 \leq 1.$$

For (26), we need:

$$2 \left( c_1 \varepsilon_0 + \frac{3L \|\nabla f_0(x)\|_2}{\mu^2} \cdot \frac{c_1 \varepsilon_0}{\mu_g} \right) \leq \varepsilon_0.$$

For (27), we need:

$$\frac{2}{\mu} \left( \frac{M_1}{2M_0} \|\nabla f_0(x)\|_2 + L_0 M_0 \right) \left\{ c_2 \varepsilon + 3 \|\nabla F(x)^\top\|_2 \cdot \frac{c_1 \varepsilon_0}{\mu_g} \right\} \leq \varepsilon_0.$$

For (28), we need:

$$\frac{1}{\mu} \left( \frac{M_1}{2M_0} \|\nabla f_0(x)\|_2 + L_0 M_0 \right) c_2 \varepsilon_0^2 \leq \frac{1}{2} \mu_g \left( \frac{c_1 \varepsilon_0}{\mu_g} \right)^2.$$

It is unenlightening but straightforward to verify that it suffices to set:

$$c_1 \cdot \max \left\{ 2 + \frac{6L \|\nabla f_0(x)\|_2}{\mu^2 \cdot \mu_g}, \frac{12}{\mu \cdot \mu_g} \left( \frac{M_1}{2M_0} \|\nabla f_0(x)\|_2 + L_0 M_0 \right) \cdot \|\nabla F(x)^\top\|_2 \right\} \leq 1$$

$$c_2 \cdot \max \left\{ 1, \frac{2}{\mu} \left( \frac{M_1}{2M_0} \|\nabla f_0(x)\|_2 + L_0 M_0 \right) \cdot \left( 2 \vee \frac{\mu_g}{c_1^2} \right) \right\} \leq 1,$$

where $a \vee b := \max\{a, b\}$.

A concerned reader may wonder whether $c_1$ and $c_2$ may be bounded away from zero, as claimed in the theorem statement: we need to ensure that $\|\nabla f_0(x)\|_2$ and $\|\nabla F(x)^\top\|_2$ do not blow up. Indeed, this holds because the iterates $x_k$ remain within a constant distance of the Pareto set. In particular, since $c_2 \leq 1$, by Condition 2, we have that the $k$th iterate satisfies:

$$\|x_k - x_{\beta_k}\| \leq \frac{\varepsilon}{\mu},$$

which follows from $\mu$-strong convexity of $f_{\beta_k}$. Thus, all iterates of the algorithm are within $\varepsilon/\mu$ of the Pareto set and also satisfy for all $k, k' \in \mathbb{N}$:

$$\|x_k - x_{k'}\| \leq R + 2\varepsilon/\mu.$$

Then, by $L_0$-Lipschitz smoothness, we can bound:

$$\|\nabla f_0(x_k)\| \leq \|\nabla f_0(x_1)\| + \|\nabla f_0(x_k) - \nabla f_0(x_1)\|$$
$$\leq \|\nabla f_0(x_1)\| + L_0 \cdot (R + 2\varepsilon/\mu). \tag{30}$$

Similarly, by $L$-Lipschitz smoothness, we also have:

$$\|\nabla F(x_k)^\top\|_2 \leq \|\nabla F(x_1)^\top\|_2 + \|\nabla F(x_k)^\top - \nabla F(x_1)^\top\|_2$$
$$\leq \|\nabla F(x_1)^\top\|_2 + nL \cdot (R + 2\varepsilon/\mu). \tag{31}$$

$\square$

### G.1 A bound on the gradient

**Lemma G.1.** *Let* $R := \mathrm{diam}\big(\mathrm{Pareto}(F)\big)$. *Then for any* $x_\beta = x^*(\beta)$,

$$\left\| \nabla F(x_\beta)^\top \right\|_{1,2} \leq LR.$$

*Proof.* By definition, we have:

$$\left\|\nabla F(x_\beta)^\top\right\|_{1,2} = \sup_{\|z\|_1=1} \left\|\sum_{i\in[n]} z_i \nabla f_i(x_\beta)\right\|_2$$

$$\overset{(i)}{\leq} \sup_{\|z\|_1=1} \sum_{i\in[n]} |z_i| \cdot \|\text{sign}(z_i) \cdot \nabla f_i(x_\beta)\|_2$$

$$\overset{(ii)}{\leq} \max_{i\in[n]} \|\nabla f_i(x_\beta)\|_2$$

$$\overset{(iii)}{\leq} \max_{i\in[n]} L\|x - x_i\|_2,$$

where (i) follows from Jensen's inequality, (ii) holds because the max is no smaller than the average, (iii) applies $L$-Lipschitz smoothness. In particular, let $x_i = \arg\min f_i(x)$, so that $\nabla f_i(x_i) = 0$. Then:

$$\|\nabla f_i(x_\beta) - \nabla f_i(x_i)\|_2 \leq L\|x_\beta - x_i\|_2.$$

The result holds because $x_\beta$ and all $x_i$'s are contained in $\text{Pareto}(F)$. $\qquad\square$

### G.1.1 Lemmas: matrix inverses

**Lemma G.2.** *Let $M : \mathbb{R}^d \to \mathbb{R}^{d\times d}$ be $L$-Lipschitz satisfying $M(x) \succeq \mu\mathbf{I}$ where $\mathbb{R}^d$ has the $\ell_2$-norm and $\mathbb{R}^{d\times d}$ the operator norm. Then, the map $x \mapsto M(x)^{-1}$ is $L/\mu^2$-Lipschitz.*

*Proof.* For short, let us denote $M(x)$ by $M_x$. Note that $\mathbf{I} = (M_x' + M_x - M_x')M_x^{-1}$, so that:

$$M_x^{-1} - M_{x'}^{-1} = M_x^{-1} - M_{x'}^{-1}(M_{x'} + M_x - M_{x'})M_x^{-1}$$
$$= M_x^{-1} - M_x^{-1} - M_{x'}^{-1}(M_x - M_{x'})M_x^{-1} = -M_{x'}^{-1}(M_x - M_{x'})M_x^{-1},$$

which is series of unenlightening algebraic manipulations. But now, we may apply $L$-Lipschitz continuity to obtain $\|M_x - M_{x'}\| \leq L\|x - x'\|$ and the $\mu$-lower bound to obtain $\|M_x^{-1}\|, \|M_{x'}^{-1}\| \leq \mu^{-1}$. Together, we obtain $L/\mu^2$-Lipschitz continuity:

$$\left\|M(x)^{-1} - M(x')^{-1}\right\| \leq \frac{L}{\mu^2}\|x - x'\|.$$

$\qquad\square$

**Lemma G.3.** *Let $M_1, \ldots, M_n$ be positive-definite matrices in $\mathbb{R}^{d\times d}$ equipped with the operator norm, and let $\Delta^{n-1}$ be equipped with the $\ell_1$ norm. Suppose the following holds:*

$$\mu\mathbf{I} \preceq M_1, \ldots, M_n \preceq L\mathbf{I}.$$

*The map $\beta \mapsto M_\beta^{-1}$ where $M_\beta := \sum_{i\in[n]} \beta_i M_i$ has bounded derivative $\|\nabla_\beta M_\beta^{-1}\|_{1,2} \leq L/\mu^2$.*

*Proof.* We can compute the derivative of the above map:

$$\nabla_\beta M_\beta^{-1} = -M_\beta^{-1}(\nabla_\beta M_\beta)M_\beta^{-1},$$

where $\nabla_\beta M_\beta d\beta = M_{d\beta}$. The upper bound on the $M_i$'s implies that $\|\nabla_\beta M_\beta\|_{1,2} \leq L$. On the other hand, the lower bound implies that $\|M_\beta^{-1}\|_2 \leq \mu^{-1}$. $\qquad\square$

### G.1.2 Lemmas: constrained optimization of strongly convex functions

**Lemma G.4.** *Let $f : \mathbb{R}^n \to \mathbb{R}$ be smooth and convex and let $\mathcal{C} \subset \mathbb{R}^n$ be a convex constraint set. Suppose that $\beta^*, \hat\beta \in \mathcal{C}$ are stationary and $\varepsilon$-approximately stationary, respectively:*

$$-\nabla f(\beta^*)^\top(\beta - \beta^*) \leq 0 \quad and \quad -\nabla f(\hat\beta)^\top(\beta - \hat\beta) \leq \varepsilon\|\beta - \hat\beta\|, \quad \forall\beta \in \mathcal{C}.$$

*Then, $f(\hat\beta) - f(\beta^*) \leq \varepsilon\|\hat\beta - \beta^*\|$. Furthermore, if $f$ is $\mu$-strongly convex, then $\|\hat\beta - \beta^*\| \leq \varepsilon/\mu$.*

*Proof.* For the first part, we apply the mean value theorem, which states that there exists some $\beta$ that is a convex combination of $\hat{\beta}$ and $\beta^*$ such that:

$$f(\hat{\beta}) - f(\beta^*) \overset{(i)}{=} \nabla f(\beta)^\top (\hat{\beta} - \beta^*)$$
$$\overset{(ii)}{\leq} \nabla f(\hat{\beta})^\top (\hat{\beta} - \beta^*)$$
$$\overset{(iii)}{\leq} \varepsilon \|\hat{\beta} - \beta^*\|,$$

where (i) applies the mean value theorem, (ii) uses the monotonicity of gradients of convex functions:

$$\left( \nabla f(\hat{\beta}) - \nabla f(\beta) \right)^\top (\hat{\beta} - \beta) \geq 0$$

and that $\hat{\beta} - \beta = \lambda(\hat{\beta} - \beta^*)$ for some $\lambda \in [0, 1]$, and (iii) applies the $\varepsilon$-stationarity condition.

For the second part, by strong convexity, we have on the one hand:

$$\left( \nabla f(\hat{\beta}) - \nabla f(\beta^*) \right)^\top (\hat{\beta} - \beta^*) \geq \mu \|\hat{\beta} - \beta^*\|^2.$$

And on the other, by stationarity and $\varepsilon$-stationarity, we have that:

$$\left( \nabla f(\hat{\beta}) - \nabla f(\beta^*) \right)^\top (\hat{\beta} - \beta^*) \geq \varepsilon \|\hat{\beta} - \beta^*\|.$$

Dividing through by $\|\hat{\beta} - \beta^*\|$ yields the result. $\qquad\square$

**Lemma G.5.** *Let $\mathcal{C} \subset \mathbb{R}^n$ be a convex constraint set with $\beta \in \mathcal{C}$, and let $Q : \mathcal{C} \to \mathbb{R}$ be a quadratic:*

$$Q(\beta') = c + v^\top (\beta' - \beta) + \frac{1}{2} C \|\beta' - \beta\|^2, \tag{32}$$

*where $c \in \mathbb{R}$, $v \in \mathbb{R}^n$, and $C > 0$. Let $\beta^* \in \mathcal{C}$ minimize $Q$. If $\|\beta^* - \beta\| \geq \varepsilon > 0$, then:*

$$Q(\beta^*) - Q(\beta) \leq -\frac{1}{2} C \varepsilon^2.$$

*Proof.* Define the quadratic function $q : \mathbb{R} \to \mathbb{R}$ by:

$$q(\lambda) = c + \lambda v^\top (\beta^* - \beta) + \frac{1}{2} C \lambda^2 \|\beta^* - \beta\|^2$$
$$= c + \frac{1}{2} C \|\beta^* - \beta\|^2 \lambda (\lambda - 2\lambda^*) \tag{33}$$

where $\lambda^* = -\frac{v^\top (\beta^* - \beta)}{C \|\beta^* - \beta\|^2}$ minimizes $q$. Restricting $Q$ to the line between $\beta$ and $\beta^*$, we get:

$$Q(\beta + \lambda(\beta^* - \beta)) = q(\lambda),$$

for $\lambda \in [0, 1]$. This follows by expanding the definition of $Q$.

Notice that $q$ monotonically decreases on the interval $0 \leq \lambda \leq \lambda^*$, and also that $q$ monotonically increases for $\lambda > \lambda^*$. Because $Q(\beta^*) = q(1)$ minimizes $Q$ on the convex set $\mathcal{C}$, $q$ must be descending on $\lambda \in [0, 1]$. Thus, $1 \leq \lambda^*$. It follows that $1 - 2\lambda^* \leq -1$. Plugging in into (33), we have:

$$Q(\beta^*) = q(1) \leq c - \frac{1}{2} C \|\beta^* - \beta\|^2.$$

Applying $Q(\beta_0) = c$ and $\|\beta^* - \beta\| \geq \varepsilon$ yields the result. $\qquad\square$

# H  Proofs for Dueling Feedback

## H.1  Proof of Lemma 9

**Lemma 9** (Bias and variance of dueling gradient estimator)**.** *Under Assumptions A to D, let $\hat{\nabla}f_0(x)$ be defined as in (14) with $\alpha = 1/8$, $m \asymp d^4 \log(d/\varepsilon_0)/\varepsilon_0^4$, $b \asymp d/\varepsilon_0^2$, and $\gamma \asymp \varepsilon_0/d$. Then:*

$$\|\mathbb{E}[\hat{\nabla}f_0(x)] - \nabla f_0(x)\|_2 \lesssim \varepsilon_0, \qquad \text{and} \qquad \mathbb{E}[\|\hat{\nabla}f_0(x) - \nabla f_0(x)\|_2^2] \lesssim \varepsilon_0^2,$$

Before proving Lemma 9, we first need the following lemma.

**Lemma H.1.** *Let Assumption D be true. Then, we have,*

$$\mathbb{E}\left[\left|\sigma^{-1}(\hat{p}) - \sigma^{-1}(p)\right|^2\right] \le 4\exp(-2m\beta^2) + \beta^2$$
$$+ \left(\exp\left(-2m(1 - \alpha - 1/l)^2\right) + \exp\left(-2m(1/u - \alpha)^2\right)\right) L_\sigma^2 C_\alpha^2 \left(|l^{-1} - \alpha|^2 + |\alpha - u^{-1}|^2\right).$$

*Proof.* We will omit the subscripts and write $\hat{p}_i$, $\tilde{p}_i = \frac{1}{m}\sum_{j\in[m]} Y_{ij}$, $p_{x+\gamma u_i, x}$ as $\hat{p}$, $\tilde{p}$, $p$ when there is no confusion. Consider the following three events.

1. For some constants $u, l > 1$, define event $E_1 := \{\alpha < \tilde{p} < 1 - \alpha \mid 1/u < p < 1/l\}$ such that $\alpha > 1/u$, and $1 - \alpha < 1/l$.

2. $E_2 := \{\tilde{p} \ge 1 - \alpha \mid 1/u < p < 1/l\}$.

3. $E_3 := \{\tilde{p} \le \alpha \mid 1/u < p < 1/l\}$.

By Hoeffding's inequality, we have,

$$P(E_1) \ge 1 - \exp\left(-2m(1 - \alpha - 1/l)^2\right) - \exp\left(-2m(1/u - \alpha)^2\right),$$
$$P(E_2) \le \exp\left(-2m(1 - \alpha - 1/l)^2\right), \qquad P(E_3) \le \exp\left(-2m(1/u - \alpha)^2\right). \tag{34}$$

Let $\nu := \tilde{p} - p$. Let the event $E_1$ be true. Then we have $\tilde{p} = \hat{p}$, and

$$\left|\sigma^{-1}(\hat{p}) - \sigma^{-1}(p)\right| \le L_\sigma C_\alpha |\nu|.$$

where $C_\alpha = \left(1 + \frac{1}{\alpha(1-\alpha)} + \frac{ul}{(l-1)}\right)$. For sufficiently small $\max(p - \alpha, 1 - \alpha - p) > \beta > 0$, consider the event, $E_4 = \{|\nu| > \beta \mid E_1\}$.

$$P(E_4) \le \frac{\exp(-2m\beta^2)}{P(E_1)}.$$

Then, since $|\nu| \le 2$,

$$\mathbb{E}\left[|\nu|^2 \mid E_1\right] \le 4\frac{\exp(-2m\beta^2)}{P(E_1)} + \beta^2. \tag{35}$$

Under $E_2$,

$$\left|\sigma^{-1}(\hat{p}) - \sigma^{-1}(p)\right| \le L_\sigma C_\alpha |\hat{p} - p| \le L_\sigma C_\alpha |\alpha - u^{-1}|. \tag{36}$$

Under $E_3$,

$$\left|\sigma^{-1}(\hat{p}) - \sigma^{-1}(p)\right| \le L_\sigma C_\alpha |\hat{p} - p| \le L_\sigma C_\alpha |l^{-1} - \alpha|. \tag{37}$$

Combining (34), (35), (36), and (37), we obtain,

$$\mathbb{E}\left[\left|\sigma^{-1}(\hat{p}) - \sigma^{-1}(p)\right|^2\right]$$
$$\le P(E_1)L_\sigma^2 C_\alpha^2 \mathbb{E}\left[|\nu|^2\right] + (P(E_2) + P(E_3)) L_\sigma^2 C_\alpha^2 \left(|l^{-1} - \alpha|^2 + |\alpha - u^{-1}|^2\right)$$
$$\le 4\exp(-2m\beta^2) + \beta^2$$
$$+ \left(\exp\left(-2m(1 - \alpha - 1/l)^2\right) + \exp\left(-2m(1/u - \alpha)^2\right)\right) L_\sigma^2 C_\alpha^2 \left(|l^{-1} - \alpha|^2 + |\alpha - u^{-1}|^2\right).$$

□

*Proof of Lemma 9.* Since Algorithm 1 ensures that $x_k$ are contained in a compact set, Assumption C implies that $\|\nabla f_0\|_2 \leq G$ for some constant $G > 0$ as observed in (30).[4]

Then, $|f_0(x + \gamma u_i) - f_0(x - \gamma u_i)| \leq G\gamma$ implying $l \leq p^{-1}_{x+\gamma u_i, x-\gamma u_i} \leq u$, where,

$$l = \sigma\left(-\gamma G\right)^{-1}, \qquad \text{and} \qquad u = \sigma\left(\gamma G\right)^{-1}, \tag{38}$$

By Lipschitz continuity of $\sigma(\cdot)$, using $\sigma(0) = 1/2$, and $\gamma \leq 1/(4L_1 G)$, we have, $(l, u) = (4/3, 4)$. First we look at the bias of $\hat{\nabla} f_0(x)$.

$$\left\|\mathbb{E}[\hat{\nabla} f_0(x)] - \nabla f_0(x)\right\|_2$$

$$\leq \left\|\mathbb{E}\left[\frac{d}{2\gamma b}\sum_{i=1}^{b}\left(\sigma^{-1}\left(\hat{p}_{x+\gamma u_i, x-\gamma u_i}\right)u_i - \sigma^{-1}\left(p_{x+\gamma u_i, x-\gamma u_i}\right)u_i\right)\right]\right\|_2$$

$$+ \left\|\frac{d}{2\gamma b}\sum_{i=1}^{b}\mathbb{E}\left[\sigma^{-1}\left(p_{x+\gamma u_i, x-\gamma u_i}\right)u_i\right] - \nabla f_0(x)\right\|_2$$

$$\leq \frac{d}{2\gamma}\left(2\exp(-m\beta^2) + \beta + 2L_\sigma C_\alpha \exp\left(-m/64\right)\right) + 3L_0 d\gamma$$

$$\leq \frac{d(3 + 2L_\sigma C_\alpha)\beta}{2\gamma} + 3L_0 d\gamma$$

$$\leq c_3 \varepsilon_0,$$

where $c_3 > 0$ is a constant. The first inequality follows by triangle inequality, the second inequality follows by Lemma H.1, Section 3 of Agarwal et al. (2010), and choosing $\alpha = \frac{1}{8}$, and the third inequality follows by choosing $m = \max\left(1/\beta^2, 64\right)\log(1/\beta)$, the fourth inequality follows by choosing $\beta = \gamma^2$, and the fifth inequality follows by choosing $\gamma = \varepsilon_0/d$. Note that with these choice of $\alpha, u, l$, one has $C_\alpha$ is bounded by a constant 27.

Now consider the variance term.

$$\mathbb{E}\left[\left\|\hat{\nabla} f_0(x) - \nabla f_0(x)\right\|_2^2\right]$$

$$\leq 2\mathbb{E}\left[\left\|\hat{\nabla} f_0(x) - \mathbb{E}\left[\frac{d}{2\gamma}\sigma^{-1}\left(p_{x+\gamma u, x-\gamma u}\right)u\right]\right\|_2^2\right]$$

$$+ 2\mathbb{E}\left[\left\|\mathbb{E}\left[\frac{d}{2\gamma}\sigma^{-1}\left(p_{x+\gamma u, x-\gamma u}\right)u\right] - \nabla f_0(x)\right\|_2^2\right]$$

$$\leq 2\mathbb{E}\left[\left\|\hat{\nabla} f_0(x) - \mathbb{E}\left[\frac{d}{2\gamma}\sigma^{-1}\left(p_{x+\gamma u, x-\gamma u}\right)u\right]\right\|_2^2\right] + 6L_0^2 d^2 \gamma^2$$

$$\leq \frac{2d^2}{\gamma^2 b}\sum_{i=1}^{b}\mathbb{E}\left[\left|\sigma^{-1}\left(\hat{p}_{x+\gamma u_i, x-\gamma u_i}\right) - \sigma^{-1}\left(p_{x+\gamma u_i, x-\gamma u_i}\right)\right|^2\right]$$

$$+ \frac{d^2}{\gamma^2 b^2}\sum_{i=1}^{b}\text{Var}\left(\sigma^{-1}\left(p_{x+\gamma u_i, x-\gamma u_i}\right)\right) + 6L_0^2 d^2 \gamma^2$$

$$\leq \frac{2d^2}{\gamma^2}\left(4\exp(-2m\beta^2) + \beta^2 + L_\sigma^2 C_\alpha^2 \exp\left(-m/32\right)\right) + \frac{72}{b}\left(G^2 d + \frac{L_0^2 d^2 \gamma^2}{2}\right) + 6L_0^2 d^2 \gamma^2$$

$$\lesssim \frac{2d^2(5 + L_\sigma^2 C_\alpha^2)\beta^2}{\gamma^2} + \frac{72}{b}\left(G^2 d + \frac{L_0^2 d^2 \gamma^2}{2}\right) + 6L_0^2 d^2 \gamma^2$$

$$\leq c_3^2 \varepsilon_0^2.$$

---

[4]To be precise, in the dueling feedback case, since we do not know $\nabla f_0(x)$, we need to make a mild assumption that there is at least one point $x_b \in \Omega$ where $\Omega$ is some compact set containing $\text{Pareto}(F)$ such that $\nabla f_0(x_b) \leq B$ where $B > 0$ is a constant. Then the algorithm can always be initiated at $x = x_b$ and the $\nabla f_0(x)$ remain uniformly bounded as per (30). This is a mild assumption because if the gradient is large everywhere, then the problem itself becomes meaningless to study. For all practical purposes, we can assume $\|\nabla f_0\|_2 \leq G$.

where the first inequality follows by Young's inequality, the second inequality follows by Section 3 of Agarwal et al. (2010), the third inequality follows by Young's and Cauchy Schwarz inequality, the fourth inequality follows by Lemma H.1, Lemma 2 of Bach and Perchet (2016), and choosing $\alpha = \frac{1}{8}$, the fifth inequality follows by choosing $m = \max\left(1/\beta^2, 64\right)\log(1/\beta)$, and the seventh inequality follows by choosing $b = d/\varepsilon_0^2$, $\beta = \gamma^2$, and $\gamma = \varepsilon_0/d$. $\qquad\square$

## H.2 Proof of Theorem 11

We expand Theorem 11 here.

**Theorem H.1** (Convergence of PMM with Dueling Feedback). *Let $F$, $f_0$, and $\sigma$ satisfy Assumptions A, B, C, and D. Fix $0 < \varepsilon^{1/2} \le \varepsilon_0 \le 1$. Let $\hat{x}_\beta$ and $\hat{\beta}$ be the approximate solutions that are returned by the black-box optimizer for $\hat{g}(\,\cdot\,; x, \beta)$ and $f_\beta(\cdot)$, defined in (13) and (2), respectively:*

$$\hat{\beta} \leftarrow \widehat{\arg\min_{\beta' \in \Delta^{n-1}}} \hat{g}(\beta'; x, \beta) \qquad and \qquad \hat{x}_\beta \leftarrow \widehat{\arg\min_{x \in \mathbb{R}^d}} f_{\hat{\beta}}(x).$$

*Given constants $c_1, c_2 > 0$, suppose the black-box optimizer achieves the following guarantees:*

1. *the approximate minimizer $\hat{\beta}$ is $O(\varepsilon_0)$-approximately stationary:*

$$-\nabla \hat{g}(\hat{\beta}; x, \beta)(\beta' - \hat{\beta}) \le c_1 \cdot \varepsilon_0 \|\beta' - \hat{\beta}\|_2, \qquad \forall \beta' \in \Delta^{n-1}.$$

2. *the approximate minimizer $\hat{x}_\beta$ is an $O(\varepsilon_0^2)$-approximate solution:*

$$\|\nabla f_{\hat{\beta}}(\hat{x}_\beta)\| \le c_2 \cdot \varepsilon.$$

*Let $(x_k, \beta_k)_k$ be the iterates of Algorithm 1 with dueling feedback. Then, choosing $\alpha = 1/8$, $m \asymp d^4 \log(d/\varepsilon_0)/\varepsilon_0^4$, $b \asymp d/\varepsilon_0^2$, and $\gamma \asymp \varepsilon_0/d$ in (14), there exist $c_1(f_0, F)$ and $c_2(f_0, F)$ bounded away from zero and some $K$ such that $\mathbb{E}[(f_0 \circ x^*)(\beta_k)]$ is monotonically decreasing for $k \in [K]$ and $(x_K, \beta_K)$ is an $(\varepsilon_0, \varepsilon)$-preference stationary point in expectation, i.e., $\|\nabla f_{\beta_K}(x_K)\|_2 \le \varepsilon$, and conditions (a) and (b) of Lemma 7 hold in expectation for $(x_K, \beta_K)$. In particular, we have the following.*

$$\mathbb{E}[-\nabla f_0(x_{K-1})^\top \widehat{\nabla} x^*(x_{K-1}, \beta_K)(\beta' - \beta_K)] \le \mathbb{E}\left[\frac{1}{2}\varepsilon_0 \|\beta' - \beta_K\|_1\right], \qquad \forall \beta' \in \Delta^{n-1}$$

$$\mathbb{E}[\mathrm{err}_{\nabla f_0}(x_{K-1}, \beta_K)] \le \frac{1}{2}\varepsilon_0$$

*Also,*

$$K \le \frac{2\mu_g \cdot \left(f^* - f_*\right)}{c_1^2 \cdot \varepsilon_0^2}.$$

**Proof of Theorem 11** Fix $k > 1$. For short, we let:

$$(x, \beta) \equiv (x_{k-1}, \beta_{k-1}) \qquad and \qquad (\hat{x}, \hat{\beta}) \equiv (x_k, \beta_k).$$

*Claim.* At each iteration, either (i) the preference improves by at least a constant on expectation:

$$\mathbb{E}[f_0(x_{\hat{\beta}}) - f_0(x_\beta)] \le -\frac{1}{2}\frac{c_1}{\mu_g} \cdot \varepsilon_0^2,$$

or (ii) the point $(\hat{x}, \hat{\beta})$ is $(\varepsilon_0, \varepsilon)$-preference stationary on expectation.

Assuming the claim holds, the theorem immediately follows: if the algorithm in $K$ steps has not found an $(\varepsilon_0, \varepsilon)$-preference stationary point, then $\mathbb{E}[f_0(x_{\beta_k})]$ must decrease every iteration by a constant. But as $f_0 \circ x^*$ is lower bounded over $\Delta^{n-1}$ by $f^*$, on expectation, this can happen at most:

$$\frac{2\mu_g \cdot \left(f^* - f_*\right)}{c_1^2 \cdot \varepsilon_0^2} \quad \text{times.}$$

*Proof of the claim.* Let $\beta^* := \arg\min_{\beta' \in \Delta^{n-1}} g(\beta'; x, \beta)$, and $\hat{\beta}^* := \arg\min_{\beta' \in \Delta^{n-1}} \hat{g}(\beta'; x, \beta)$. Lemma G.4 shows that an approximate stationary point $\hat{\beta}$ of a strongly convex function is close to the exact point $\hat{\beta}^*$:

$$\|\hat{\beta} - \hat{\beta}^*\|_2 \leq \frac{c_1 \varepsilon_0}{\mu_g} =: \delta, \tag{39}$$

where we let $\delta$ denote this constant for short. Since $g(\beta'; x, \beta)$ is a strongly convex function and simplex is a compact convex set, by tilt stability of solution mapping of strongly convex functions (see Proposition 2G.4 of Dontchev and Rockafellar (2009)), we have,

$$\left\|\hat{\beta}^* - \beta^*\right\|_2 \leq \frac{1}{\mu_g} \left\|(\hat{\nabla} f_0(x) - \nabla f_0(x))^\top \hat{\nabla} x^*(x, \beta)\right\|_2 \leq \frac{M_0}{\mu_g} \left\|\hat{\nabla} f_0(x) - \nabla f_0(x)\right\|_2. \tag{40}$$

The last inequality follows by Lemma E.2.

There are two cases, leading to either (1) $O(\varepsilon_0)$-preference stationarity or (2) $O(\varepsilon_0^2)$-constant descent. The two cases depend on the suboptimality of $\beta$.

Case 1: $\|\beta - \beta^*\|_2 < 2\delta$. Here, $\beta$ is fairly close to the optimum $\beta^*$ of the true surrogate. We show that the approximate stationarity of $\hat{\beta}$ with respect to the noisy surrogate implies approximate expected preference stationarity. Similar to Theorem 10, $(\hat{x}, \hat{\beta})$ is expected $(\varepsilon_0, \varepsilon)$-preference stationary provided:

$$\|\nabla f_{\hat{\beta}}(\hat{x})\|_2 \leq \varepsilon_0 \tag{41}$$

$$-\mathbb{E}\left[\nabla f_0(x)^\top \widehat{\nabla} x^*(x, \hat{\beta})(\beta' - \hat{\beta})\right] \leq \mathbb{E}\left[\frac{1}{2}\varepsilon_0 \|\beta' - \hat{\beta}\|_1\right], \qquad \forall \beta' \in \Delta^{n-1} \tag{42}$$

$$\mathbb{E}[\mathrm{err}_{\nabla f_0}(x, \hat{\beta})] \leq \frac{1}{2}\varepsilon_0 \tag{43}$$

Observe that,

$$\|\beta - \hat{\beta}\|_2 \leq \|\beta - \beta^*\|_2 + \|\beta^* - \hat{\beta}^*\|_2 + \|\hat{\beta}^* - \hat{\beta}\|_2 < 3\delta + \frac{M_0}{\mu_g}\left\|\hat{\nabla} f_0(x) - \nabla f_0(x)\right\|_2, \tag{44}$$

combining (39), (40), and the condition that $\|\beta^* - \beta\|_2 < 2\delta$.

Note that (41) follows trivially by Condition 2 of Theorem H.1. One can efficiently find such a $\hat{\beta}$ with $\log(1/\varepsilon_0)$ iteration complexity as it just requires optimizing a strongly-convex objective over a compact convex set.

Now we show (42). We have the following bound.

$$\begin{aligned}
&- \hat{\nabla} f_0(x)^\top \hat{\nabla} x^*(x, \beta)(\beta' - \hat{\beta}) \\
=& - \nabla \hat{g}(\hat{\beta}; x, \beta)(\beta' - \hat{\beta}) + \mu_g \left\langle \hat{\beta} - \beta, \beta' - \hat{\beta} \right\rangle \\
\leq& c_1 \varepsilon_0 \left\|\beta' - \hat{\beta}\right\|_1 + \mu_g \left(\left\|\hat{\beta} - \hat{\beta}^*\right\|_2 + \left\|\hat{\beta}^* - \beta^*\right\|_2 + \|\beta^* - \beta\|_2\right) \left\|\beta' - \hat{\beta}\right\|_1 \\
\leq& \left(c_1 \varepsilon_0 + 3\delta + \frac{M_0}{\mu_g}\left\|\hat{\nabla} f_0(x) - \nabla f_0(x)\right\|_2\right) \left\|\beta' - \hat{\beta}\right\|_1.
\end{aligned} \tag{45}$$

The equality follows by the definition of $\hat{g}$. The first inequality follows by Condition 1 of Theorem 11, triangle inequality, and Holder's inequality. The second inequality follows by (39), (40), and by the condition $\|\beta^* - \beta\|_2 < 2\delta$ in Case 1.

Now, for all $\beta' \in \Delta^{n-1}$ we have,

$$- \nabla f_0(x)^\top \widehat{\nabla} x^*(x, \hat{\beta})(\beta' - \hat{\beta})$$

$$\overset{(i)}{\leq} - \nabla f_0(x)^\top \widehat{\nabla} x^*(x, \beta)(\beta' - \hat{\beta}) + \|\nabla f_0(x)^\top (\widehat{\nabla} x^*(x, \hat{\beta}) - \widehat{\nabla} x^*(x, \beta))\|_\infty \cdot \|\beta' - \hat{\beta}\|_1$$

$$\overset{(ii)}{\leq} - \hat{\nabla} f_0(x)^\top \widehat{\nabla} x^*(x, \beta)(\beta' - \hat{\beta}) + M_0 \left\| \hat{\nabla} f_0(x) - \nabla f_0(x) \right\|_2 \left\| \beta' - \hat{\beta} \right\|_1$$

$$\quad + \|\nabla f_0(x)^\top (\widehat{\nabla} x^*(x, \hat{\beta}) - \widehat{\nabla} x^*(x, \beta))\|_\infty \cdot \|\beta' - \hat{\beta}\|_1$$

$$\overset{(iii)}{\leq} \left( c_1 \varepsilon_0 + 3\delta + \frac{M_0}{\mu_g} \left\| \hat{\nabla} f_0(x) - \nabla f_0(x) \right\|_2 \right) \|\beta' - \hat{\beta}\|_1 + M_0 \left\| \hat{\nabla} f_0(x) - \nabla f_0(x) \right\|_2 \left\| \beta' - \hat{\beta} \right\|_1$$

$$\quad + \|\nabla f_0(x)\|_2 \cdot \frac{L}{\mu^2} \|\beta - \hat{\beta}\|_2 \cdot \|\beta' - \hat{\beta}\|_1$$

$$\overset{(iv)}{\leq} \left( c_1 \varepsilon_0 + \left( 1 + \frac{L}{\mu^2} \|\nabla f_0(x)\|_2 \right) 3\delta + M_0 \left( \frac{1}{\mu_g} + 1 + \frac{L}{\mu^2 \mu_g} \|\nabla f_0(x)\|_2 \right) \left\| \hat{\nabla} f_0(x) - \nabla f_0(x) \right\|_2 \right) \|\beta' - \hat{\beta}\|_1$$

$$(46)$$

where (i) adds and subtracts $\nabla f_0(x)^\top \widehat{\nabla} x^*(x, \beta)(\beta' - \hat{\beta})$ and applies Hölder's inequality, (ii) follows by adding and subtracting $-\hat{\nabla} f_0(x)^\top \widehat{\nabla} x^*(x, \beta)(\beta' - \hat{\beta})$, Lemma E.2, Holder's inequality, and the fact that $\|x\|_\infty \leq \|x\|_2$, (iii) follows by (45), Lemma E.2, and Lemma G.3, and (iv) bounds $\|\beta - \hat{\beta}\|_2$ using (25), and (iv) follows by (44). Taking conditional expectation on both sides, using Lemma 9, and Cauchy-Schwarz inequality, we get,

$$\mathbb{E} \left[ -\nabla f_0(x)^\top \widehat{\nabla} x^*(x, \hat{\beta})(\beta' - \hat{\beta}) \mid x, \beta \right]$$

$$\leq \left( c_1^2 \varepsilon_0^2 + \left( 1 + \frac{L}{\mu^2} \|\nabla f_0(x)\|_2 \right)^2 9\delta^2 + c_3^2 M_0^2 \left( \frac{1}{\mu_g} + 1 + \frac{L}{\mu^2 \mu_g} \|\nabla f_0(x)\|_2 \right)^2 \varepsilon_0^2 \right)^{\frac{1}{2}} (\mathbb{E}[\|\beta' - \hat{\beta}\|_1^2])^{\frac{1}{2}}.$$

Taking expectation on both sides and by the definition of $c_1$, and observing $\left\| \beta' - \hat{\beta} \right\|_1 \leq 2$, we have,

$$\mathbb{E} \left[ -\nabla f_0(x)^\top \widehat{\nabla} x^*(x, \hat{\beta})(\beta' - \hat{\beta}) \right] \leq \frac{1}{2} \varepsilon_0 \mathbb{E}[\|\beta' - \hat{\beta}\|_1].$$

To show (43), we have:

$$\mathrm{err}_{\nabla f_0}(x, \hat{\beta})$$

$$\overset{(i)}{=} \mathrm{err}_{\nabla f_0}(x, \beta) + \frac{1}{\mu} \left( \frac{M_1}{2M_0} \|\nabla f_0(x)\|_2 + L_0 M_0 \right) \left( \|\nabla f_{\hat{\beta}}(x)\|_2 - \|\nabla f_\beta(x)\|_2 \right)$$

$$\overset{(ii)}{\leq} \mathrm{err}_{\nabla f_0}(x, \beta) + \frac{1}{\mu} \left( \frac{M_1}{2M_0} \|\nabla f_0(x)\|_2 + L_0 M_0 \right) \|\nabla F(x)^\top\|_2 \cdot \|\hat{\beta} - \beta\|_2$$

$$\overset{(iii)}{\leq} \frac{1}{2} \cdot \frac{2}{\mu} \left( \frac{M_1}{2M_0} \|\nabla f_0(x)\|_2 + L_0 M_0 \right) \left( c_2 \varepsilon + \left( 3\delta + \frac{M_0}{\mu_g} \left\| \hat{\nabla} f_0(x) - \nabla f_0(x) \right\|_2 \right) \|\nabla F(x)^\top\|_2 \right).$$

$$(47)$$

where (i) expands out $\mathrm{err}_{\nabla f_0}$, (ii) uses the fact that $\beta \mapsto \|\nabla F(x)^\top \beta\|_2$ is $\|\nabla F(x)^\top\|_2$-Lipschitz in $\beta$ with respect to the $\ell_2$-norm, and (iii) applies the definition of $\mathrm{err}_{\nabla f_0}$ and (44). Taking conditional expectation on both sides, and by Lemma 9,

$$\mathbb{E} \left[ \mathrm{err}_{\nabla f_0}(x, \hat{\beta}) \mid x, \beta \right]$$

$$\leq \frac{1}{2} \cdot \frac{2}{\mu} \left( \frac{M_1}{2M_0} \|\nabla f_0(x)\|_2 + L_0 M_0 \right) \left( c_2 \varepsilon + \left( 3\delta + \frac{c_3 M_0}{\mu_g} \varepsilon_0 \right) \|\nabla F(x)^\top\|_2 \right)$$

Taking expectation on both sides, and by definitions of $c_1$ and $c_2$, we have,

$$\mathbb{E} \left[ \mathrm{err}_{\nabla f_0}(x, \hat{\beta}) \right] \leq \frac{1}{2} \varepsilon_0.$$

Case 2: $\|\beta^* - \beta\|_2 \geq 2\delta$. Here $\beta$ is suboptimal and $\beta^*$ achieves a large descent. Akin to (29), we have,

$$f_0(x_{\beta^*}) - f_0(x_\beta) \leq -\frac{3}{2}\mu_g\delta^2. \tag{48}$$

Now we have,

$$f_0(x_{\hat{\beta}}) - f_0(x_\beta)$$

$$\overset{(i)}{\leq} g(\hat{\beta}; x, \beta) - f_0(x_\beta)$$

$$= g(\beta^*; x, \beta) - f_0(x_\beta) + \left(g(\hat{\beta}; x, \beta) - g(\hat{\beta}^*; x, \beta)\right) + \left(g(\hat{\beta}^*; x, \beta) - g(\beta^*; x, \beta)\right)$$

$$\overset{(ii)}{\leq} -\frac{3}{2}\mu_g\delta^2 + \nabla g(\hat{\beta}; x, \beta)^\top(\hat{\beta} - \hat{\beta}^*) + \frac{\mu_g}{2}\left\|\hat{\beta}^* - \beta^*\right\|_2^2$$

$$\overset{(iii)}{\leq} -\frac{3}{2}\mu_g\delta^2 - \nabla\hat{g}(\hat{\beta}; x, \beta)^\top(\hat{\beta}^* - \hat{\beta}) + \left(\hat{\nabla}f_0(x) - \nabla f_0(x)\right)^\top \hat{\nabla}x^*(x, \beta)^\top(\hat{\beta} - \hat{\beta}^*)$$

$$\quad + \frac{\mu_g}{2}\left\|\hat{\beta}^* - \beta^*\right\|_2^2$$

$$\overset{(iv)}{\leq} -\frac{3}{2}\mu_g\delta^2 + c_1\varepsilon_0\left\|\hat{\beta} - \hat{\beta}^*\right\|_2 + \left\|\hat{\nabla}f_0(x) - \nabla f_0(x)\right\|_2\left\|\hat{\nabla}x^*(x, \beta)\right\|_2\left\|\hat{\beta} - \hat{\beta}^*\right\|_2$$

$$\quad + \frac{\mu_g}{2}\left\|\hat{\beta}^* - \beta^*\right\|_2^2$$

$$\overset{(v)}{\leq} -\frac{3}{2}\mu_g\delta^2 + c_1\varepsilon_0\delta + \frac{M_0\delta}{\mu_g}\left\|\hat{\nabla}f_0(x) - \nabla f_0(x)\right\|_2 + \frac{M_0^2}{2\mu_g}\left\|\hat{\nabla}f_0(x) - \nabla f_0(x)\right\|_2^2.$$

where (i) uses the majorizing property of $g$, (ii) applies (48), and strong convexity of $g(\cdot; x, \beta)$, (iii) follows from the definition of $\hat{g}(\cdot; x, \beta)$, (iv) follows from Hölder's inequality, and choice of $c_1$, (v) follows by (39), (40), and Lemma E.2.

Taking conditional expectation on both sides, using Lemma 9

$$\mathbb{E}[f_0(x_{\hat{\beta}}) - f_0(x_\beta) \mid x, \beta] \leq -\frac{3}{2}\mu_g\delta^2 + c_1\varepsilon_0\delta + \frac{c_3M_0\delta\varepsilon_0}{\mu_g} + \frac{c_3^2M_0^2\varepsilon_0^2}{2\mu_g}.$$

Taking expectation on both sides,

$$\mathbb{E}[f_0(x_{\hat{\beta}}) - f_0(x_\beta)] \leq -\frac{3}{2}\mu_g\delta^2 + c_1\varepsilon_0\delta + \frac{c_3M_0\delta\varepsilon_0}{\mu_g} + \frac{c_3^2M_0^2\varepsilon_0^2}{2\mu_g}.$$

Observe that, choosing parameters properly in Lemma 9, we can always ensure

$$c_3 \leq \min(\mu_g c_1/(8M_0), c_1/(12M_0)).$$

Then, we have,

$$\mathbb{E}[f_0(x_{\hat{\beta}}) - f_0(x_\beta)] \leq -\frac{1}{4}\mu_g\delta^2.$$

Thus, in expectation, the preference improves by at least a constant. The choices of $c_1$ and $c_2$ follows similarly as in Theorem 10.

$\blacksquare$

# I Details to Visualization of Toy Examples

Figure 2 visualizes learning dynamics for two different Pareto-constrained optimization problems. The code can be found at https://github.com/geelon/preference-pareto.

**Preference optimization on a non-smooth Pareto set**   The left sub-figure is an example of optimization on the Pareto set described in Example 1. In particular, the preference function is:

$$f_0(x) = \frac{1}{2}(x - v_0)^\top \begin{bmatrix} 1 & 0.35 \\ 0.35 & 2 \end{bmatrix} (x - v_0) \qquad \text{and} \qquad v_0 = \begin{bmatrix} 0.2 \\ 0.75 \end{bmatrix}.$$

Hyperparameters used using the PNG algorithm by Ye and Liu (2022) include the Pareto-set approximation threshold $e = 0.001$, learning rate $\xi = 0.01$, and regularization parameter $\alpha_t = 0.01$. These were not particularly tuned. The two different phases in the algorithm can be clearly observed in the dynamics. In particular, when the iterates are close to the Pareto set, the PNG dynamics minimizes $f_0$. Otherwise, it aims to move toward the Pareto set in a direction that also optimizes the preference. This leads to the jagged appearance of the PNG dynamics. We ran 40k iterations.

The PMM algorithm visualize in this example alternates between updating $x_k$ and $\beta_k$. Each update step for $\beta_k$ is implemented by an approximate gradient descent, making use of $\hat{\nabla} x^*$, followed by a projection back onto the simplex. The learning rate for $\beta$ chosen here was $0.1$. Then, the update step for $x_k$ consists of a single step of gradient descent on $f_{\beta_k}$ with learning rate of $0.1$. A total of 1500 iterations was run. These hyperparameters were not specifically tuned.

**Preference optimization where first-order information is insufficient**   The right sub-figure visualizes the counterexample given in Example D.1.

The PNG algorithm in this example used parameters $e = 0.001$, $\xi = 0.1$, and $\alpha = 0.01$. We ran the algorithm for 26k iterations. The PMM algorithm in this example is the same as the previous example. The learning rate chosen here was $0.1$. The inner loop used 1 step of gradient descent with learning rate of $0.1$. We ran 1500 iterations and did not attempt to tune hyperparameters.

