# OpenReview forum: "Preference Optimization on Pareto Sets: On a Theory of Multi-Objective Optimization"
_NeurIPS.cc/2025/Conference — NeurIPS 2025 poster_

### Official Review · Reviewer_YrFB · 2025-07-03

**Clarity:** 2
**Significance:** 3
**Originality:** 3
**Rating:** 4
**Confidence:** 4

**Summary:**

This paper proposes a method to tackle the optimization on the Pareto set problem, with application to preference optimization. Specifically, the upper-level objective is smooth and possibly nonconvex, while the lower-level objectives are smooth and strongly convex. An algorithm based on majorization-minimization is proposed to solve this problem.  Theoretical analysis on the convergence of the proposed algorithm, and the Bias and Variance bounds of Dueling Gradient Estimator for preference optimization are provided.

**Questions:**

1. Section 6.3.

- Many notations are not introduced in this section.
For example, what are $u_i, d, b, \gamma$?

- Equation (16) needs more explanation.
  - What is the intuition for this estimate?
  - What are the required assumptions for the preference model and the preference samples? e.g., the underlying distributions that generated these samples?
  - How to choose different link functions in practice?

2. Assumption D.
Why is the Lipschitz constant of $\sigma^{-1}$ depending on $x$?
If it depends on $x$, it cannot be called "Lipschitz".

3. How is the operator $\widehat{\arg\min}$ defined? It is used in the algorithm and Theorem 9 but not explicitly defined. What is the black-box optimizer?

**Ethical Concerns:**

["NO or VERY MINOR ethics concerns only"]

**Final Justification:**

The authors actively engaged in the discussion and promised to make some important revisions. I will raise my score to 4 if those revisions are included.

**Limitations:**

1. The theory and the proposed method only apply to strongly convex lower-level objectives.
2. The proposed method may require second-order information, can be computationally very expensive.

**Paper Formatting Concerns:**

No.

**Quality:**

2

**Strengths And Weaknesses:**

1. Missing references.

There are many prior works that have already tackled the optimization on the Pareto set, or semivector bilevel optimization problem. See a few examples listed below. These works should be discussed and compared at least in the related works. Among these works, [1] is very close to the method proposed in this paper.

[1] A. Roy, G. So, Y. Ma. Optimization on Pareto sets: On a theory of multi-objective optimization, arXiv:2308.02145

[2] T. Giovannelli, G. D. Kent, and L. N. Vicente. Bilevel optimization with a multi-objective lowerlevel problem: Risk-neutral and risk-averse formulations. Optimization Methods and Software, 2024.

[3] L. Chen, Q. Xiao, E. H. Fukuda, X. Chen, K. Yuan, T. Chen. Efficient First-Order Optimization on the Pareto Set for Multi-Objective Learning under Preference Guidance, arXiv:2504.02854


2. Computational complexity is not discussed sufficiently.

The paper should provide more discussion on the iteration complexity and convergence rate of the proposed algorithm, and compare them with those of the existing algorithms.

3. No experiments are provided to test the effectiveness/efficiency of the proposed method/algorithm and to compare with the existing methods.


4. Some parts lack clarity.

- Section 6.3. See **Questions-1**.

- Assumption D. See **Questions-2**.

- Algorithm. See **Questions-3**.

---

> ### Author Rebuttal · Authors · 2025-07-31
>
> Thank you for your careful review and detailed feedback. We will especially take into account where it seems that the writing was unclear and improve our exposition. Besides that, there were two consistent concerns: (1) relationship/comparison to prior work, (2) practicality. We first address those now, before answering specific questions below.
>
> ## Response to main concerns
>
> ### Relationship to prior work/comparison with other methods
>
> Thank you for sharing the related works. We are indeed familiar with [1], but it turns out that a comparison with [1] is not very meaningful. While [2] does consider a similar setting to us, there are no convergence guarantees provided for the proposed algorithms. [3] follows up on [1] and is also quite interesting; it was concurrent with our NeurIPS submission, but we will update our related works going forward. Besides these, the most related work to our setting is the PNG algorithm (Ye and Liu 2022), against which we theoretically compare in Appendix D.1.
>
> Out of all of these references prior to our work, there were only heuristics for this problem. The literature lacked a proper definition of stationarity and algorithms with provable behavior. Actually, intuitive heuristics easily go wrong. We proved that there is a large class of very natural stationary conditions—including the one found by the PNG algorithm—that will *reject* the true preference-optimal solution. This means that the prior algorithm in literature *provably avoids* the correct solution, even in the strongly convex setting!
>
> In contrast, our work is the first to establish a well-justified solution concept that is efficiently and provably achieved by a simple algorithm, under two forms of feedback. Theoretically, there is nothing to compare against. And since we proved that the closest prior algorithm in literature converges to the wrong solution, it is also not meaningful to compare against them experimentally.
>
> ### Practicality: strongly convex objectives and second-order information
>
> We agree that many practical problems have objectives that are not strongly convex. Indeed, such problems, such as LLM fine-tuning, serves as a key motivation for our framework. But, in developing a principled approach, we found the problem to be highly non-trivial due to the non-convexity, non-smoothness, and the implicit nature of the constraint set (even when the objectives are simply quadratics). Here, we have found that even the strongly convex setting is already theoretically very rich, and that it has helped us to significantly deepen our understanding of the nature and challenges of multi-objective optimization.
>
> And so while our main contributions are theoretical, we believe they are still of significance to the multi-objective optimization community, as we have greatly simplified the problem conceptually. We believe this may help researchers avoid the same pitfall as the PNG algorithm (which we also initially fell into). Additionally, we have already shared this work with part of the community (non-archival), and know of other groups that are developing further results on top of this.
>
> Specifically regarding strong convexity, we are already working on extending these results to the non-convex setting, and we’ve found it to be quite interesting and challenging (see also our response to Reviewer 9L4B). And on second-order information, we have shown that second-order information is necessary. However, we are hopeful that there are computationally-efficient estimators (e.g. using ideas from [I,J]), and we are currently working on this as well.
>
> The completion of either of these results would arguably be substantial enough to be published in separate papers; this paper is already packed with many results. We believe that our work already makes a significant and meaningful contribution to the literature, and that it provides a solid theoretical foundation upon which to develop algorithms that work under less restrictive conditions, and that are more sample and computationally efficient.
>
> Of course, we welcome differences of opinions from the reviewer, but if so, we would really appreciate it if the reviewer would also articulate more specifically what results would be needed to make this work ready for publication.
>
> ## Specific questions
>
> ### Computation complexity
>
> The iteration complexity and the convergence rate of our algorithm, $K \leq \frac{2 \mu_g\cdot \big(f^* - f_*\big)}{ c_1^2 \cdot \epsilon_0^2}$, are provided for the first-order feedback in Theorem 9 (page 8) and for the dueling feedback in Theorem 10 (page 9). The proofs of the theorems are provided in Appendix G and Appendix H.2 respectively.
>
> ### Clarification for Section 6.3
>
> **What are $u_i,d,b,\gamma$?**
>
> In (16), {$u_i$}$_{i=1}^b$ are $b$ unit vectors chosen uniformly at random from the unit sphere $S^{d-1}$ where $d$ is the ambient dimension of the space - same as introduced in Section 2. $\gamma>0$ is a scalar. We will update this notations in the final version.
>
> **Equation (16).**
>
> Under zeroth-order feedback, i.e., when only $f_0(x)$ can be computed, one has the following gradient estimator:
>
> $$\tilde{\nabla} f_0(x)=\frac{d(f_0(x+\gamma u)-f_0(x-\gamma u))}{2\gamma}u,$$
>
> where $u\in S^{d-1}$ is chosen uniformly at random. Then, $\|\mathbb{E}[\tilde{\nabla} f_0(x)]-\nabla f_0(x)\|_2=O(d\gamma)$, which is small for small enough $\gamma$.
>
> So estimating $ \nabla f_0(x) $ only requires estimating function value differences which is implicitly encoded in the preference model. If $\mathbb{E}[Y(x+\gamma u, x-\gamma u)]$ were known, one has
>
> $$f_0(x+\gamma u)-f_0(x-\gamma u)=\sigma^{-1}(\mathbb{E}[Y(x+\gamma u,x-\gamma u)]).$$
>
> Estimating $\nabla f_0(x)$ reduces to estimating $\mathbb{E}[Y(x+\gamma u, x-\gamma u)]$, the mean of a Bernoulli random variable. A natural estimator is the empirical average
>
> $$\tilde{p} = \frac{1}{m} \sum_{j=1}^m q_j(x,u), \quad \text{where } q_j(x,u) = Y_j(x+\gamma u, x-\gamma u),$$
>
> interpretable as averaging the feedback from querying $m$ independent individuals for preferences between two points. The caveat is that for common preference models, $\sigma^{-1}(x)$ becomes unbounded as we approach $x = 0,1$ (e.g., Bradley–Terry with $ \sigma^{-1}(x) = \log(x/(1-x))$). So we apply clipping:
>
> $$\hat{p} = \max(\alpha, \min(\tilde{p}_{x+\gamma u, x-\gamma u}, 1 - \alpha)).$$
>
> Here, $\alpha$ is chosen so that clipping error is balanced with the approximation error in $\mathbb{E}[\tilde{\nabla} f_0(x)]$ and the stochastic error $\text{err}_{\nabla f_0}$. This leads to the following estimator:
>
> $$\tilde{\nabla} f_0(x) = \frac{d}{2\gamma} \sigma^{-1}\left( \hat{p}_{x+\gamma u, x-\gamma u} \right) u.$$
>
> Now, to approximate $\mathbb{E}[\tilde{\nabla} f_0(x)] \approx \nabla f_0(x)$ (with error $O(d\gamma)$ ), we average over $b$ i.i.d. directions {$u_i$}$_{i=1}^b$, and arrive at (16):
>
> $$\hat{\nabla} f_0(x) = \frac{d}{2\gamma b} \sum_{i=1}^b \sigma^{-1}\left( \hat{p}_{x+\gamma u_i, x-\gamma u_i} \right) u_i.$$
>
> We plan to improve the intuition in the final version.
>
> **Assumption D.**
>
> Our preference model (Definition 5) assumes that the binary variable $Y(x_1, x_2)$ satisfies $\mathbb{E}[Y(x_1, x_2)] = \sigma(f_0(x_2) - f_0(x_1))$. The query points $x_1, x_2$ are non-random, so no distributional assumption is imposed on them. Characterizing the preference model reduces to specifying assumptions on $\sigma$, as in Assumption D.
>
> **Choice of link function**
>
> The choice of preference function is problem-specific. Our goal, however, is not to model preferences but to provide an algorithm for Pareto-constrained optimization which can be used as black-box by an user with any preference model (Definition 5) they may choose. As long as Assumption D holds—which is true for common models like logistic, softplus, and tanh—our algorithm converges at the rate stated in Theorem 10, ensuring broad applicability. However, some common preference models include Logistic (Bradley-Terry)/Softmax [A,B,C,D,E] and Tanh [F,G,H].
>
>
> **Lipschitz constant**
>
> By $L_\sigma (1 + (x(1 - x))^{-1})$-Lipschitz continuity, we mean that $\sigma^{-1}$ is locally Lipschitz, but we agree that this can be made less confusing. More precisely, we assume $\left| \frac{d}{dx} \sigma^{-1}(x) \right| \leq C(1 + (x(1 - x))^{-1})$ for some constant $C > 0$, which implies via the mean value theorem:
>
> $$|\sigma^{-1}(x) - \sigma^{-1}(y)| \leq L_\sigma \left(1 + \frac{1}{x(1 - x)} + \frac{1}{y(1 - y)}\right) |x - y|$$
>
> for some $L_\sigma > 0$. This is the bound used in Lemma 1 (Appendix H), and we will revise the terminology in the final version.
>
> **Definition of $\widehat{\mathrm{argmin}}$**
>
> The operator $\widehat{\arg\min}$ denotes the approximate minimizer from the black-box optimizer referenced in Theorem 9. As discussed in Section 6.1 and Remark 2, this can be any suitable off-the-shelf optimizer suitable, including (projected) gradient descent. We will clarify this in the paper.
>
> [A]Maystre et.al, "Choicerank: identifying preferences from node traffic in networks." ICML 2017.
> [B]Chu et.al, "Preference learning with Gaussian processes." ICML 2005.
> [C]Hunter, "MM algorithms for generalized Bradley-Terry models." AOS 2004.
> [D]Chen et.al, "Pairwise ranking aggregation in a crowdsourced setting." ICWDM 2013.
> [E]Ouyang et.al, "Training language models to follow instructions with human feedback." NeurIPS 2022.
> [F]Orlando, "A discrete mathematical model for chaotic dynamics in economics: Kaldor’s model on business cycle." MCS 2016.
> [G]Tuev et.al, "Using modified hyperbolic tangent function to approximate the current–voltage characteristics of field–effect transistors." Tomsk Polytechnic University, 2009.
> [H]Giulio. "Power Exhaust Data Analysis and Modeling of Advanced Divertor Configurations." 2018.
> [I]Huang et.al, "Efficiently escaping saddle points in bilevel optimization." JMLR 2025.
> [J]Kwon et.al, "A fully first-order method for stochastic bilevel optimization." ICML 2023.

---

> > ### Comment · Reviewer_YrFB · 2025-08-08
> >
> > Thanks for the rebuttal. Some of my concerns have been addressed. Overall, I appreciate the theoretical studies in the paper. However, I still have remaining major concerns detailed below.
> >
> > **Comparison with related works.** It would be better if the authors could provide a detailed discussion and comparison with the most relevant works in the rebuttal and include them in the revision. The current rebuttal did not directly address my questions and concerns.
> >
> > - Regarding the comparison with the related works [1,2,3] in my review, it seems that this work is very close to [1], but with an additional section discussing the application to preference optimization. [2] addresses a semivectorial bilevel optimization problem with lower-level strictly convex functions, [3] addresses the optimization on the Pareto set problem with lower-level non-convex functions. Both [2] and [3] require weaker assumptions. The authors should at least ```discuss these differences in the rebuttal and include them in the revision```. It would also be good to check whether there are additional missing related works.
> >
> > - I respectively disagree with the claim that "Out of all of these references prior to our work, there were only heuristics for this problem. The literature lacked a proper definition of stationarity and algorithms with provable behavior." In fact, ```in the semivectorial bilevel optimization literature, which covers the optimization on the Pareto set problem, there are already many works addressing the definition of necessary optimality conditions under the same setting where the lower-level objectives are strongly convex```. Many of these works also consider reformulating the lower-level problem through linear scalarization, equivalent to "lifting to Pareto manifold" in this paper. Those works should also be included and compared. See a few listed below.
> >
> > [a] Optimality conditions for minimization over the (weakly or properly) efficient set, Bolintineanu, 1993.
> >
> > [b] Necessary conditions for nonlinear suboptimization over the weakly-efficient set, Bolintineanu, 1993.
> >
> > [c] Optimization over the efficient set of multi-objective convex optimal control problems, Bonnel et al., 2010.
> >
> > [d] Semivectorial bilevel programming versus scalar bilevel programming, Dempe et al., 2019.
> >
> >
> > - I find the answer "Since we proved that the closest prior algorithm in literature converges to the wrong solution, it is also not meaningful to compare against them experimentally." not satisfactory at all. To be more specific, the results I expect are ```at least some toy experiments with lower-level strongly convex objectives, to show your proposed algorithm can correctly converge to a meaningful stationary solution, while PNG cannot```, which further verifies your theoretical results. I believe this is basic for publication in NeurIPS, and can strengthen your paper.
> >
> > **No experimental results or implementation details provided.** This paper studies the optimization on the Pareto set problem, and proposes a new algorithm. For an optimization algorithm paper, typically at least toy experiments are performed and convergence curves are reported to justify the effectiveness of the proposed algorithm. Furthermore, the paper has a dedicated section with an application to preference optimization, but with ```many missing definitions and implementation details``` in the submitted version, as pointed out in my review. Without those details or doing actual experiments using the proposed algorithm, it is hard for others to follow the work and implement the proposed algorithm. Conducting experiments and reporting at least the convergence curve and hyperparameter choices helps the readers better understand how to implement the algorithm, and how the algorithm works in practice.
> >
> >
> >
> >
> > I have made my reviews and suggestions more detailed and specific now. If the authors can properly address these concerns and revise the paper accordingly, I will raise my final score.

---

> ### Author Response · Authors · 2025-08-06
>
> Dear Reviewer,
>
> Since the discussion period is nearing the end, we wanted to make sure that our response fully addresses your questions. If you have any further questions/concerns we would love to address that. If we have answered your questions satisfactorily, please let us know.
>
> Best,
> Authors

---

> ### Author Response · Authors · 2025-08-09
>
> Thank you for taking the time to write this detailed review and response. Regardless of outcome, we really appreciate how sincerely you have approached reviewing.
>
> **Prior work.** Yes, we are quite aware that, except for the section on preference feedback, the results we present are extremely close to [1]. But to reiterate, it is not meaningful to compare ourselves to [1]. We cannot be more specific, but we can reassure the reviewer that the contents of our submission has never been published before at a conference/journal. And that prior to our work, there was no algorithm to our knowledge that obtained provable convergence. As we mentioned before, [3] was concurrent with our NeurIPS submission, following up on [1].
>
> Thank you for sharing those additional references. Unfortunately, we aren’t able to provide a proper comparison here, since there isn’t enough time between receiving your comments and the end of the discussion period. But, we agree that it is definitely worthwhile to update our related work section, and we will do so.
>
> Having quickly looked through these additional references, it does not seem that our claim that we provide the first algorithm with provable convergence is wrong. It will take more time to understand the relationship between prior notions of stationary conditions, and we will be careful not to overclaim in the paper (perhaps the claim in the rebuttal was too broad). In any case, we do believe that our definition of *approximate* preference stationarity (Definition 5) still makes a meaningful contribution: Proposition 6 shows that it has provable optimization-theoretic meaning, and Lemma 7 shows that it can be verified using approximate information (one does not need to exactly solve the lower level optimization problem to know whether it has been achieved). This latter point is important in terms of “usability” of the solution concept. Another aspect of our contribution regarding the solution concept is that we showed the insufficiency of first-order information, helping to explain why second-order terms appear in this notion of stationarity.
>
> **Experiments.** Regarding experiments, we will include them in revisions (and other reviewers also asked for this). We have run toy experiments (and they reflected the behavior we proved: PNG converges to the wrong solution). But we agree that it is more convincing and helpful to visualize them, and that these experiments can help readers obtain an intuition about what our algorithm is doing.
>
> **Preference feedback section.** Again, thank you for pointing out where we needed to improve the presentation. Hopefully, our rebuttal above helped clarify those definitions for you here? We will make that section more readable and make sure everything is clearly defined in the revisions.
>
> &nbsp;
>
> **Summary:** At a high level, it seems that there is no issue concerning the importance of the problem, theoretical soundness, nor completeness of results. The main concerns are (a) whether our work is novel, (b) connection to related work, (c) providing experiments, and (d) improving clarity for the preference section.
>
> We do believe our work (a) makes new and fundamental contributions to multi-objective optimization, and are aware of the works the reviewer has cited, and that other groups are already developing on top of our work. For the remaining critiques (b,c,d), we completely agree: we will iron out the specific relationship to past works/ideas, include experiments, and improve clarity. Since there isn’t a need to develop new results, these are all definitely doable by the deadline for final revisions. We would love to be able to publish our results sooner than later, and not wait for the next publication cycle. If this is adequate, we would really appreciate a positive score. In any case, thank you again for the time for giving us this feedback.

---

> > ### Comment · Reviewer_YrFB · 2025-08-09
> >
> > Thanks for the response. I hope the authors will make the promised revisions, and I will update my score accordingly.

---

> > > ### Author Response · Authors · 2025-08-09
> > >
> > > We will, and thank you.

---

### Official Review · Reviewer_i8uk · 2025-07-03

**Clarity:** 3
**Significance:** 2
**Originality:** 3
**Rating:** 4
**Confidence:** 2

**Summary:**

The paper considers Pareto-constrained optimization, or constrained optimization of a user-defined preference function over Pareto-optimal solutions. It addresses the challenges of the non-smooth, non-convex Pareto set by lifting the problem to a "Pareto manifold" diffeomorphic to the $n-1$ simplex. The authors define a "preference stationarity" condition and relax it to a $(\epsilon_0, \epsilon)$-approximate version for practical optimization algorithms. The main algorithmic contribution is the Pareto Majorization-Minimization (PMM) algorithm, a two-step method that alternates between fitting a quadratic surrogate that upper-bounds $f_0$ and minimizing it over the simplex. PMM can accommodate gradient-based and dueling-based feedback and converges to an $\epsilon_0$-approximate preference stationary solution in $O(\epsilon_0^{-2})$ iterations.

**Questions:**

- Assumption A requires each objective $f_i$ to be strongly convex, which may not apply in many practical MOO settings. How can the Pareto-manifold lift be extended to just smooth (or even non smooth) objectives? What would be the technical challenges? So far as the paper is claiming relevance to RLHF, can the authors empirically demonstrate robustness to violating these assumptions?
- Can the authors suggest some practical heuristics for determining when an inner solve is "good enough?"
- The sample complexity for the dueling-based feedback scales poorly with dimension and $\epsilon_0$. Can the authors include some experiments or suggest (e.g., variance reduction) methods to lower the sensitivity to dimension or $\epsilon_0$?

**Ethical Concerns:**

["NO or VERY MINOR ethics concerns only"]

**Final Justification:**

My concerns regarding related work and reframing (as a theory paper not directly validated on LLMs, as previously suggested throughout the text) have been addressed.

**Limitations:**

The limitations are mentioned throughout the text (e.g., Remark 3), but it would be helpful to include a separate limitations section.

**Quality:**

3

**Strengths And Weaknesses:**

### Strengths
- While dense, the paper is well written and the narrative clear through the definition of the lifted Pareto manifold. The "weight" parameterization is intuitive and illustrated effectively in Fig 1
- The preference stationarity condition is shown to be provably necessary for preference optimality and improves on earlier notions of stationarity that could miss optimal decisions
- The PMM algorithm has an $O(\epsilon_0^{-2})$ last-iterate convergence rate for both gradient-based and dueling-based feedback
- Approximate proxy using only the local Hessian/gradient information is enough to verify approximate stationarity, by Lemma 7

### Weaknesses
- The framework assumes that all objectives are $\mu$-strongly convex with Lipschitz Hessians (Assumption A), which may rule out more complex real-world MOO tasks.
- The dueling-based feedback scales poorly with dimension and $\epsilon_0$, as $O(d^4/\epsilon_0^4 \log (d/\epsilon_0))$
- Solving the inner problem every iteration may be resource-expensive for large models.
- The paper is purely theoretical. Experiments on real RLHF or MOO benchmark problems would help validate constants or verify robustness to assumption violations, particularly Assumption A
- The related work section seems to omit existing work on multi-objective stein variational gradient descent (e.g., [1]) and multi-objective Bayesian optimization, which similarly uses a Bayesian surrogate (e.g., [3-6]). In particular, [4] considers preference provided in the form of preference weights and [6] uses copulas and a game-theoretic argument for the preferred solution.

[1] Liu, Xingchao, Xin Tong, and Qiang Liu. "Profiling pareto front with multi-objective stein variational gradient descent." 2021.

[2] Belakaria, Syrine, et al. "Uncertainty-aware search framework for multi-objective Bayesian optimization." 2020.

[3] Knowles, Joshua. "ParEGO: A hybrid algorithm with on-line landscape approximation for expensive multiobjective optimization problems." (2006).

[4] Paria, Biswajit, Kirthevasan Kandasamy, and Barnabás Póczos. "A flexible framework for multi-objective bayesian optimization using random scalarizations." 2020.

[5] Emmerich, M., and Klinkenberg. The computation of the expected improvement in dominated hypervolume of pareto front approximations. 2008.

[6] Binois, Mickaël, et al. "The Kalai-Smorodinsky solution for many-objective Bayesian optimization." 2020.

---

> ### Author Rebuttal · Authors · 2025-07-30
>
> Thank you to Reviewer i8uk for taking the time to review and for the detailed feedback. It seems that the reviewer found the theoretical contributions strong, but raises concerns about the applicability to practical multi-objective settings such as for LLM fine-tuning, specifically because of (a) strong convexity assumptions, (b) how sample complexity scales with different parameters of the problem, and (c) the computational costs of solving the inner optimization problem.
>
>
> **Main contributions: theoretical vs. practical**
>
> We agree that this work’s main contributions are theoretical, although practical problems such as LLM fine-tuning serves as a key motivation for our framework. But, in developing a principled approach, we found the problem to be highly non-trivial due to the non-convexity, non-smoothness, and the implicit nature of the constraint set (even when the objectives are simply quadratics). We greatly simplify the problem conceptually.
>
> Prior to our work, there were only heuristics for this problem. The literature lacked a proper definition of stationarity and algorithms with provable behavior. Actually, intuitive heuristics easily go wrong. We proved that there is a large class of very natural stationary conditions—including one in existing literature, and another we had initially considered—that will *reject* the true preference-optimal solution. Practically, this means that the prior algorithm in literature *provably avoids* the correct solution, even in the strongly convex setting!
>
> In contrast, our work is the first to establish a well-justified solution concept that is efficiently and provably achieved by a simple algorithm, under two forms of feedback. The issue of practicality, especially in the context of LLM fine-tuning, is certainly a valid one, but also in a sense beyond the scope of this work: each of the goals of going beyond non-convexity, significantly improving sample complexity, reducing computational complexity are quite non-trivial and can be papers in themselves (indeed, we are currently pursuing these directions, and know of other groups doing so). We believe that our work already makes a significant and meaningful contribution to the literature, and that it provides a solid theoretical foundation upon which to develop algorithms achieving the above goals. As the reviewer puts it, this work already presents quite a "dense" set of results.
>
> Of course, we welcome differences of opinions from the reviewer, but if so, we would really appreciate it if the reviewer would also articulate more specifically what remaining results would make this work ready for publication.
>
> In the following, we respond to more specific points.
>
>
> **Generalization to non-convex objectives**
>
> Extending to the non-convex setting is definitely an important future direction, and one we’ve found to be quite interesting and challenging. Some challenges:
>
> 1. Without strict convexity, linear scalarization can miss out on Pareto optimal solutions: not every Pareto solution $x^*$ is a (local) minimizer of $f_\beta$ for some convex combination $\beta \in \Delta^{n-1}$ of the objectives.
>
> 2. In fact, the Pareto optimal points can unintuitively even correspond to local maximizers of $f_\beta$! This can happen even if the objectives are quasiconvex (thus, not non-convex in an arbitrary way; an example below).
>
> 3. And if the objectives are arbitrarily non-convex, we also lose the diffeomorphism between the Pareto manifold and the simplex (though local diffeomorphisms may remain).
>
> We have some ideas and are pursuing this direction, but it would probably be better to present results there in a separate paper.
>
> *Example of Pareto optimal point that is a local maximizer*. Let $q_1, q_2 : \mathbb{R} \to \mathbb{R}$ be the quadratics $q_1(x) = (x - 1)^2$ and $q_2 = (x + 1)^2$. And let us aim to minimize the objectives $f_1, f_2 : \mathbb{R} \to \mathbb{R}$, where $f_i = \sigma \circ q_i$, and $\sigma$ is any strictly monotonic increasing function. Then, the Pareto set of $(f_1, f_2)$ coincides with the Pareto set of $(q_1, q_2)$, which is the interval $[-1, 1]$.
>
> Consider $\sigma(z) = \exp(z/T)$, where $T > 0$ is some temperature parameter. For sufficiently small $T$, the Pareto optimal solution $x = 0$ will be the local *maximizer* of the weighted objective $\frac{1}{2} f_1 + \frac{1}{2} f_2$.
>
>
> **Computational cost of inner solve and practical heuristics**
>
> At least under our setting of strong convexity and Lipschitz smoothness, the cost of the inner solve is dimension-independent, with linear convergence rate. Moreover, our convergence analysis for the PMM algorithm quantifies how accurately the inner solve needs to be to achieve a target approximation error. In particular, to obtain $\epsilon$-approximate preference stationarity, Theorem 9 shows that it is sufficient to optimize the inner objective $f_\beta$ to approximate stationarity $\| \nabla f_\beta(\hat{x})\| < O(\epsilon)$. This takes around $O(\log \frac{1}{\epsilon})$ steps of gradient descent.
>
>
> **Improving sample efficiency in dueling feedback setting**
>
> We agree that a very practically-motivated problem is how to increase sample complexity of preference optimization under dueling feedback. Currently, we do not know how suboptimal the dependence on dimension is, nor on other parameters such as $\epsilon_0$. It would be of great interest to explore how to reduce these dependencies or to show that in fact they are necessary. Still, this work is the first to obtain meaningful convergence guarantees under either forms of feedback.
>
>
> **Relationship to prior works**
>
> Thank you for sharing these references to other works; we can contextualize our work among them. Classically speaking, methods in multi-objective optimization can be categorized as *a priori*, *a posteriori*, and *interactive*.
>
> In *a priori* methods, the decision maker knows beforehand what their preference is, and the role of the MOO algorithm is to find the most-preferred Pareto-optimal solution. The bulk of our work falls under this category, with the exception of the section on dueling feedback, which is more interactive. The reference [6] is of this type, but their focus is on introducing one specific preference function motivated from game theory.
>
> In *a posteriori* methods, the decision maker needs to first learn the Pareto set before deciding on their preference/making the final decision. In this case, the role of the MOO algorithm is to approximate the whole Pareto set. That is, to provide a shortlist of Pareto optimal options that would satisfy a wide variety of preferences of a downstream decision maker. The references [1,2,3,5] listed above fall under this category. (We do not operate under this setting).
>
> And *interactive* methods are in between, where the decision maker repeatedly interacts with the MOO algorithm, where both can refine their search and preferences over time. Our section on dueling feedback is of this form, and so is the reference [4] provided. However, the goals are somewhat different. We aim to recover the most preferred solution, while [4] aims to sample a diverse set of solutions from the Pareto set related to a user-specified distribution/prior over solutions.

---

> > ### Comment · Reviewer_i8uk · 2025-08-02
> >
> > Thank you for addressing my comments. The positioning wrt different categories of prior work is very helpful. It seems that the concerns about practical applicability -- particularly in what settings the assumptions would hold true -- were shared by other reviewers. These expectations may have been set by the references to LLM alignment as "motivating examples." I would suggest removing these references and instead including a discussion of concrete examples (e.g., in portfolio optimization) that may be more amenable to the assumptions made, so that we can focus on the theory contributions and refrain from overemphasizing application-readiness. As this seems to be a matter of editing the narrative, I have raised my score.

---

> > > ### Author Response · Authors · 2025-08-06
> > >
> > > Thank you so much. We appreciate it.
> > >
> > > - Authors

---

> > > ### Author Response · Authors · 2025-08-07
> > >
> > > Dear Reviewer,
> > >
> > > We really appreciate your decision to raise our score. Would you mind double checking the form? It just seems that elsewhere when the score has been modified, it becomes hidden. Here, we seem to see the previous score of 3.
> > >
> > > Thanks again!
> > > Sincerely,
> > > Authors

---

> > > > ### Comment · Reviewer_i8uk · 2025-08-07
> > > >
> > > > I believe the final scores are hidden from the authors until decision release. I will raise my score to a 4.

---

> > > > > ### Author Response · Authors · 2025-08-07
> > > > >
> > > > > We appreciate it, thank you!

---

### Official Review · Reviewer_9L4B · 2025-07-03

**Clarity:** 3
**Significance:** 3
**Originality:** 3
**Rating:** 5
**Confidence:** 3

**Summary:**

This paper introduced the Pareto manifold which is a ‘lifting’ of the Pareto set. Pareto manifold is a smooth manifold. By connecting with the simplex to introduce an (approximate) stationarity condition for the Pareto-constrained optimization, the papers shows that any non-trivial, local stationary condition requires more than first-order information on the manifold. Further, the paper proposes Pareto Majorization-Minimization Algorithm which converges to an approximate preference stationary point of Pareto-constrained optimization with proved complexity.

The paper is well written, and the method is novel. The solved problem is also interesting and challenging.

**Questions:**

a. The Related Work section clarifies that existing stationary conditions are overly restrictive or non-necessary. Is it possible to provide more detailed comparison in both theoretical and empirical perspectives?
b. Will the correlation between objectives impact the PMM's performance? For instance, what if two objectives are conflicting?

**Ethical Concerns:**

["NO or VERY MINOR ethics concerns only"]

**Final Justification:**

I appreciate the authors' feedback, and would like to maintain my rating.

**Limitations:**

The paper is purely theoretical. While there is some justification that the main goal is theory, the absence of any synthetic or motivating example to illustrate how the PMM algorithm performs in practical problems and visualize convergence or solution quality. For example, running the proposed algorithm on toy quadratic objectives.

**Paper Formatting Concerns:**

No concern

**Quality:**

4

**Strengths And Weaknesses:**

Strengths: The paper describes the motivation, problem setting, and major techniques clearly.

Weaknesses: Beyond what's stated in the conclusion, are there intuitions derived from the developed results about how the framework might break down or need to be adapted in nonconvex objective settings?

Though the paper is a theory paper, it would be good to include a practical example to visualize how the algorithm works and how the algorithm compares to other multi-objective optimization approaches.

---

> ### Author Rebuttal · Authors · 2025-07-30
>
> Thank you for your careful and positive review.
>
> **Generalization to non-convex objectives**
>
> Extending to the non-convex setting is definitely an important future direction, and one we’ve found to be quite interesting and challenging. Some challenges:
>
> 1. Without strict convexity, linear scalarization can miss out on Pareto optimal solutions: not every Pareto solution $x^*$ is a (local) minimizer of $f_\beta$ for some convex combination $\beta \in \Delta^{n-1}$ of the objectives.
>
> 2. In fact, the Pareto optimal points can unintuitively even correspond to local maximizers of $f_\beta$! This can happen even if the objectives are quasiconvex (thus, not non-convex in an arbitrary way; an example below).
>
> 3. And if the objectives are arbitrarily non-convex, we also lose the diffeomorphism between the Pareto manifold and the simplex (though local diffeomorphisms may remain).
>
> We have some ideas and are pursuing this direction, but it would probably be better to present results there in a separate paper.
>
> *Example of Pareto optimal point that is a local maximizer*.
>
> Let $q_1, q_2 : \mathbb{R} \to \mathbb{R}$ be the quadratics $q_1(x) = (x - 1)^2$ and $q_2 = (x + 1)^2$. And let us aim to minimize the objectives $f_1, f_2 : \mathbb{R} \to \mathbb{R}$, where $f_i = \sigma \circ q_i$, and $\sigma$ is any strictly monotonic increasing function. Then, the Pareto set of $(f_1, f_2)$ coincides with the Pareto set of $(q_1, q_2)$, which is the interval $[-1, 1]$.
>
> Consider $\sigma(z) = - \exp(- z/T)$, where $T > 0$ is some temperature parameter (the objectives $f_i$ look like upside-down Gaussians). For sufficiently small $T$, the Pareto optimal solution $x = 0$ will be the local *maximizer* of the weighted objective $\frac{1}{2} f_1 + \frac{1}{2} f_2$ (which looks like an upside-down mixture of Gaussians).
>
>
> **Visualization of algorithm/comparison with prior definitions of stationarity**
>
> We agree that visualizing the algorithm would be helpful, especially in helping demonstrate the issues with earlier definitions of stationarity. We will update the paper with experiments and visualizations.
>
>
> **Impact of (anti)-correlation across objectives**
>
> Our convergence guarantees are independent of the correlation across objectives. While we’re not ruling out the possibility that (anti)-correlation can have some effect on convergence speed, from our analysis, it seems that the more fundamental quantities are the condition number of Hessians $\nabla^2 f_i$ for each of the objectives.
>
> In fact, in the extreme zero-sum setting, where two objectives are perfectly anti-correlated $f_1 = - f_2$, every point is Pareto optimal. In that case, the preference optimization over the Pareto set is “easy” since it becomes an unconstrained optimization problem (of course, this example is excluded from our setting since we require all objectives to be strongly convex).
>
> In any case, we think this is a good type of question to pursue and refine in further research, since the goal of multi-objective optimization is to provide guidance on how to navigate conflicting objectives.

---

> > ### Comment · Reviewer_9L4B · 2025-08-05
> >
> > Thank you for the further clarifications. My previous concerns have been addressed, and I acknowledge the theoretical contributions of this work.

---

> ### Author Response · Authors · 2025-08-06
>
> Thank you. We appreciate it.
>
> Authors

---

### Official Review · Reviewer_H4W1 · 2025-07-11

**Clarity:** 3
**Significance:** 3
**Originality:** 3
**Rating:** 4
**Confidence:** 4

**Summary:**

This paper studies the problem of selecting the pareto optimal solutions among potential many candidates for multi-objective optimization. The authors formulate the problem as Pareto-constrained optimization to optimize a preference function constrained to a set of Pareto optimal solutions.  Subsequently, the authors propose a reformulation of the problem where the constraint set is redefined in terms of an appropriately defined manifold, which allows the authors to introduce an algorithm with a last-iterate convergence rate of $O\left(K^{-1 / 2}\right)$ to stationarity under strong assumptions.

**Questions:**

1. Given the strong assumptions (A, B, C), Can the authors provide concrete examples that meet these assumptions so that people who read the paper can better understand these assumptions? Without these, it is difficult to understand how practical of these problems? For example, Assumption A itself excludes lots of practical settings, such as the Portfolio optimization problem mentioned in the motivating examples, which has a linear objective: Let $f_1(x)$ be the expected return, where $f(x)=\sum_{i=1}^n r_i x_i$
Let $f_2(x)$ be the expected risk, where $f_xx)=\sum_{i=1}^n \sum_{j=1}^n \sigma_{i j} x_i x_j$

2. Can the authors provide a concrete example of $f_0(x)$? As a potential user of the proposed Pareto majorization-minimization algorithm, one needs to have a sense of when and how to cast the problem into the preference setting of this paper. For example, the authors mention that in fairness-aware learning,  $f_0$ is the social welfare function; In neural architecture search, $f_0$ can be a user-specific preference reflecting trade-offs (e.g., prioritizing low latency). Can the authors mathematically formulate them?

3. The authors mention that the primary motivation is to seek the most preferred solution from a large model like LLM that is retrained to satisfy a number of objectives. However, I doubt that these objectives, if any, behind the pre-training of any LLMs satisfy the assumptions (e.g., strongly convex, smooth, etc) in the paper. I'd like to see to what extent these settings and theoretical results resonate with the motivations.

**Ethical Concerns:**

["NO or VERY MINOR ethics concerns only"]

**Final Justification:**

I will raise my score to 4. Thanks for the answers to my questions, especially those examples in 1 & 2. While I appreciate the ongoing work for extending the theories from convex to non-convex, I can not raise my score to 5 as the primary motivation is to seek the most preferred solution from a large model like LLM that is pretrained to satisfy many desiderata and the theory is not about that thus far.

**Limitations:**

yes

**Quality:**

3

**Strengths And Weaknesses:**

Strengths

The paper studies a practical problem of selecting the pareto optimal solutions among potential many candidates.  The authors proposed Pareto Majorization-Minimization Algorithm with theoritical guarantees under restricted assumptions that converges to an ( $\varepsilon_0, \varepsilon$ )-approximate preference stationary point of Pareto-constrained optimizationwith iteration complexity $O\left(\varepsilon_0^{-2}\right)$.

Weaknesses

My major comments on the weaknesses of the paper are about the practicality of the settings (e.g., assumptions). Specially,

1. Given the strong assumptions (A, B, C), Can the authors provide concrete examples that meet these assumptions so that people who read the paper can better understand these assumptions? Without these, it is difficult to understand how practical of these problems? For example, Assumption A itself excludes lots of practical settings, such as the Portfolio optimization problem mentioned in the motivating examples, which has a linear objective.

2. Can the authors provide a concrete example of $f_0(x)$? As a potential user of the proposed Pareto majorization-minimization algorithm, one needs to have a sense of when and how to cast the problem into the preference setting of this paper. For example, the authors mention that in fairness-aware learning,  $f_0$ is the social welfare function; In neural architecture search, $f_0$ can be a user-specific preference reflecting trade-offs (e.g., prioritizing low latency). Can the authors mathematically formulate them?

3. The authors mention that the primary motivation is to seek the most preferred solution from a large model like LLM that is retrained to satisfy a number of objectives. However, I doubt that these objectives, if any, behind the pre-training of any LLMs satisfy the assumptions (e.g., strongly convex, smooth, etc) in the paper. I'd like to see to what extent these settings and theoretical results resonate with the motivations.

---

> ### Author Rebuttal · Authors · 2025-07-31
>
> Thank you to Reviewer H4W1 for taking the time to review and for the detailed feedback. It seems that the reviewer found the problem we study well-motivated, but is mainly concerned about the practicality of the setting we study theoretically. We address this first, and more specific concerns below.
>
> **Main contributions: theoretical vs. practical**
>
> We agree that this work’s main contributions are theoretical, although practical problems such as LLM fine-tuning serves as a key motivation for our framework. But, in developing a principled approach, we found the problem to be highly non-trivial due to the non-convexity, non-smoothness, and the implicit nature of the constraint set (even when the objectives are simply quadratics). We greatly simplify the problem conceptually.
>
> Prior to our work, there were only heuristics for this problem. The literature lacked a proper definition of stationarity and algorithms with provable behavior. Actually, intuitive heuristics easily go wrong. We proved that there is a large class of very natural stationary conditions—including one in existing literature, and another we had initially considered—that will *reject* the true preference-optimal solution. Practically, this means that the prior algorithm in literature *provably avoids* the correct solution, even in the strongly convex setting!
>
> In contrast, our work is the first to establish a well-justified solution concept that is efficiently and provably achieved by a simple algorithm, under two forms of feedback. The issue of practicality, especially in the context of LLM fine-tuning, is certainly a valid one. But, we believe that our work already makes a significant and meaningful contribution to the literature, and that it provides a solid theoretical foundation upon which to develop algorithms that work under less restrictive conditions, and that are more sample and computationally efficient.
>
>
> **Question 1: concrete examples of objectives**
>
> *Portfolio optimization*
>
> Let $x=[x_1, x_2, \cdots, x_n]^\top$ and let $\Sigma\in\mathbb{R}^{n\times n}\succ \mu I$ ($\mu>0$), i.e.,  $\Sigma$ is positive definite.
>
> Consider the objectives: uncertainty regularized expected return $f_a(x)=-\sum_{i}r_ix_i+ x^\top \Sigma x$  (In your notation $-f_1+f_2$, the minus sign is needed as we are minimizing)[1], and the quadratic transaction cost $f_b(x)=\|x-x_p\|_2^2$ [2,3]. $\nabla^2 f_a(x)=\Sigma\succ \mu I$, and $\nabla^2 f_b(x)=I\succ I $.
>
> Then, the norm of Σ is bounded as $\sigma_{ij}$ are constants. Since, $\nabla^2 f_a(x),\nabla^2 f_b(x)$ are constants it satisfies Assumption B. Since Assumption C is on $f_0$, we address it in our answer to your Q2.
>
> *Classification with sub-populations*
>
> Consider classification using regularized logistic regression where the population has $k$ subpopulations, i.e., $(x_j^i,y_j^i) \sim D_i$ for $j = 1,\ldots, n$ where $D_i$ is the data distribution of the $i$-th subpopulation. This is a $k$ objective MOO where $f_i(\theta)=\frac1n\sum_{j=1}^n\log (1+\exp(-y_j^i{x_j^i}^\top\theta))]+\|\theta\|_2^2$, $i=1,\cdots,k$. Then, $(2+m_2^i)I\succ \nabla^2f_i(\theta)\succ 2I $ which implies Assumption A, and $\nabla^2 f_i(\theta)$ is $0.1m_3^i$-Lipschitz continuous implying Assumption B where $m_2^i,m_3^i$ are 2nd and 3rd sample moments of the $i$-th population.
>
> [1]DeMiguel et. al., "Optimal versus naive diversification: How inefficient is the 1/N portfolio strategy?." The review of Financial studies 22, no. 5 (2009): 1915-1953.
>
> [2]Chen et. al., "A note on portfolio optimization with quadratic transaction costs." arXiv preprint arXiv:2001.01612 (2020).
>
> [3]DeMiguel et. al., "Multiperiod portfolio optimization with general transaction costs." Available at SSRN (2013).
>
> [4]Han et. al., "Merton’s portfolio problem under Volterra Heston model." Finance Research Letters 39 (2021): 101580.
>
> [5]Tong et. al., "A smoothing method for solving portfolio optimization with CVaR and applications in allocation of generation asset." Applied Mathematics and Computation 216, no. 6 (2010): 1723-1740.
>
> [6]Zafar et. al., Fairness Beyond Disparate Treatment \& Disparate Impact (2017).
>
>
> **Question 2: Concrete examples of preferences**
>
> *Portfolio optimization*
>
>  Several popular preference function $f_0$ choices satisfy Assumption C:
>
> 1. Constant Relative Risk Aversion in Merton’s portfolio problem, $f_0(x)=\frac{1}{\gamma}\exp(-x^\top r)$, $\gamma>0$ [4].
>
> 2. Smoothed version of Conditional Value-at-Risk (CVaR) $f_0(x) = \min_{\zeta \in \mathbb{R}} \{ \zeta + \frac{1}{\alpha} \mathrm{E} [ \phi(-x^\top r - \zeta) ] \}
> $ where $\phi(u) = \frac{1}{\beta} \log(1 + e^{\beta u})$[5].
>
>
> *Classification with sub-populations*
>
> There are several social utility functions that satisfy Assumption C:
>
> 1. User-defined social utility for public policy making $f_0 = \sum_{i=1}^n w_i^* f_i(\theta)$ where $w_i^*$ for $i = 1,\ldots, k$ are a given set of weights.
>
> 2. Demographic parity fairness $f_0(\theta)=\mathrm{E}[(z-E(z))(\sigma(\theta^\top x)-E(\sigma(\theta^\top x)))]$ where $\sigma(\cdot)$ is the logit function [6].
>
>
> **Question 3: practicality of assumptions**
>
> We agree that it is of practical importance to extend our work to less restrictive settings, especially to non-convex objectives. However, this criticism can in a sense be leveled against the whole field of (single-objective) convex optimization. But, as it is with convex optimization, here we have found that even the strongly convex setting is already theoretically very rich, and that it has helped us to significantly deepen our understanding of the nature and challenges of multi-objective optimization.
>
> We are already working on extending these results to the non-convex setting, and we’ve found it to be quite interesting and challenging (see also our response to Reviewer 9L4B for specifics). We believe that the completion of those results would be substantial enough to be published in a separate paper; this paper is already packed with many results.
>
> Of course, we welcome differences of opinions from the reviewer, but if so, we would really appreciate it if the reviewer would also articulate more specifically what results would be needed to make this work ready for publication.

---

> ### Author Response · Authors · 2025-08-06
>
> Dear Reviewer,
>
> Since the discussion period is nearing the end, we wanted to make sure that our response fully addresses your questions. If you have any further questions/concerns we would love to address that. If we have answered your questions satisfactorily, please let us know.
>
> Best,
> Authors

---

### Note · Authors · 2025-08-13

We thank the reviewers for their constructive feedback and summarize our main contributions and discussions below.

**Contributions**

1. Our work proposes *the first algorithm with a provable convergence rate of $O(\epsilon_0^{-2})$* to a meaningful notion of stationarity for the problem of preference function $f_0$ optimization over Pareto Set.

2. Our definition of approximate preference stationarity (Def. 5) is meaningful: Prop. 6 gives it an optimization-theoretic basis, Lem. 7 shows it is verifiable with approximate information emphasizing the "usability" of the solution concept. Moreover, we showed the insufficiency of first-order information explaining the need for second-order terms in this notion of stationarity.

We acknowledge our main contribution is theoretical. The suggested changes—adding references and toy experiments—are feasible within the deadline, as reviewers have recognized our theoretical contribution and requested no new theoretical results.

&nbsp;

**Technical Challenges**

1. Defining a meaningful stationarity condition. We resolve this issue by reframing the optimization over the Pareto manifold. We will compare with the stationarity notions introduced in the semivectorial optimization literature as suggested by Reviewer YrFB.

2. The implicit nature of the constraint set (Pareto set) as the map $x^*(\beta)$ is unknown. We overcome this issue by defining a majorizing surrogate (see Section 5.1).

&nbsp;

**Responses to Key Feedback**

_Practicality_: We provide two practical examples satisfying our assumptions (details in our reply to Reviewer H4W1).

1. Portfolio Optimization.
 - Objectives: uncertainty regularized expected return, transaction cost,
 - Preference: Risk Aversion or CVaR.

2. Classification with sub-populations.
 - Objectives: Sub-population losses.
 - Preference: User-defined social utility for public policy making or Demographic parity fairness.


_Toy Experiments_: We will include the toy experiments and compare them against PNG in the final version.

_Missing References_: We thank Reviewers i8uk and YrFB for the suggested references and will add comparisons to better contextualize our work.

_Generalization to non-convex objectives_: We restrict ourselves to the strongly convex case as it poses considerable theoretical challenges. Extending to nonconvex objectives is non-trivial (see our reply to Reviewer i8uk). We leave this to future work.

---

### Decision · Program_Chairs · 2025-09-17

**Decision:**

Accept (poster)

**Comment:**

This paper introduces a new algorithm with a provable convergence rate for preference function optimization over the Pareto set, supported by a meaningful notion of approximate preference stationarity. The theoretical contributions are solid, including the insight that first-order information is insufficient, motivating the use of second-order terms. While reviewers noted missing references, lack of toy experiments, and limited discussion of practical motivation, the rebuttal satisfactorily addresses these issues: the authors commit to adding comparisons, provide concrete examples of practical relevance, and will include illustrative experiments. Overall, the paper makes a clear and rigorous theoretical advance, and I support acceptance.